# Learning Reconfigurable Representations for Multimodal Federated Learning with Missing Data

**Duong M. Nguyen**[†]
University of Illinois Urbana-Champaign, US
nmduongg@illinois.edu

**Trong Nghia Hoang**[†]
Washington State University, US
trongnghia.hoang@wsu.edu

**Thanh Trung Huynh**
VinUniversity, Vietnam
trung.ht@vinuni.edu.vn

**Quoc Viet Hung Nguyen**
Griffin University, Australia
henry.nguyen@griffith.edu.au

**Phi Le Nguyen**[†]
Hanoi University of Science and Technology, Vietnam
lenp@soict.hust.edu.vn

## Abstract

Multimodal federated learning in real-world settings often encounters incomplete and heterogeneous data across clients. This results in misaligned local feature representations that limit the effectiveness of model aggregation. Unlike prior work that assumes either differing modality sets without missing input features or a shared modality set with missing features across clients, we consider a more general and realistic setting where each client observes a different subset of modalities and might also have missing input features within each modality. To address the resulting misalignment in learned representations, we propose a new federated learning framework featuring locally adaptive representations based on learnable client-side embedding controls that encode each client's data-missing patterns.

These embeddings serve as reconfiguration signals that align the globally aggregated representation with each client's local context, enabling more effective use of shared information. Furthermore, the embedding controls can be algorithmically aggregated across clients with similar data-missing patterns to enhance the robustness of reconfiguration signals in adapting the global representation. Empirical results on multiple federated multimodal benchmarks with diverse data-missing patterns across clients demonstrate the efficacy of the proposed method, achieving up to 36.45% performance improvement under severe data incompleteness. The method is also supported by a theoretical analysis with an explicit performance bound that matches our empirical observations. Our source codes are provided at https://github.com/nmduonggg/PEPSY

## 1 Introduction

Due to the rapid advances in IoT technologies [4, 23] and growing concerns over privacy protection [52], there are now numerous emerging multimodal federated learning (MMFL) scenarios in which clients observe different subsets of input modalities and must collaborate to train a common model without sharing data. These scenarios introduce two interrelated data-missing events: (1) clients may have access to only a subset of feature modalities [8, 45] (e.g., one device collects audio while another collects physiological signals), and (2) inputs within each modality

---

[†] Corresponding authors: Duong M. Nguyen, Trong Nghia Hoang, Phi Le Nguyen.

39th Conference on Neural Information Processing Systems (NeurIPS 2025).

may be partially missing due to sensor failures or intermittent recording [64, 39]. These challenges fundamentally disrupt the implicit assumption of traditional federated learning (FL) methods [38, 22, 21, 66, 67, 32, 51, 31, 11, 40, 37, 16, 50, 61, 58, 41], which presume that all local models are trained on a common set of feature modalities.

**Challenge.** When local models are optimized over different feature subsets, they tend to map inputs into incompatible representation spaces. Aggregating such models without proper alignment risks collapsing informative representations into entangled or degraded ones, ultimately reducing global performance. This problem is further exacerbated by heterogeneous data-missing patterns across clients, both in terms of available modalities and partial input observations [42, 56] (see Fig. 1). These compounded patterns are common in real-world applications such as wearable health monitoring, distributed environmental sensing, and smart infrastructure, where data collection is increasingly decentralized and sensor failures occur more frequent. Effectively addressing both missing modalities and missing features is essential for enabling next-generation distributed computing infrastructures, where learning must operate over heterogeneous, fragmented, and privacy-preserved data sources.

**Limitation of Prior Work.** Despite growing interest in multimodal and federated learning, most existing work focuses on idealized settings where all clients observe the same set of modalities. As a result, the general MMFL setting, where both events of data-missing occur, remains largely unaddressed. Existing approaches can be grouped into the following directions:

First, several efforts extend FL to multimodal inputs by designing universal representations [65, 60, 6, 43, 46], but they assume all clients observe the same modalities, ignoring modality heterogeneity. Second, centralized data imputation methods, including heuristic imputation [68, 69], neural imputation [9, 57, 18, 14, 12, 19], deterministic reconstruction using available modalities [63, 7, 47, 36, 20, 44, 39], and generative approaches [17, 25, 2, 62, 30, 29], require access to all data-missing patterns to ensure consistent imputation, and thus cannot be applied to federated settings. Third, it is also possible to leverage pre-trained multimodal foundation models (FMs) [12, 19, 1, 5, 48, 59, 33] to provide consistent data imputation, but in many scientific domains such as healthcare, there is no FM that spans all feature modalities. Most recently, a few recent FL-specific works [8, 45, 64, 39] begin to investigate these data-missing challenges in isolation. However, when both modalities and input features are missing, these methods fail to achieve satisfactory performance (see Section 4).

**Fundamental Gap.** In hindsight, what remains missing in these approaches is a mechanism to capture and communicate how each client's local view of the data is shaped by its specific patterns of missing information. Since the server cannot observe the training data, it lacks the context needed to align or reconfigure representations for any particular client. Conversely, each client is only aware of its own data-missing context and cannot fully interpret or adapt the aggregated representation to its local setting. This reveals the need to learn a shareable data-missing profile for each client, which summarizes the characteristics of its local data-missing patterns, providing more specific instructions to reconfigure the shared model towards local data contexts.

**Solution Vision.** The above reasoning motivates the following key insight and hypothesis. It is possible to learn and internalize specific traits in each client's data-missing profile into a set of embedding controls which can be used to reconfigure the shared model towards the local context. In this view, embeddings with similar content can also be aggregated which enables collaboration among clients with similar data-missing profiles. This design enables each client to adapt the shared model to its own incomplete data view, without requiring data sharing or retraining, providing a robust solution to multimodal federated learning with missing data.

**Technical Contributions.** To substantiate the above vision, we have made the following contributions:

**1.** We develop a new multimodal federated learning framework (PEPSY) with a client-side design that encodes the characteristic traits of each client's feature modalities, data specifics, and data-missing patterns into a set of local embedding controls. These local embeddings are communicated to the server, where they can be aligned and aggregated to capture commonalities across clients with similar data-missing profiles. The aggregated embeddings then serve as instructions to reconfigure the shared representation in a manner that is adaptive to each client's local context (Section 2).

**2.** We develop a rigorous theoretical analysis which establishes a direct bound on the expected performance of PEPSY over random patterns of missing data in terms of the training loss, demonstrating its stable performance and highlight the effectiveness of the proposed method (Section 3).

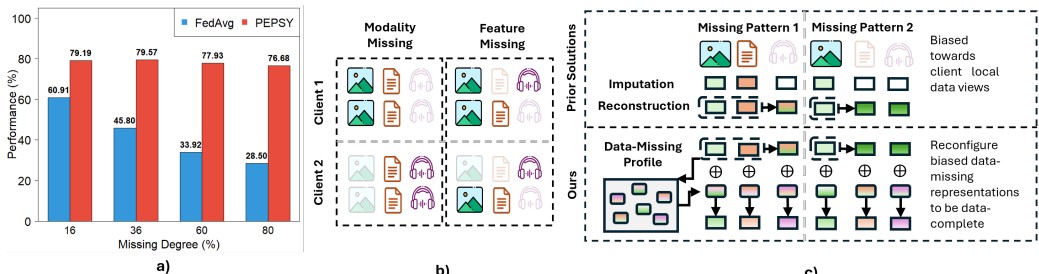

**Figure 1:** From left to right: **(a)** Performance comparison showing that FedAvg degrades rapidly with increasing missing data, while our framework PEPSY remains robust; **(b)** Illustration of two types of data-missing events in MMFL systems: (1) missing modalities and (2) missing input features; **(c)** Conceptual illustration highlighting the key distinction between our approach and prior work (see Section 2).

**3.** We evaluate the performance of our proposed framework against existing baselines through extensive experiments on the PTBXL [53] and SleepEDF [24] datasets. The results show that our method consistently outperforms existing baselines across numerous multi-modal data missing scenarios, establishing new SOTA performance in multimodal federated learning (Section 4).

## 2 Multimodal Federated Learning (MMFL) with Missing Data

### 2.1 Problem Formulation and Method Overview

**Standard Problem Formulation.** In a MMFL system, there are $K$ clients, each with a local dataset $\mathcal{D}_k$ consisting of $|\mathcal{D}_k|$ multimodal observations $(\boldsymbol{x}_d, \boldsymbol{y}_d)$, where $\boldsymbol{x}_d$ denotes the input instance and $\boldsymbol{y}_d$ represents the corresponding label. Each instance $\boldsymbol{x}_d$ may miss some modalities, represented by a missing set $\mathcal{S}_d \subset \mathcal{M}$, where $\mathcal{M}$ is the full set of modalities. The goal is to learn a global model $\boldsymbol{\theta}^*$ by minimizing the following loss function:

$$\boldsymbol{\theta}^* = \underset{\boldsymbol{\theta}}{\operatorname{argmin}} \frac{1}{K} \sum_{k=1}^{K} \ell_k(\boldsymbol{\theta}), \text{ with } \ell_k(\boldsymbol{\theta}) \triangleq \mathcal{L}\big(f(\mathcal{D}_k; \boldsymbol{\theta})\big), \tag{1}$$

where $f(\mathcal{D}_k; \boldsymbol{\theta})$ denotes multimodal prediction model with paramterer $\boldsymbol{\theta}$ over dataset $\mathcal{D}_k$, and $\mathcal{L}\big(f(\mathcal{D}_k; \boldsymbol{\theta})\big)$ is an average loss of $\boldsymbol{\theta}$ over dataset $\mathcal{D}_k$. Following [8, 45, 64, 39], $\boldsymbol{\theta}$ can be decomposed into two main modules: feature extractor $\boldsymbol{\theta}_e$ and post-processing head (including fusion and prediction) $\boldsymbol{\theta}_p$. Accordingly, $f$ can be expressed as $f(\mathcal{D}_k; \boldsymbol{\theta}) \triangleq f_p\big(f_e(\mathcal{D}_k; \boldsymbol{\theta}_e); \boldsymbol{\theta}_p\big)$, where $f_e(\cdot)$ denotes the feature extractor and $f_p(\cdot)$ represents the post-processing head.

**Reconfigured Problem Formulation.** As each client in MMFL only observes its own data-missing local view, the representations it produces are potentially biased. Based on this, we introduce a so-called *data-missing profile* $\Psi$, with $\tau$ *embedding controls*, i.e., $\Psi \triangleq \{\boldsymbol{\psi}_p\}_{p=1}^{\tau}$, to reconfigure these biases into data-complete features. This results in $f(\mathcal{D}_k; \boldsymbol{\theta}, \Psi)$ as a reconfigured version of original prediction model,

$$f(\mathcal{D}_k; \boldsymbol{\theta}, \Psi) \triangleq f_p\Big(f_e(\mathcal{D}_k; \boldsymbol{\theta}_e) \circ r(\mathcal{D}_k; \Psi); \boldsymbol{\theta}_p\Big), \tag{2}$$

where $\circ$ denotes set concatenation, and $r(\mathcal{D}_k; \Psi)$ represents a so-called *relevance function* that returns relevant embeddings for each $\boldsymbol{x}_d \in \mathcal{D}_k$. Intuitively, this relevance function captures missing pattern information needed to reconfigure instances in $\mathcal{D}_k$, which can be learned by rewriting Eq. 1 as:

$$\boldsymbol{\theta}^*, \Psi^* = \underset{\boldsymbol{\theta}, \Psi}{\operatorname{argmin}} \frac{1}{K} \sum_{k=1}^{K} \Big\{ \ell_k(\boldsymbol{\theta}, \Psi) - u_k(\boldsymbol{\theta}, \Psi) \Big\}, \tag{3}$$

$$\text{where, } \ell_k(\boldsymbol{\theta}, \Psi) \triangleq \mathcal{L}\big(f(\mathcal{D}_k; \boldsymbol{\theta}, \Psi)\big) \text{ and } u_k(\boldsymbol{\theta}, \Psi) \triangleq \mathcal{R}\big(r(\mathcal{D}_k; \Psi)\big). \tag{4}$$

where $\mathcal{R}$ estimates relevance between each instance in $\mathcal{D}_k$ and its embeddings selected by $r(\cdot)$. Intuitively, minimizing $\ell_k(\boldsymbol{\theta}, \Psi)$ leads to neural components $\boldsymbol{\theta}$ that extract data-missing features, which are reconfigured by $\Psi$ for predictions. Conversely, maximizing $u_k(\boldsymbol{\theta}, \Psi)$ enables $\Psi$ to adapt to local context, effectively distilling missing patterns.

**Method Overview.** An overview of our proposal is in Fig. 2a. Formally, PEPSY operates over multiple communication rounds, each consisting of client-side training and server-side aggregation.

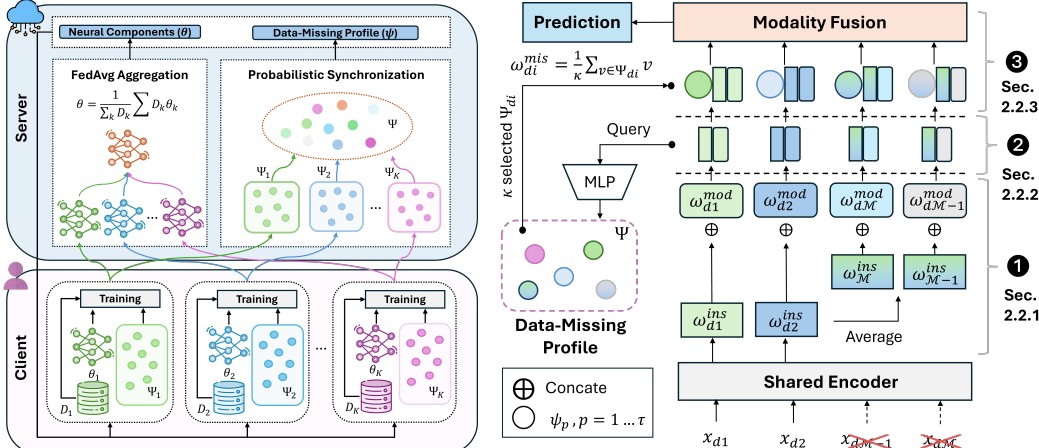

**(a) Overall Workflow.** PEPSY has two stages: client training and server aggregation. After local training, client parameters are sent to the server to perform aggregation, which includes FedAvg [38] and probabilistic synchronization.

**(b) Client Design.** Each client ❶ extracts modality- and data-specific features ($\boldsymbol{w}^{\text{ins}}$, $\boldsymbol{w}^{\text{mod}}$), then ❷ queries the data-missing profile $\Psi$ to form $\boldsymbol{w}^{\text{mis}}$ as the missing-pattern feature. ❸ Finally, $\boldsymbol{w}^{\text{mis}}$ reconfigures ($\boldsymbol{w}^{\text{mod}}$, $\boldsymbol{w}^{\text{ins}}$) into data-complete features for downstream tasks.

**Figure 2:** Overview of the overall server-client workflow of PEPSY and its client design.

On the client side, each client ❶ extracts information from its local dataset, potentially with some modalities missing, and ❷ leverages the extracted information to select relevant embeddings from $\Psi$ for each instance $\boldsymbol{x}_d$, thereby ❸ constructing data-complete representations. Further details are provided in Sections 2.2.1 and 2.2.2, respectively. To ensure final representations are faithfully data-complete, we enforce these features to be comparable with full-modality features before fusion and prediction (see Section 2.2.3). On the server side, due to variable size of data-missing profile per client, we treat the data-missing profile aggregation as a non-parametric clustering problem, as presented in Section 2.3[0]. This process repeats for $T$ rounds until convergence.

## 2.2 Client Design

This section explains how clients learn the data-missing profile and use it to reconfigure biases caused by limited local data views. An overview of the client design is shown in Fig. 2b.

### 2.2.1 Data-Missing Representations

**Intuitions.** In the presence of missing modalities, the information within a multimodal instance can be decomposed into three components: modality-specific (distinguishing different modalities); data-specific (capturing the integrity of the individual instance); and missing-pattern information (distinguishing different missing patterns). Based on this decomposition, we extract these components to construct a comprehensive data-missing profile for each client.

Formally, given an instance $\boldsymbol{x}_d = \{\boldsymbol{x}_{di}, \forall i \in \mathcal{M} \setminus \mathcal{S}_d\}$ we first construct (1) **modality-specific** features $\{\boldsymbol{w}_{di}^{\text{mod}}\}$ and (2) **data-specific** features $\{\boldsymbol{w}_{di}^{\text{ins}}\}$. The former are represented by learnable embeddings $W^{\text{mod}} = \{\boldsymbol{w}_i^{\text{mod}}\}_{i=1}^{|\mathcal{M}|}$ to ensure data invariance and are shared across all instances, i.e., $\boldsymbol{w}_{di}^{\text{mod}} = \boldsymbol{w}_i^{\text{mod}} (\forall d)$. The latter are constructed by mapping and normalizing each observed modality $\boldsymbol{x}_{di} (\forall i \in \mathcal{M} \setminus \mathcal{S}_d)$ to corresponding representations, denoted as $\boldsymbol{h}_{di}$. For missing modalities, we use a common averaging approach [54, 28] to reconstruct their features, resulting in the formulation:

$$\boldsymbol{w}_{di}^{\text{ins}} \triangleq \mathbf{I}(i \notin \mathcal{S}_d)\boldsymbol{h}_{di} + \mathbf{I}\Big(i \in \mathcal{S}_d\Big)\Big(\frac{1}{|\mathcal{M}| - |\mathcal{S}_d|} \sum_{j \notin \mathcal{S}_d} \boldsymbol{h}_{dj}\Big), \tag{5}$$

where $\mathbf{I}$ depicts an indicator function. To ensure the feature reconstruction in Eq. 5 are truly data-specific, we introduce a data-specific loss that regularizes the features from the same instance's

---

[0]Other neural components can be aggregated effectively using FedAvg [38]

available modalities to be closer than those from different instances:

$$\mathcal{L}_{ds}(\boldsymbol{x}_d, S_d) \triangleq \sum_{i,j \notin S_d} - \log \frac{\exp(\tilde{\boldsymbol{h}}_{di} \tilde{\boldsymbol{h}}_{dj}^\top)}{\sum_{d_1, d_2 \neq d_1}^{|\mathcal{D}|} \sum_{k_1 \notin S_{d_1}, k_2 \notin S_{d_2}} \exp(\tilde{\boldsymbol{h}}_{d_1 k_1} \tilde{\boldsymbol{h}}_{d_2 k_2}^\top)}, \tag{6}$$

where $\tilde{\boldsymbol{h}}_{di}$ represents the $\ell_2$-normalized feature of $\boldsymbol{h}_{di}$. Intuitively, minimizing $\mathcal{L}_{ds}$ ensures $\boldsymbol{h}_{di}$ preserves instance identity across modalities while reducing the impact of missing patterns $\mathcal{S}_d$, thereby improving prediction consistency and stability. This is justified by the theorem in Section 3.

**Remark.** While $\boldsymbol{w}_{di}^{\text{mod}}$ encodes modality-specific information and $\boldsymbol{w}_{di}^{\text{ins}}$ captures data-specific details influenced by the missing pattern $\mathcal{S}_d$ (Eq. 5), together they comprehensively represent the data-missing information in $\boldsymbol{x}_d$. This combined information can be distilled into the data-missing profile, allowing future clients leverage similar data views to handle their local context.

### 2.2.2 Embedding Controls Selection

**Intuition.** Since data-missing features reflect the client's local missing patterns, learning data-missing profiles requires interaction between these features and embedding controls. We model this interaction as a query-key matching process that selects the most relevant embeddings for each instance to distill and reconfigure, formulating the final data-complete features. Details are below.

Given data-missing representations $(\boldsymbol{w}_{di}^{\text{mod}}, \boldsymbol{w}_{di}^{\text{ins}})$, $i \in \mathcal{M}$, from $\boldsymbol{x}_d$, we allow it to select the relevant embeddings from $\Psi$ for reconfiguration. The relevance between each modality $\boldsymbol{x}_{di}$ and a particular embedding control $\boldsymbol{\psi}_p$ ($p = 1 \ldots \tau$), denoted as $\gamma(\boldsymbol{x}_{di}, \boldsymbol{\psi}_p)$, is defined as follows:

$$\gamma(\boldsymbol{x}_{di}, \boldsymbol{\psi}_p) \triangleq e\left(\mathbf{q}(\boldsymbol{x}_{di}), \mathbf{k}(\boldsymbol{\psi}_p)\right), \tag{7}$$

where $e(\cdot, \cdot)$ depicts the cosine similarity, $\mathbf{q}(\boldsymbol{x}_{di}) \triangleq \text{MLP}^1([w_{di}^{\text{mod}} \circ \boldsymbol{w}_{di}^{\text{ins}}])$ fully captures data-missing information from $\boldsymbol{x}_d$. Here $\mathbf{k}(\cdot)$ is an identity function to distill the original information directly from $\boldsymbol{x}_d$ to $\boldsymbol{\psi}_p$, allowing accurate reconfiguration from $\boldsymbol{\psi}_p$ without distortion. To prevent the model from distributing data-missing information in $\boldsymbol{x}_d$ too broadly and diluting learned data-missing profile, we only allow $\kappa$ relevant embedding controls selected for each instance, with $\kappa \ll |\boldsymbol{\Psi}|$. To enforce this, we introduce a regularization term:

$$\mathcal{R} \triangleq \sum_d^{|\mathcal{D}|} \sum_i^{|\mathcal{M}|} \sum_{\boldsymbol{v} \in \Psi_{di}} \gamma(\boldsymbol{x}_{di}, \boldsymbol{v}), \tag{8}$$

where $\Psi_{di}$ is the set of the $\kappa$ most relevant embeddings for each modality $\boldsymbol{x}_{di}$ within the client's local data-missing profile. This regularizer encourages each instance to focus on a small, relevant subset of embedding controls, promoting more precise relevance assessment and better distillation. We use the averaged embedding to represent the whole selected set $\Psi_{di}$, resulting in missing-pattern representation $\boldsymbol{w}_{di}^{\text{mis}}$. The final representation is then formed as $\boldsymbol{w}_{di} = [\boldsymbol{w}_{di}^{\text{mod}} \circ \boldsymbol{w}_{di}^{\text{ins}} \circ \boldsymbol{w}_{di}^{\text{mis}}]$.

### 2.2.3 Reconfiguration Regularization and Modality Fusion

**Reconfiguration Regularization.** By leveraging the missing profile, we form the final representation $\boldsymbol{w}_{di}$ by concatenating three types of information $\boldsymbol{w}_{di}^{\text{mod}}$, $\boldsymbol{w}_{di}^{\text{ins}}$ and $\boldsymbol{w}_{di}^{\text{mis}}$. To ensure the final representation faithfully reflects the full-modality information of instance $\boldsymbol{x}_d$, we introduce a contrastive loss $\mathcal{L}_{\text{rc}}$ as a reconfiguration signal. This loss encourages the projected representations $\hat{\boldsymbol{w}}_{di}$ of $\boldsymbol{w}_{di}$ from the same instance $\boldsymbol{x}_d$ to be close (similar to Eq. 6). Intuitively, this regularization guides the data-missing embeddings to reshape representations into data-complete forms, hence ensuring effective reconfiguration signals. Note that $\hat{\boldsymbol{w}}_{di}$, $\forall i \in \mathcal{M}$, are used solely for regularization.

**Modality Fusion.** Since $\hat{\boldsymbol{w}}_{di}$ provides a high-level representation of the original feature, we leverage the similarity among $\{\hat{\boldsymbol{w}}_{di}\}$ as attention weights to fuse $\{\boldsymbol{w}_{di}\}$ together and form a so-called *cross-modal representation* $\{\hat{\boldsymbol{c}}_{di}\}$: Finally, we combine the cross-modal representation $\hat{\boldsymbol{c}}_{di}$ and the original representation $\boldsymbol{w}_{di}$ to obtain the final representation $\boldsymbol{c}_{di}$ of instance $\boldsymbol{x}_d$: $\boldsymbol{c}_{di} = \boldsymbol{\alpha}_{di} \hat{\boldsymbol{c}}_{di} + (1 - \boldsymbol{\alpha}_{di}) \boldsymbol{w}_{di}$, where $\boldsymbol{\alpha}_{di}$ is computed by a learnable function $s([\boldsymbol{w}_{di} \circ \hat{\boldsymbol{c}}_{di}])$, with $\circ$ denoting element-wise concatenation. The resulting representation $\boldsymbol{c}_{di}$ is then passed to the prediction head, ensuring that it captures both the completeness of the data and enriched cross-modal contextual information.

---

[1]MLP denotes a linear projector

**Training Objective**. After producing final prediction using a prediction head, the client model is evaluated by a task-specific loss function $\mathcal{L}_{task}$. Overall, the training objective for the local model is $\mathcal{L} \triangleq \mathcal{L}_{task} + \lambda(\mathcal{L}_{ds} + \mathcal{L}_{rc}) - \eta\mathcal{R}$, where $\lambda$ and $\eta$ are weighting coefficients that control the contributions of $\mathcal{L}_{ds}$, $\mathcal{L}_{rc}$, and the relevance $\mathcal{R}$, respectively.

### 2.3 Server Aggregation

While traditional server aggregation algorithms [38] can aggregate common neural components among clients, it struggles with our data-missing profiles due to alignment issues. Local data-missing profiles are learned in arbitrary orders across clients, leading to misalignment where identical embedding positions may represent different data-missing patterns. Consequently, directly merging these representations can produce suboptimal results. To overcome this, we frame data-missing profile alignment as a clustering task that groups embeddings from diverse client views into a global profile. Since each client may select a different number of embeddings within its data-missing profile $\psi$, this becomes a non-parametric clustering problem [58, 34, 27]. This study adopts PFPT [58] as the profile aggregation method, enabling the number of clusters to adapt dynamically to data complexity, or missingness level in our context. Each client refines the global profile using its private data, producing locally augmented controllers whose size and complexity reflect the client's missingness level. Using PFPT's non-parametric nature, the server clusters similar controllers and updates the global profile to reflect the missingness complexity of the whole system, which is then shared with clients for the next training round. This process allows PEPSY to align missingness profiles across clients and effectively handle heterogeneous data-missing patterns (see Section 4 for details).

## 3 Theoretical Analysis

Ideally, we expect the model's predictions to remain robust even in the absence of certain modalities. In this section, we present a theoretical analysis of the convergence behavior of our model's output for a given instance $x$ under two conditions: when all modalities are available and when some are missing. Specifically, we demonstrate that our training objectives are designed to minimize the discrepancy between these two prediction outcomes.

**Theorem 3.1** *Let $x \in \mathcal{D}$ be an arbitrary instance with a missing modality pattern $\mathcal{S} \subset \mathcal{M}$, where $\mathcal{M}$ denotes the full set of modalities. Suppose $y_x^{\mathcal{S}}$ and $y_x^{\emptyset}$ represent the model's outputs at test time when $x$ is missing modalities in $\mathcal{S}$, and when all modalities are present, respectively. Let $\mathbb{E}_{x,\mathcal{S}}$ denote the expectation over all instances $x$ and all possible missing patterns $\mathcal{S}$. Then, if the client model is $\mu$-Lipschitz continuous, the distance between $y_x^{\mathcal{S}}$ and $y_x^{\emptyset}$ can be bounded by the empirical training loss as follows:*

$$\mathbb{E}_{x,\mathcal{S}}[|y_x^{\mathcal{S}} - y_x^{\emptyset}|] \leq \mathcal{O}\left(\mu|S|\sqrt{\frac{\mathbb{E}_{x,\mathcal{S}}[\mathcal{L}_{ds}(x,S)]}{(|\mathcal{M}| - |S|)^2} + \log\frac{|\mathcal{M}|^2}{(|\mathcal{M}| - |S|)^2}}\right). \tag{9}$$

**Observation 1.** Theoretical analysis shows that the expected deviation caused by missing modality patterns $\mathcal{S}$ is controlled by our proposed loss $\mathcal{L}_{ds}$, which is directly minimized during training. Reducing $\mathcal{L}_{ds}$ lowers the model's dependency on missing data, tightening the theoretical error bound and ensuring stable, reliable predictions despite incomplete inputs. Thus, our loss design both mitigates the impact of missing modalities and improves generalization across diverse test conditions.

**Observation 2.** In the ideal case where the solution is optimal, i.e., $\mathbb{E}_{x,\mathcal{S}}[\mathcal{L}_{ds}(x,\mathcal{S})] = 0$, the right-hand side of Eq. 9 simplifies to $\mathcal{O}\left(\mu|\mathcal{S}|\sqrt{\log M^2 - \log(M - |\mathcal{S}|)^2}\right)$. When $\mathcal{S} \to \emptyset$, i.e., all modalities are available, both sides of the bound converge to zero as expected. In the worst-case scenario, where $|\mathcal{S}| = |\mathcal{M}| - 1$, the right-hand side becomes $\mathcal{O}\left(\mu(|\mathcal{M}| - 1)\sqrt{2\log|\mathcal{M}|}\right)$, depending only on the Lipschitz constant $\mu$. This aligns with the intuition that $\mathcal{L}_{ds}(\cdot, \cdot)$ minimizes modality discrepancies within a shared embedding space but does not constrain the model's global behavior, leaving the remaining deviation governed by the smoothness of the learned function, as reflected in $\mu$.

Overall, the stability of PEPSY to varying missing patterns depends on three factors: the alignment quality of data-specific features ($\mathcal{L}_{ds}(\cdot, \cdot)$), the number of missing modalities ($|\mathcal{S}|$), and the smoothness of the learned model, characterized by the Lipschitz constant $\mu$. Theorem 3.1 supports PEPSY's effectiveness in federated learning, which matches our empirical observation.

**Table 1:** Performance of baselines on the PTBXL and EDF datasets under various missing patterns in train and test sets, for both IID and Non-IID scenarios. The best and second-best results are highlighted in **bold red** and blue, respectively.

| Dataset | $p_m \backslash p_s$ | Method | IID | | | | | Non-IID | | | | |
|---|---|---|---|---|---|---|---|---|---|---|---|---|
| | | | 0.2 | 0.4 | 0.6 | 0.8 | 1.0 | 0.2 | 0.4 | 0.6 | 0.8 | 1.0 |
| PTBXL | 0.2 | FedProx [31] | 73.43 ± 0.38 | 73.64 ± 1.01 | 71.42 ± 1.18 | 71.37 ± 2.50 | 69.93 ± 4.61 | 54.01 ± 3.66 | 51.15 ± 5.30 | 50.06 ± 12.22 | 54.89 ± 1.54 | 44.17 ± 1.31 |
| | | MIFL [45] | 73.52 ± 1.45 | 70.95 ± 1.90 | 71.41 ± 1.46 | 56.66 ± 22.68 | 69.99 ± 3.05 | 50.99 ± 2.38 | 47.16 ± 3.16 | 49.39 ± 1.75 | 51.37 ± 2.55 | 50.78 ± 4.76 |
| | | FedInMM [64] | 69.78 ± 5.16 | 69.27 ± 3.21 | 66.16 ± 3.01 | 65.49 ± 2.25 | 65.45 ± 2.70 | 34.17 ± 6.82 | 40.48 ± 10.87 | 41.23 ± 11.34 | 40.52 ± 11.20 | 40.31 ± 10.70 |
| | | FedMSplit [8] | 54.84 ± 22.31 | 53.63 ± 21.72 | 52.12 ± 21.55 | 52.50 ± 21.52 | 55.84 ± 13.22 | 42.75 ± 3.56 | 42.58 ± 6.07 | 41.62 ± 6.06 | 40.27 ± 3.09 | 39.39 ± 1.66 |
| | | FedMAC [39] | 78.56 ± 0.47 | 77.30 ± 0.81 | 76.25 ± 0.49 | 75.49 ± 1.07 | 74.70 ± 0.83 | 58.26 ± 4.81 | 58.55 ± 3.02 | 54.98 ± 7.74 | 50.94 ± 1.25 | 48.38 ± 0.59 |
| | | PEPSY | 78.81 ± 0.72 | 77.43 ± 0.88 | 76.75 ± 1.47 | 76.13 ± 0.25 | 75.41 ± 0.82 | 71.45 ± 0.39 | 69.70 ± 2.08 | 66.92 ± 2.83 | 68.26 ± 2.56 | 66.75 ± 5.32 |
| | 0.8 | FedProx [31] | 72.76 ± 0.57 | 70.24 ± 1.61 | 68.77 ± 2.30 | 65.24 ± 4.94 | 33.79 ± 3.39 | 48.43 ± 1.25 | 42.08 ± 0.53 | 34.17 ± 3.14 | 27.32 ± 1.67 | 29.97 ± 1.31 |
| | | MIFL [45] | 69.90 ± 1.14 | 65.36 ± 2.12 | 55.44 ± 6.44 | 50.61 ± 14.99 | 35.39 ± 6.90 | 44.26 ± 3.87 | 37.75 ± 12.67 | 32.67 ± 8.82 | 28.12 ± 6.03 | 29.67 ± 2.54 |
| | | FedInMM [64] | 63.10 ± 2.77 | 61.92 ± 1.53 | 60.36 ± 0.16 | 56.95 ± 2.13 | 35.31 ± 13.56 | 49.81 ± 17.45 | 46.41 ± 14.99 | 42.95 ± 12.72 | 42.37 ± 12.21 | 36.70 ± 14.23 |
| | | FedMSplit [8] | 54.77 ± 20.66 | 49.56 ± 18.20 | 45.82 ± 16.29 | 43.97 ± 15.87 | 23.91 ± 2.18 | 51.03 ± 2.09 | 44.51 ± 0.77 | 38.25 ± 4.49 | 29.91 ± 6.11 | 28.33 ± 2.26 |
| | | FedMAC [39] | 74.25 ± 0.48 | 73.06 ± 0.65 | 70.36 ± 0.75 | 67.17 ± 2.98 | 41.51 ± 6.64 | 53.05 ± 0.41 | 51.03 ± 3.19 | 36.95 ± 0.18 | 45.90 ± 4.45 | 43.29 ± 1.54 |
| | | PEPSY | 76.25 ± 0.77 | 75.96 ± 1.82 | 76.42 ± 0.98 | 75.08 ± 1.65 | 45.07 ± 0.26 | 63.01 ± 3.95 | 65.40 ± 1.01 | 69.19 ± 0.16 | 60.40 ± 7.11 | 53.07 ± 2.66 |
| EDF | 0.2 | FedProx [31] | 44.08 ± 0.59 | 43.54 ± 0.62 | 43.99 ± 0.57 | 35.65 ± 12.22 | 34.02 ± 14.46 | 34.58 ± 13.80 | 44.61 ± 0.63 | 44.02 ± 0.30 | 32.25 ± 11.94 | 44.27 ± 0.34 |
| | | MIFL [45] | 44.19 ± 0.73 | 44.27 ± 0.96 | 43.15 ± 0.83 | 43.32 ± 2.19 | 43.54 ± 0.27 | 43.17 ± 1.76 | 43.35 ± 2.26 | 44.05 ± 0.35 | 32.74 ± 15.73 | 44.42 ± 0.33 |
| | | FedInMM [64] | 40.39 ± 0.14 | 40.39 ± 0.09 | 40.24 ± 0.11 | 40.33 ± 0.12 | 40.37 ± 0.21 | 40.99 ± 0.98 | 40.73 ± 0.57 | 40.46 ± 0.24 | 40.87 ± 0.94 | 40.43 ± 0.26 |
| | | FedMSplit [8] | 41.91 ± 2.31 | 36.47 ± 11.44 | 43.09 ± 2.20 | 43.77 ± 1.47 | 41.42 ± 2.80 | 42.95 ± 1.37 | 33.98 ± 14.43 | 42.88 ± 1.15 | 26.08 ± 13.54 | 43.43 ± 1.11 |
| | | FedMAC [39] | 39.00 ± 12.45 | 40.43 ± 10.29 | 41.85 ± 7.58 | 43.58 ± 5.47 | 43.01 ± 1.39 | 38.60 ± 12.32 | 39.44 ± 9.62 | 41.04 ± 6.87 | 43.13 ± 4.66 | 43.96 ± 1.80 |
| | | PEPSY | 48.76 ± 5.41 | 49.37 ± 4.43 | 48.70 ± 4.03 | 49.27 ± 3.30 | 46.87 ± 2.46 | 54.84 ± 3.32 | 50.28 ± 4.11 | 54.50 ± 0.14 | 51.07 ± 5.24 | 53.35 ± 6.13 |
| | 0.8 | FedProx [31] | 41.49 ± 3.69 | 31.15 ± 11.57 | 33.73 ± 4.92 | 19.72 ± 6.91 | 33.53 ± 14.10 | 43.87 ± 0.44 | 24.34 ± 14.02 | 34.56 ± 13.11 | 34.56 ± 12.99 | 34.17 ± 11.53 |
| | | MIFL [45] | 44.51 ± 0.45 | 42.25 ± 1.67 | 42.99 ± 0.91 | 41.07 ± 0.61 | 42.40 ± 1.65 | 43.42 ± 1.41 | 43.83 ± 0.90 | 43.01 ± 0.99 | 42.99 ± 1.00 | 42.40 ± 0.70 |
| | | FedInMM [64] | 40.31 ± 0.13 | 40.29 ± 0.11 | 40.26 ± 0.14 | 40.25 ± 0.02 | 40.52 ± 0.01 | 40.84 ± 0.77 | 40.81 ± 0.79 | 40.50 ± 0.37 | 40.31 ± 0.14 | 40.36 ± 0.22 |
| | | FedMSplit [8] | 41.44 ± 3.16 | 32.99 ± 13.22 | 42.21 ± 1.42 | 36.64 ± 6.21 | 43.02 ± 0.47 | 35.71 ± 10.75 | 42.75 ± 1.64 | 33.54 ± 13.70 | 41.87 ± 1.94 | 43.38 ± 0.50 |
| | | FedMAC [39] | 43.77 ± 1.52 | 42.54 ± 2.39 | 41.51 ± 0.73 | 41.80 ± 2.14 | 26.33 ± 1.47 | 46.01 ± 0.98 | 45.73 ± 0.99 | 45.66 ± 0.49 | 46.22 ± 0.84 | 34.21 ± 8.87 |
| | | PEPSY | 54.02 ± 1.41 | 49.02 ± 0.38 | 49.23 ± 1.47 | 52.78 ± 4.49 | 46.91 ± 3.70 | 48.95 ± 2.14 | 51.52 ± 0.60 | 50.97 ± .44 | 50.96 ± 1.99 | 46.07 ± 0.02 |

# 4 Empirical Evaluation

## 4.1 Experimental Settings

**Dataset and Missing Modality Simulation.** Our approach is evaluated on two datasets: PTBXL [53] (12 modalities) and Sleep-EDF [24] (5 modalities). Each dataset is split into 80% for training and 20% for testing, with the former distributed across $K$ clients in both IID and Non-IID settings. Following [39], we define $p_s$ as the ratio of samples with missing modalities, and $p_m$ as the ratio of missing modalities within those samples[2]. The *missing degree* is then defined as $p_m \times p_s$, representing the overall proportion of instances with missing modalities. Using these definitions, we simulate modality missing patterns by constructing a binary matrix $\phi(\mathcal{D}_k)$, where $\phi(\mathcal{D}_k)^{[i,m]} \in \{0, 1\}$ indicates whether modality $m$ is missing (0) or available (1) for sample $i$. The incomplete dataset $\hat{\mathcal{D}}_k = \mathcal{D}_k \odot \phi(\mathcal{D}_k)$, where $\odot$ denotes element-wise multiplication, is then used for the experiments. Details for modality missing patterns simulation is presented in Appendix B.

**Baselines and Evaluation Metrics.** We compare PEPSY with five baselines: FedProx [31], FedMSplit [8], MIFL [45], FedInMM [64], and FedMAC [39]. FedProx disregards missing modalities, FedMSplit and MIFL focus on modality-missing event, while FedInMM and FedMAC address feature-missing events. These baselines provide a comprehensive benchmark for evaluating our method. We use accuracy on the server's dataset as a performance metric for the whole system. Implementation details are provided in Appendix C.

## 4.2 Performance under Similar Missing Statistics between Training and Testing

**Results under the IID setting.** Table 1 shows that PEPSY consistently outperforms other methods in most experimental scenarios with varying missing statistics in IID settings. For the PTBXL dataset, when the missing degree is low (e.g., $p_m = 0.2$), the differences are minimal, with all methods achieving similar accuracy. However, as the missing degree increases (e.g., $p_m = 0.8$), PEPSY maintains a significant advantage, outperforming other methods. This trend is even more pronounced in the EDF dataset, where PEPSY surpasses the baselines by up to 11.67% in all missing scenarios. While most methods experience substantial performance drops, PEPSY remains robust, achieving the highest accuracy in 40/40 cases. This is because the data-missing profile provides an informative reconfiguration signal that reprograms feature construction for more robust predictions.

**Results under the Non-IID setting.** In the complex Non-IID setting, PEPSY again outperforms all other methods, as shown in Table 1. On the PTBXL dataset, PEPSY surpasses FedMAC and other approaches by nearly 15.83% in slightly missing scenarios ($p_m = 0.2$), maintaining its advantage even as missing patterns become more extreme, with 64.69% accuracy at $p_m/p_s = 0.8/0.8$. On

---

[2]A tuple $(p_m, p_s)$ is called *missing statistic*.

**Table 2:** Performance of baselines under various missing statistics, where the missing statistics of the clients and server are *different*.

| | | Method | Testing missing statistics ($p_m/p_s$) | | | | | | |
|---|---|---|---|---|---|---|---|---|---|
| | | | 0.2/0.2 | 0.4/0.4 | 0.6/0.6 | 0.8/0.8 | 1.0/0.4 | 0.6/1.0 | 0.8/1.0 |
| Training missing statistics ($p_m/p_s$) | 0.0/0.0 | FedProx [31] | 70.24% | 57.75% | 38.84% | 34.68% | 66.46% | 29.89% | 25.85% |
| | | MIFL [45] | 75.79% | 73.27% | 72.38% | 65.32% | 73.64% | 63.05% | 46.15% |
| | | FedInMM [64] | 77.18% | 73.90% | 68.98% | 55.86% | 72.63% | 51.70% | 38.97% |
| | | FedMSplit [8] | 70.24% | 57.76% | 38.84% | 34.68% | 66.46% | 29.89% | 25.85% |
| | | FedMAC [39] | 79.07% | 79.45% | 77.30% | 73.39% | 77.30% | 74.02% | 63.68% |
| | | PEPSY | 79.07% | 79.19% | 79.57% | 77.55% | 78.31% | 77.93% | 76.78% |
| | 1.0/0.5 | FedProx [31] | 77.05% | 75.66% | 74.02% | 66.84% | 74.40% | 69.61% | 54.15% |
| | | MIFL [45] | 73.77% | 74.02% | 72.38% | 66.58% | 73.90% | 70.62% | 62.55% |
| | | FedInMM [64] | 44.77% | 44.39% | 42.12% | 42.25% | 44.01% | 35.06% | 31.52% |
| | | FedMSplit [8] | 77.05% | 75.66% | 74.02% | 66.84% | 74.40% | 69.61% | 59.14% |
| | | FedMAC [39] | 75.91% | 76.55% | 76.04% | 72.51% | 75.79% | 73.01% | 69.99% |
| | | PEPSY | 77.68% | 75.66% | 77.18% | 75.91% | 75.91% | 74.40% | 74.15% |
| | 0.5/1.0 | FedProx [31] | 36.57% | 33.42% | 31.15% | 34.43% | 36.19% | 27.49% | 26.61% |
| | | MIFL [45] | 38.71% | 35.81% | 31.90% | 34.93% | 41.11% | 27.49% | 28.75% |
| | | FedInMM [64] | 53.47% | 50.06% | 45.02% | 39.34% | 54.86% | 44.52% | 36.70% |
| | | FedMSplit [8] | 36.57% | 33.42% | 31.15% | 34.43% | 36.19% | 27.49% | 26.61% |
| | | FedMAC [39] | 59.27% | 59.02% | 60.40% | 59.77% | 59.02% | 53.85% | 44.64% |
| | | PEPSY | 61.41% | 62.17% | 60.91% | 61.29% | 61.67% | 59.52% | 58.76% |

\* *All experimental results reported in Tab. 2 and Tab. 3 are conducted under the IID setting. The best and second-best results are highlighted in **bold red** and blue, respectively.*

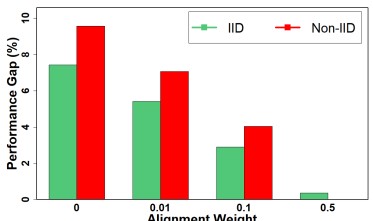

**Figure 3:** Impact of alignment loss on performance deviation.

**Table 3:** Ablation studies on different aggregation methods.

| pm\ps | Method | 0.2 | 0.4 | 0.6 | 0.8 | 1.0 |
|---|---|---|---|---|---|---|
| 0.2 | FedAvg | 63.02% | 64.19% | 65.44% | 59.01% | 56.75% |
| | FedProx | 71.24% | 69.48% | 68.85% | 59.77% | 62.55% |
| | SynFedProx | 69.86% | 61.29% | 71.63% | 68.10% | 62.29% |
| | PEPSY | 71.12% | 72.64% | 69.11% | 71.88% | 71.12% |
| 0.6 | FedAvg | 68.60% | 64.94% | 58.64% | 58.13% | 41.74% |
| | FedProx | 65.45% | 62.04% | 58.39% | 58.26% | 45.78% |
| | SynFedProx | 71.25% | 50.57% | 65.83% | 65.20% | 58.51% |
| | PEPSY | 70.87% | 69.23% | 68.47% | 68.98% | 58.76% |
| 1.0 | FedAvg | 69.86% | 67.09% | 62.54% | 61.03% | - |
| | FedProx | 69.86% | 65.32% | 57.75% | 54.47% | - |
| | SynFedProx | 66.08% | 61.92% | 64.31% | 50.57% | - |
| | PEPSY | 71.25% | 67.21% | 68.60% | 59.14% | - |

the EDF dataset, PEPSY similarly outperforms FedMAC by a significant gap and retains its lead in challenging scenarios. Across both datasets, PEPSY consistently maintains superior performance as the degree of missingness increases, highlighting its robustness to data heterogeneity and diverse missing patterns in federated contexts.

### 4.3 Performance under Different Missing Statistics between Training and Testing

We evaluated the effectiveness of our proposal by conducting more experiments with varying missing statistics between clients and servers in the IID setting. Table 2 shows that PEPSY outperforms other methods across different missing statistics. When clients have no missing data ($p_m/p_s = 0.0/0.0$), PEPSY achieves the highest accuracy in most testing missing scenarios, surpassing other baselines by an average of 3.45%. This trend continues with high client missing rates ($p_m/p_s = 1.0/0.5$), demonstrating robustness to extreme missing patterns. In the challenging inter-client missing scenario ($p_m/p_s = 0.5/1.0$), PEPSY outperforms competitors by up to 14%, highlighting PEPSY's ability to maintain consistent performance across diverse and complex client-server missing patterns.

### 4.4 Ablation Studies

**Impact of Server Aggregation Algorithms.** We conduct an ablation study on server aggregation methods to assess the effectiveness of probabilistic alignment (denoted as Syn) in our proposed framework (see Tab. 3). Denoting probabilistic synchronization as Syn, we compare FedAvg [38], FedProx [31], and their probabilistic alignment variants SynFedAvg (which is used in PEPSY) and SynFedProx. The results show that combining FedAvg and Syn significantly improves both performance and robustness in PEPSY, surpassing others and persists at higher missing rates. This is because the probabilistic synchronization mitigates inconsistent modality patterns, while skewed data distributions have less impact in this setting, then can be handled by FedAvg. These results highlight the effectiveness of server aggregation of PEPSY across diverse missing patterns.

**Impact of Alignment Loss.** Fig. 3 illustrates the effect of alignment loss on PEPSY's performance by varying the alignment weight. The model is trained in a full-modality scenario ($p_m/p_s = 0.0/0.0$) and tested on both full-modality ($p_m/p_s = 0.0/0.0$) and extreme-missing scenarios ($p_m/p_s = 0.8/1.0$). We assess the performance gap between these scenarios to evaluate the impact of alignment loss. As expected, increasing the alignment weight reduces the performance gap in both IID and Non-IID settings, demonstrating the contrastive regularizer's effectiveness in instance-aware alignment and improving model robustness. Importantly, these results support the theoretical bound outlined in 3.

**Impact of Data-Missing Profile.** To demonstrate the effectiveness of our proposed data-missing profile in handling data-missing events, we compare PEPSY with its variant, PEPSY-NP (No Profile), where the data-missing profile is excluded, across various missing statistics. As shown in Fig. 5a,

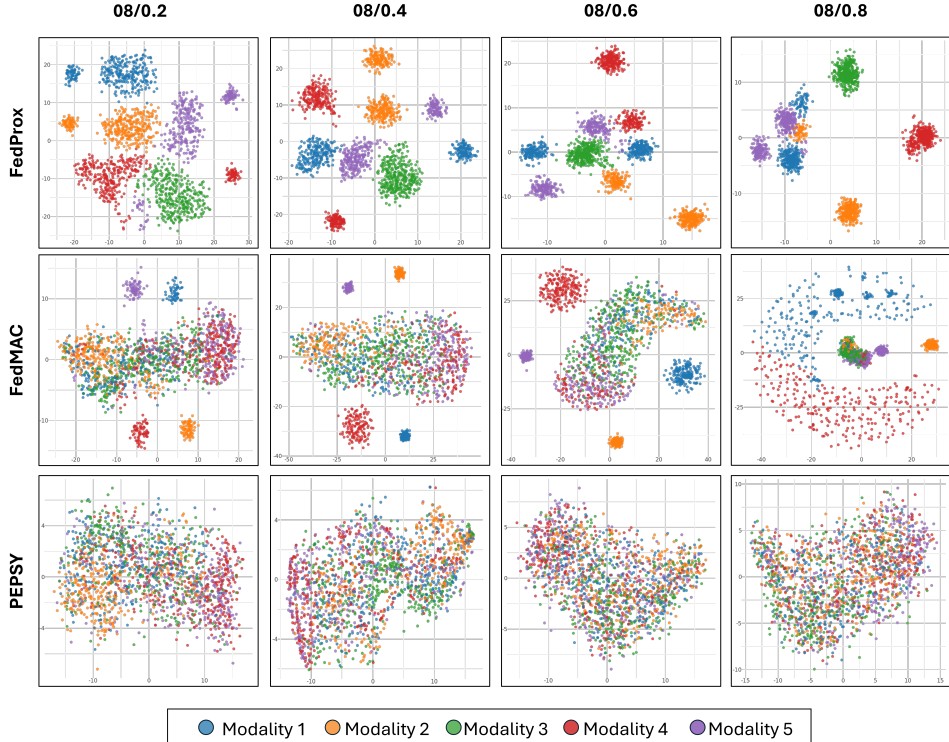

**Figure 4:** Modality representations of different methods under multiple missing scenarios. We train and provide t-SNE 2D visualizations of modality representations constructed by three methods, including our proposal, in different $p_m/p_s$ settings. All experiments are conducted on EDF dataset, nonIID setting.

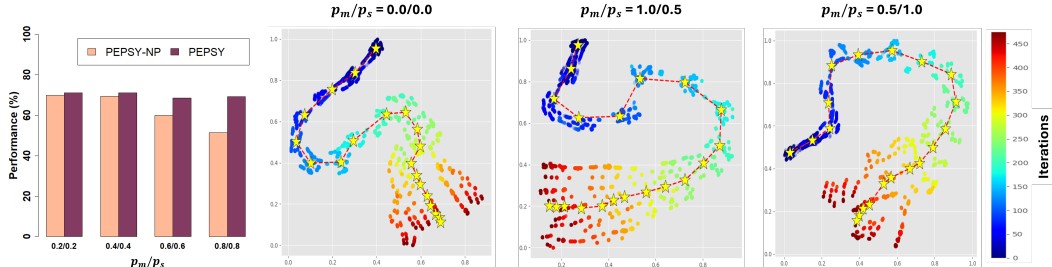

**(a)** Impact of control pool on proposal's performance under different missing scenarios.

**(b)** Visualization of global control embeddings after 500 training iterations under different missing scenarios. The reduced distance between consecutive iterations indicates convergence, while the variation shows that the embeddings capture different aspects from each client.

**Figure 5:** Ablation studies on our proposed data-missing profile.

incorporating missing profile consistently enhances PEPSY's performance in all test cases, with significant gains as the number of modalities missed increase. This is because more missing modalities causes greater variation data-missing patterns across clients, making the shared data-missing profile essential to reconfigure those variability.

**Data-Missing Profile Diversity and Convergence.** To analyze the behavior of the learned data-missing profile, we visualize the 2D t-SNE embeddings of global profile for the PTBXL dataset over 500 communication rounds under different missing settings (see Fig. 5b). The centroids of the embeddings, computed every 25 iterations, are marked by stars, with their update trajectory shown by a dashed red spline curve. As training progresses, the distance between successive centroids decreases, indicating convergence. Additionally, the spread of the embeddings gradually expands relative to their centroid, reflecting their adaptation to the diverse missing patterns across clients, suggesting that these embeddings are effectively optimized to handle varying client's local context.

**Modality Alignment Analysis.** Fig. 4 compares modality alignment among our proposed PEPSY and two baselines, FedProx and FedMAC, representing a standard FL method and the next-best performer in most experiments. Both FedProx and FedMAC fail to align modalities, reflecting their dependence on specific data-missing patterns - FedProx lacks an alignment mechanism, while FedMAC discards modality-specific cues. In contrast, PEPSY, guided by a shareable data-missing profile, reduces sensitivity to missing patterns and achieves clear modality alignment after training. More experimental results can be found in Appendix F.

## 5   Related Works

**Multimodal Learning and Missing Modalities.** Multimodal learning has gained attention for its potential to improve knowledge in centralized settings, particularly in the medical domain, where combining modalities is crucial for diagnostic accuracy [4, 23, 13, 55]. However, most methods assume full modality availability, which is often not the case in real-world scenarios with missing modalities. To address this, several approaches have been proposed: Zhang et al. [68] and Zhou et al. [69] use heuristic and statistical imputation, while neural imputation methods [9, 57, 18, 14, 12] learn imputation models before inference. Pretrained foundation models [12, 19, 1, 5, 48, 59, 33, 10] can be leveraged to transfer knowledge to imputation embeddings, and generative techniques such as VAEs, GANs, or diffusion models [17, 25, 9, 57, 2, 18, 62, 14, 30, 29] can build new imputation models. However, both approaches have clear limitations: the first requires large public datasets, often unavailable in sensitive domains like healthcare, while the second requires full-modality data at the outset. Other works [63, 7, 47, 36, 20, 44, 54] rely on available modalities to extract or reconstruct missing representations by decomposing each modality into modality- and data-specific features.

**Multimodal Federated Learning.** Driven by growing concerns over data privacy, security and transfer ineffectiveness, federated learning (FL) [38], a collaborative learning paradigm is introduced to allow multiple devices to train a shared model while keeping their local data private. This approach preserves privacy and reduces data transfer overhead [32, 51, 31, 15, 11, 40, 26, 35, 61] have, however, mostly focused on uni-modal data (e.g., image or text) while the rapid advancement of mobile phones and Internet of Things (IoT) devices [4, 23] has increasingly led to the collection of multimodal datasets. Therefore, prior works [23, 60, 42, 56] have extensively explored multimodal federated learning (MMFL), ranging from modality fusion to feature construction to enable richer and more comprehensive representations, which in turn enhances model performance and robustness. This new multimodal data paradigm has motivated a growing body of research on MMFL.

**Tackling Missing in Multimodal Federated Learning.** A key challenge in MMFL is inconsistent learning progress across clients due to heterogeneous modality combinations, arising from modality missing (inter-client missing) and input feature missing (intra-client missing) [43, 39]. Modality missing occurs when clients have different modality combinations, each dataset remaining complete [42, 45, 56], while input feature missing reflects the absence of specific modalities within an individual client's dataset, mimicking real-world scenarios [43, 39]. Initial efforts [8, 45] focused on modality missing, and recent approaches such as FedInMM [64] and FedMAC [39] have addressed input feature missing but failed when both data-missing events occur, limiting their applications. This highlights the need for solutions that effectively manage these data-missing events in multimodal federated learning, ensuring stable and robust solution under different levels of heterogeneity.

## 6   Conclusion

This paper presents a novel solution to the challenge of missing modalities in multimodal federated learning. We propose PEPSY, a method that captures each client's local data-missing view in a data-missing profile. This profile is then used to reconfigure data-missing biased representations to be faithfully data-complete. On the server side, these profiles are aggregated probabilistically into a global data-missing profile for the entire system, allowing collaboration among clients with similar data views. Theoretical analysis confirms PEPSY's stability across diverse missing modality scenarios, while empirical results demonstrate that it outperforms existing methods by up to 36.45% in addressing missing modalities in heterogeneous settings. PEPSY thus offers a flexible and stable solution for complex federated systems, with strong potential for real-world applications.

## Acknowledgement

This work is financially supported by VinUniversity under Grant No VUNI.2122.SG04. This work utilized GPU compute resource at SDSC and ACES through allocation CIS230391 from the Advanced Cyberinfrastructure Coordination Ecosystem: Services and Support (ACCESS) program [3], which is supported by U.S. National Science Foundation grants #2138259, #2138286, #2138307, #2137603, and #2138296. We would like to thank the Thomas and Margaret Huang Endowed Professor in Signal Processing and Data Science at the University of Illinois Urbana-Champaign, US and fellowship granted by VinUni-Illinois Smart Health Center, VinUniversity, Vietnam for supporting the authors' conference travel.

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

## A  Broader Statement of Impact

This research addresses the challenge of heterogeneous missing data in multimodal federated learning. Our novel design and theoretical analysis help bridge gaps between incomplete multimodal clients in fragmented systems by effectively handling diverse missing data patterns. This enables practical applications in privacy-sensitive multimodal settings with highly incomplete data. While the potential real-world use of our methods could raise ethical concerns, these are indirect and unpredictable consequences beyond the scope of this work. Our experiments rely solely on publicly available datasets, and no ethical issues arise from our evaluation process.

## B  Missing Modality Simulation

This section details how we simulate missing modality in a comprehensive and controllable way. Following [39], we define two types of ratio in missing modality, denoted as $p_s$ and $p_m$. First, $p_s$, namely sample ratio, is the ratio of samples with missing modalities over a given dataset. Second, $p_m$ is modality ratio, and used as the ratio of missing modalities within those samples. For simplicity, a pair of $(p_m, p_s)$ can be called *missing statistics*, since it reflects statisitcs of modality missing in both detailed and overall views (see Fig. 6. The *missing degree* is then defined as $p_m \times p_s$, representing the overall proportion of instances with missing modalities. These missing statistic can remodel the an arbitrary dataset $\mathcal{D}$ via a missing matrix:

$$\phi(\mathcal{D}) = \begin{bmatrix} b_1^1 & \dots & b_1^{|\mathcal{M}|} \\ \vdots & \ddots & \vdots \\ b_{|\mathcal{D}|}^1 & \dots & b_{|\mathcal{D}|}^{|\mathcal{M}|} \end{bmatrix}, \tag{10}$$

where $b_{dm} \in \{0, 1\}$ indicates whether modality $m$ is missing (0) or available (1) for the $d$ - th sample. Here, $|\mathcal{M}|$ is the cardinality of $\mathcal{M}$, and $|\mathcal{D}|$ is the number of samples. The incomplete dataset $\hat{D}$ can be obtained by multiplying $\mathcal{D}$ by the missing matrix:

$$\hat{x}_i = [x_{d1}, \dots, x_{d|\mathcal{M}|}] \odot [b_{d1}, \dots, b_{d|\mathcal{M}|}], \tag{11}$$

where $\odot$ represents element-wise multiplication. Examples of incomplete datasets are shown in Fig. 6. In this work, we apply the same $(p_m/p_s)$ pairs for all clients in our experiments.

## C  Implementation Details

**Dataset Preparation.**  All baselines use data from the PTBXL and EDF datasets. The PTBXL dataset contains 3,963 clinical samples across five classes. Each sample includes 12 modalities, corresponding to electrocardiogram (ECG) recordings, and is labeled with a single class. Details can be found in [45]. The EDF dataset consists of 197 full-night polysomnographic (PSG) recordings with five key modalities (excluding rectal temperature and biomarkers). Each recording is segmented into multiple sleep stages, including Wake and stages S1–S4. For this work, we relabel S1 and S2 as N1 and N2, and merge S3 and S4 into N3, resulting in a 5-class classification problem [24]. We segment all sleep recordings into individual signals, each representing a sleep pattern, creating a unified dataset of 8,755 signals. This unified dataset is used for all experiments. Both datasets are divided into training and testing sets with ratio 80/20. The testing are used for evaluation on the server side, while the training sets are split to all clients following IID or NonIID settings. For NonIID setting, we use Dirichlet distribution with $\alpha = 0.5$ to distribute training data points. All modalities in this work are signal-based modality.

**Hyperparmeter Settings.**  All methods in this work use an Inception Network as the modality encoder, following [45]. Experiments are run on an A6000 GPU with 48GB of memory. For classification, we use Cross Entropy Loss for $\mathcal{L}_{task}$. The embedding dimension is set to $C = 128$. There are $K = 32$ clients in total, with 10 clients randomly selected to participate in each training round. Each selected client trains the model for $E = 3$ epochs per round. Optimization is done using Stochastic Gradient Descent (SGD) [49]. Communication with the server occurs over $T = 1000$ rounds. Both the alignment contrastive weight ($\lambda$) and the relevance regularization weight ($\eta$) are set to 0.1 for all experiments. However, $\lambda$ is increased to 0.2 when $p_m \in \{0.8, 1.0\}$, corresponding

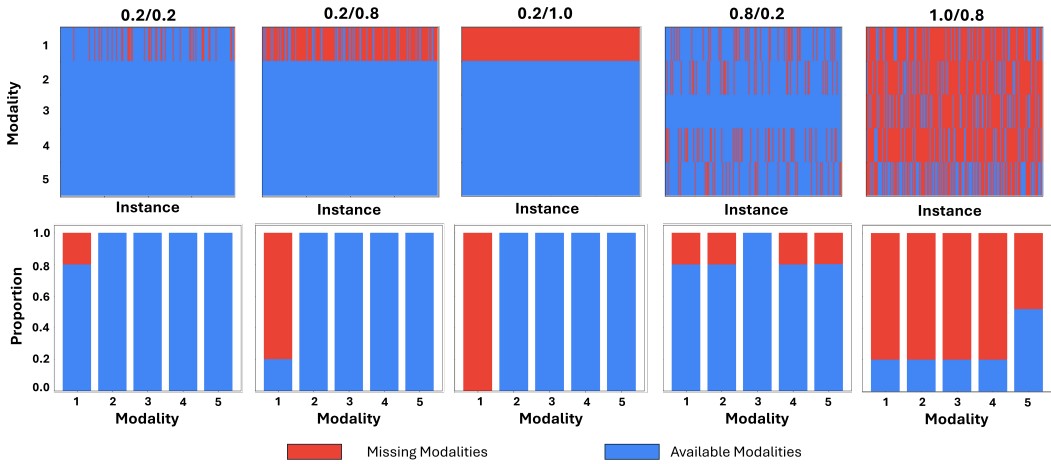

**Figure 6:** Examples of incomplete datasets $\hat{\mathcal{D}}$ with varying missing statistics ($p_m/p_s$). By controlling these missing statistics, we create diverse evaluation scenarios that reflect real-world conditions.

**Table 4:** Hyperparameter setting for all baselines and our PEPSY

| Dataset | Method | $p\_m$ | Batch Size | Communication Round ($T$) | Eps. in Local Training ($E$) | Contrastive Weight ($\lambda$) | Optimizer & Learning Rate | Total Clients ($K$) | Sampled Clients |
|---|---|---|---|---|---|---|---|---|---|
| PTBXL | FedProx MIFL FedInMM FedMSplit FedMAC PEPSY | 0.2 | 32 | 1000 | 3 | 0.1 | SGD lr: 0.01 | 32 | 10 |
| | | 0.4 | 32 | 1000 | 3 | 0.1 | SGD lr: 0.01 | 32 | 10 |
| | | 0.6 | 32 | 1000 | 3 | 0.1 | SGD lr: 0.01 | 32 | 10 |
| | | 0.8 | 32 | 1000 | 3 | 0.2 | SGD lr: 0.01 | 32 | 10 |
| | | 1.0 | 32 | 1000 | 3 | 0.2 | SGD lr: 0.01 | 32 | 10 |
| EDF | FedProx MIFL FedInMM FedMSplit FedMAC PEPSY | 0.2 | 128 | 500 | 3 | 0.1 | SGD lr: 0.1 | 32 | 10 |
| | | 0.4 | 128 | 500 | 3 | 0.1 | SGD lr: 0.1 | 32 | 10 |
| | | 0.6 | 128 | 500 | 3 | 0.1 | SGD lr: 0.1 | 32 | 10 |
| | | 0.8 | 128 | 500 | 3 | 0.2 | SGD lr: 0.1 | 32 | 10 |
| | | 1.0 | 128 | 500 | 3 | 0.2 | SGD lr: 0.1 | 32 | 10 |

to extreme missing modality scenarios that require stronger alignment. Detailed hyperparameter settings are listed in Tab. 4. Unless otherwise specified, we use the original configurations from the referenced papers.

# D  Theorem Proof

## D.1  Theorem Setup

This section provides the initial setup for our proof for Theorem 3.1. From now on, we remove the subscript indicating instance index in our notation for simplicity. Following notations in Section 1, our proposed method described in Section 2 can be expressed as composition of two internal functions: $\hat{\boldsymbol{y}} = f_p\big(\{\boldsymbol{w}_i\}_{i=1}^{|\mathcal{M}|}\big) = f_p\big(\{f_e(\boldsymbol{x}_i)\}_{i=1}^{|\mathcal{M}|}\big)$. Here, $f_p(\cdot)$ and $f_e(\cdot)$ are post-process head and feature extractor, respectively. In specific, $f_e(\cdot)$ takes each modality $\boldsymbol{x}_i$ as input and generates a modality representation $\boldsymbol{w}_i$ (as shown in Section 2) by concatenating three types of information including modality-specific ($\boldsymbol{w}_i^{\mathrm{mod}}$), data-specific ($\boldsymbol{w}_i^{\mathrm{ins}}$) and missing-pattern ($\boldsymbol{w}_i^{\mathrm{mis}}$) features, i.e., $\boldsymbol{w}_i = [\boldsymbol{w}_i^{\mathrm{mod}} \circ \boldsymbol{w}_i^{\mathrm{ins}} \circ \boldsymbol{w}_i^{\mathrm{con}}]$. In addition, to make the proof easy to follow, we denote $\boldsymbol{h}_i$ and $\boldsymbol{u}$ as extraction for present modalities and imputation for missing modalities, respectively, as described in Section 2.2.1, leading to follow-up notation of modality representations $\boldsymbol{w}_i$ and $\boldsymbol{w}_i(\boldsymbol{u})$. To clarify, if the notation $\boldsymbol{h}_i$ is used for missing modality, i.e., $i \in \mathcal{S}$, it means that $\boldsymbol{h}_i$ here is the "true" feature if that modality presents. We use this notation in our proof from now on.

**Assumption D.1** *The post-processing head $f_p$ is Lipschitz continuous with respect to the input vector $\boldsymbol{x}$, i.e., there exists a constant $L > 0$ such that for all $\boldsymbol{x}, \boldsymbol{y} \in \mathbb{R}^n$, the following condition holds:*

$$\|f_p(\boldsymbol{x_i}) - f_p(\boldsymbol{x_j})\| \leq L\|\boldsymbol{x_i} - \boldsymbol{x_j}\|,$$

*where $f_p : \mathbb{R}^n \to \mathbb{R}^m$ is the post-processing head, $\|\cdot\|$ denotes the chosen norm (here the $\ell_2$-norm), and $L$ is a Lipschitz constant.*

**Assumption D.2** *During test time, all parameters of the proposed framework are bounded. Specifically, for any weight matrix $A$, we have:*

$$\epsilon_A^- \leq \|A\| \leq \epsilon_A^+,$$

*where $\|\cdot\|$ denotes the $\ell_2$-norm and $\epsilon_A^-$ and $\epsilon_A^+$ are positive constants that bound the spectral norm of $A$. This assumption similarly applies to the output representations that are transformed by the learned weight matrices.*

In Assumption D.1, we assume that the neural network used as the post-processing head in our proposed design is Lipschitz continuous. This assumption is widely accepted in the machine learning community due to its relevance in ensuring stable and smooth behavior of the model.

Assumption D.2 states that the learned parameters of the network are bounded during test time. This assumption is reasonable and holds true in most real-world scenarios, where the model parameters are deterministic and constrained within known ranges during inference. Such bounds are typically enforced either through explicit regularization during training or through implicit constraints imposed by the training process itself (e.g., gradient clipping or weight normalization). Therefore, this assumption is not only theoretically sound but also consistent with common practices in machine learning.

**Remark D.3** *(Bounded Extracted Representations) In our data-specific representation extraction, each output feature $\boldsymbol{h}_i$ of modality $i$ is normalized to zero mean and unit variance (via Batch Normalization layer), followed by a learned scaling ($\gamma$) and shift ($\beta$) parameters. When Assumption D.2 holds, we have $\epsilon_\gamma^- \leq \|\gamma\| \leq \epsilon_\gamma^+$ and $\epsilon_\beta^- \leq \|\beta\| \leq \epsilon_\beta^+$ and derive:*

$$\|\boldsymbol{h}_i\| = \|\gamma\bar{\boldsymbol{h}}_i + \beta\| \leq \|\gamma\| \cdot \|\bar{\boldsymbol{h}}_i\| + \|\beta\|. \tag{12}$$

*where $\bar{\boldsymbol{h}}_i$ is batch-normalized $h_i$. Since the normalized term has unit variance, its norm is bounded by $\sqrt{C}$, where $C$ is the feature dimension. Hence,*

$$\max(\epsilon_\gamma^- \sqrt{C} - \epsilon_\beta^+, \, 0) \leq \|\boldsymbol{h}_i\| \leq \epsilon_\gamma^+ \sqrt{C} + \epsilon_\beta^+, \tag{13}$$

*Let $\epsilon_{\gamma\beta}^- \triangleq \max(\epsilon_\gamma^- \sqrt{C} - \epsilon_\beta^+, \, 0)$ and $\epsilon_{\gamma\beta}^+ \triangleq \epsilon_\gamma^+ \sqrt{C} + \epsilon_\beta^+$, Eq. 13 shows that $\|h_i\|$ is bounded within a deterministic range. Consequently, the imputation feature derived by taking average of available modalities is bounded for the same reason.*

### D.2 Theoretical Analysis in Simple Case

In this section, we first investigate the behavior of PEPSY in a simple case of missing modality before further generalization. Let consider the deviations of our proposal when feeding full-modality input and one missing the first $|\mathcal{S}|$ out of $\mathcal{M}$ modalities, i.e., $\mathcal{S}_f = \{1, \ldots, |\mathcal{S}|\}$ as follows:

$$\|\boldsymbol{y}^{\mathcal{S}} - \boldsymbol{y}^{\emptyset}\| \tag{14}$$

$$= \left\| f_p\Big(\{\boldsymbol{w}_i\}_{i=|\mathcal{S}|+1}^{|\mathcal{M}|}, \{\boldsymbol{w}_j(\boldsymbol{u})\}_{j=1}^{|\mathcal{S}|}\Big) - f_p\Big(\{\boldsymbol{w}_i\}_{i=1}^{|\mathcal{M}|}\Big) \right\| \tag{15}$$

$$= \left\| \frac{1}{|\mathcal{S}|}\Big( \|\boldsymbol{w}_1(\boldsymbol{u}) - \boldsymbol{w}_1\| \nabla_{\boldsymbol{w}_1(\boldsymbol{u})} f_p(\boldsymbol{w}_1) + \cdots + \|\boldsymbol{w}_{|\mathcal{M}|}(\boldsymbol{u}) - \boldsymbol{w}_{|\mathcal{M}|}\| \nabla_{\boldsymbol{w}_{|\mathcal{M}|}(\boldsymbol{u})} f_p(\boldsymbol{w}_{|\mathcal{M}|}) \Big) \right\| \tag{16}$$

$$= \frac{1}{|\mathcal{S}|} \sum_{i=1}^{|\mathcal{S}|} \|\boldsymbol{w}_i(\boldsymbol{u}) - \boldsymbol{w}_i\| \nabla_{\boldsymbol{w}_i(\boldsymbol{u})} f_p(\boldsymbol{w}_i) \tag{17}$$

Here, we use first-order Taylor approximation $|\mathcal{S}|$ times to transform Eq, 15 to Eq. 16. Since $f_p(\cdot)$ is $L$-Lipschitz (see Assumption D.1), Eq 17 can be transformed as:

$$\|\boldsymbol{y}^{\mathcal{S}_f} - \boldsymbol{y}^{\emptyset}\| \tag{18}$$

$$\leq L \sum_{i=1}^{|\mathcal{M}|} \|\boldsymbol{w}_i(\boldsymbol{u}) - \boldsymbol{w}_i\| \tag{19}$$

$$\leq L \sum_{i=1}^{|\mathcal{M}|} \|\left[\boldsymbol{w}_i^{\mathrm{mod}} \circ \boldsymbol{u} \circ \boldsymbol{w}_i^{\mathrm{con}}\right] - \left[\boldsymbol{w}_i^{\mathrm{mod}} \circ \boldsymbol{h}_i \circ \boldsymbol{w}_i^{\mathrm{con}}\right]\| \tag{20}$$

$$= L \sum_{i=1}^{|\mathcal{M}|} \left\| 0 \circ (\boldsymbol{u} - \boldsymbol{h}_i) \circ \operatorname*{argmax}_{\psi_p}\left(e\big(\mathbf{q}(\boldsymbol{w}_i^{\mathrm{mod}} \circ u), \psi_p\big)\right) - \operatorname*{argmax}_{\psi_p}\left(e\big(\mathbf{q}(\boldsymbol{w}_i^{\mathrm{mod}} \circ h_i), \psi_p\big)\right)\right\| \tag{21}$$

where $\boldsymbol{w}_i(\boldsymbol{u})$ is the imputed representation for modality $i$, obtained using the imputation data-specific feature $\boldsymbol{u}^{\mathrm{ins}}$, and $\boldsymbol{w}_i$ is the original modality feature. Here, we represent the query-key matching function $\operatorname*{argmax}_{\psi_p} e(\mathbf{q}(\boldsymbol{w}_i^{\mathrm{mod}} \circ \boldsymbol{w}_i^{\mathrm{ins}_i}), \psi_p)$ as an approximate attention selecting the $\psi_p$ with the highest weight, by using softmax function $\sigma(\cdot, \cdot) \triangleq \mathrm{softmax}(e(\mathbf{q}(\cdot), \cdot)$. For simplicity, we use $\tilde{\sigma}$ as a Lipschitz constant of this approximated similarity function. Considering individual modality component $\|w_i(u) - w_i\|$, these lead to the following derivations:

$$\|\boldsymbol{w}_i(\boldsymbol{u}) - \boldsymbol{w}_i\| \tag{22}$$

$$\approx \left\| 0 \circ (\boldsymbol{u} - \boldsymbol{h}_i) \circ \left\{ \sum_{p=1}^{\tau} \left[\sigma\big(\boldsymbol{w}_i^{\mathrm{mod}} \circ \boldsymbol{u}, \psi_p\big) - \sigma\big(\boldsymbol{w}_i^{\mathrm{mod}} \circ \boldsymbol{h}_i, \psi_p\big)\right] \odot \psi_p \right\} \right\| \tag{23}$$

$$\leq \|\boldsymbol{u} - \boldsymbol{h}_i\| + \sum_{p=1}^{\tau} \left\|\sigma\big(\boldsymbol{w}_i^{\mathrm{mod}} \circ \boldsymbol{u}, \psi_p\big) - \sigma\big(\boldsymbol{w}_i^{\mathrm{mod}} \circ \boldsymbol{h}_i, \psi_p\big)\right\| \odot \|\psi_p\|. \tag{24}$$

$$\leq \|\boldsymbol{u} - \boldsymbol{h}_i\| + \tilde{\sigma} \sum_{p=1}^{\tau} \|\boldsymbol{u} - \boldsymbol{h}_i\| \times \|\psi_p\| \tag{25}$$

$$\leq \big(1 + \tilde{\sigma}\tau \max_{\psi_p}(\epsilon_{\psi_p}^+)\big)\|\boldsymbol{u} - \boldsymbol{h}_i\| \tag{26}$$

$$\leq \mu\sqrt{\|\boldsymbol{u}\|^2 + \|\boldsymbol{h}_i\|^2 - 2\boldsymbol{u}\boldsymbol{h}_i^{\top}}. \tag{27}$$

where $\mu = 1 + \tilde{\sigma}\tau \max_{\psi_p}(\epsilon_{\psi_p}^+)$ and $\epsilon_{\psi_p}^+$ denotes upperbound of embedding controls, which is fixed in test time. Taking the summation over all $i$, we obtain:

$$\sum_{i=1}^{|\mathcal{S}|} \|\boldsymbol{w}_i(\boldsymbol{u}) - \boldsymbol{w}_i\| \tag{28}$$

$$\leq \mu \sum_{i=1}^{|\mathcal{S}|} \sqrt{\|\boldsymbol{u}\|^2 + \|\boldsymbol{h}_i\|^2 - 2\boldsymbol{u}\boldsymbol{h}_i^{\top}} \tag{29}$$

$$\leq \mu \sum_{i=1}^{|\mathcal{S}|} \left(\left\|\frac{1}{|\mathcal{M}| - |\mathcal{S}|} \sum_{j=|\mathcal{S}|+1}^{|\mathcal{M}|} h_j\right\|^2 + \|\boldsymbol{h}_i\|^2 - \frac{2}{|\mathcal{M}| - |\mathcal{S}|} \sum_{j=|\mathcal{S}|+1}^{|\mathcal{M}|} \boldsymbol{h}_j\boldsymbol{h}_i^{\top}\right)^{\frac{1}{2}}. \tag{30}$$

Here, $\boldsymbol{u}$ represents the imputed data-specific representation, computed as the mean of corresponding features from the available modalities (see Section 2.2.1). This justifies the transformation from Eq. 29 to Eq. 30. Based on Remark D.3, we have:

$$\sum_{i=1}^{|\mathcal{S}|} \|\boldsymbol{w}_i(\boldsymbol{u}) - \boldsymbol{w}_i\| \tag{31}$$

$$\leq \mu \sum_{i=1}^{|\mathcal{S}|} \left( \frac{2}{|\mathcal{M}| - |\mathcal{S}|} (|\mathcal{M}| - |\mathcal{S}|)\epsilon_{\gamma\beta}^{+2} + \epsilon_{\gamma\beta}^{+2} - \frac{2}{|\mathcal{M}| - |\mathcal{S}|} \sum_{j=|\mathcal{S}|+1}^{|\mathcal{M}|} \boldsymbol{h}_j \boldsymbol{h}_i^\top \right)^{\frac{1}{2}}. \tag{32}$$

$$\leq \mu \sum_{i=1}^{|\mathcal{S}|} \left( 3\epsilon_{\gamma\beta}^{+2} - \frac{2}{|\mathcal{M}| - |\mathcal{S}|} \sum_{j=|\mathcal{S}|+1}^{|\mathcal{M}|} \boldsymbol{h}_j \boldsymbol{h}_i^\top \right)^{\frac{1}{2}} \tag{33}$$

$$\leq \mu \sum_{i=1}^{|\mathcal{S}|} \left( 3\epsilon_{\gamma\beta}^{+2} - \frac{2}{|\mathcal{M}| - |\mathcal{S}|} \sum_{j=|\mathcal{S}|+1}^{|\mathcal{M}|} \boldsymbol{h}_j \boldsymbol{h}_i^\top \right)^{\frac{1}{2}} \tag{34}$$

$$\leq \mu \sum_{i=1}^{\mathcal{S}} \left( \frac{1}{|\mathcal{M}| - |\mathcal{S}|} \sum_{j=|f|+1}^{|\mathcal{M}|} 3\epsilon_{\gamma\beta}^{+2} - \frac{1}{|\mathcal{M}| - |\mathcal{S}|} \sum_{j=|\mathcal{S}|+1}^{\mathcal{M}} 2\boldsymbol{h}_j \boldsymbol{h}_i^\top \right)^{\frac{1}{2}} \tag{35}$$

$$\leq \sqrt{\frac{\mu^2}{|\mathcal{M}| - |\mathcal{S}|}} \sum_{i=1}^{\mathcal{S}} \left( \sum_{j=|\mathcal{S}|+1}^{\mathcal{M}} 3\epsilon_{\gamma\beta}^{+2} - 2\boldsymbol{h}_j \boldsymbol{h}_i^\top \right)^{\frac{1}{2}} \tag{36}$$

Here, the bound on $\|\boldsymbol{w}_i(\boldsymbol{u}) - \boldsymbol{w}_i\|$ highlights how the interaction terms between $\boldsymbol{h}_j$ and $\boldsymbol{h}_i$ contribute to the overall norm. Furthermore, the right-handed side of 36 is non-negative showing the validity of this transformation. If we further substitute Eq. 36 in Eq. 19, we obtain an intermediate inequality:

$$\|\boldsymbol{y}^{\mathcal{S}_f} - \boldsymbol{y}^\emptyset\| \leq \sqrt{\frac{\mu^2}{|\mathcal{M}| - |\mathcal{S}|}} \sum_{i=1}^{|\mathcal{S}|} \left( \sum_{j=|\mathcal{S}|+1}^{|\mathcal{M}|} 3\epsilon_{\gamma\beta}^2 - 2\boldsymbol{h}_j \boldsymbol{h}_i^\top \right)^{\frac{1}{2}} \tag{37}$$

where we restate $\mu^2 \leftarrow \mu^2 L$ without loss of generalization since both $\mu$ and $L$ are constant.

### D.3 Theoretical Analysis Generalization

In this section, we extend the bound in Eq. 37, originally derived assuming the first $|\mathcal{S}|$ modalities out of $\mathcal{M}$ are missing. The current bound assumes the missing modalities are the first $|\mathcal{S}|$ in order. We generalize this to the case where any subset $\mathcal{S} \subset \mathcal{M}$ of size $|\mathcal{S}|$ is missing. To do this, we generalize bound in Eq. 37 over missing modality set $\mathcal{S}$, and over all instances of an arbitrary dataset $\mathcal{D}$.

#### D.3.1 Generalize over Missing Modality Set.

Given $\mathcal{M}$ is the set of all modalities, with cardinality $|\mathcal{M}|$, we define $\mathcal{S} \subseteq \mathcal{M}$ as a subset representing the missing modalities, with cardinality of $|\mathcal{S}|$. For each missing modality $i \in \mathcal{S}$, we define a random variable $Z_S^i$ as follows:

$$\boldsymbol{z}_{\mathcal{S}}^i = \sum_{j \notin \mathcal{S}} (3\epsilon_{\gamma\beta}^2 - 2\boldsymbol{h}_j \boldsymbol{h}_i^\top), \tag{38}$$

The expected value of $\sqrt{Z_S^i}$, averaged over all possible missing subsets $\mathcal{S}$, is then computed as the following equation:

$$\mathbb{E}\left[ \sqrt{Z_S^i} \right] = \frac{1}{|\mathcal{S}|\binom{|\mathcal{M}|}{|\mathcal{S}|}} \sum_{\mathcal{S} \subseteq \mathcal{M}} \sum_{i \in \mathcal{S}} \sqrt{\boldsymbol{z}_{\mathcal{S}}^i} \tag{39}$$

$$\leq \sqrt{\frac{1}{|\mathcal{S}|\binom{|\mathcal{M}|}{|\mathcal{N}|}} \sum_{\mathcal{S} \subseteq \mathcal{M}} \sum_{i \in \mathcal{S}} \sum_{j \notin \mathcal{S}} (3\epsilon_{\gamma\beta}^2 - 2\boldsymbol{h}_j \boldsymbol{h}_i^\top)} \tag{40}$$

where $\binom{|\mathcal{M}|}{|\mathcal{S}|}$ denotes the number of ways to choose $|\mathcal{S}|$ elements from $\mathcal{M}$. To derive Eq.40 from Eq. 39, we apply the Jensen's inequality due to the concavity of square root function.

**Observation.** The term $\sum_{\mathcal{S} \subseteq \mathcal{M}} \sum_{i \in \mathcal{S}} \sum_{j \notin \mathcal{S}} (3\epsilon_{\gamma\beta}^2 - 2\boldsymbol{h}_j \boldsymbol{h}_i^\top)$ means that we are summing over all subsets $\mathcal{S} \subseteq \mathcal{M}$ of fixed size $|\mathcal{S}|$. For each subset, we sum over all ordered pairs $(i, j)$ where $i \in \mathcal{S}$ and $j \notin \mathcal{S}$. For a fixed pair $(i, j)$ with $i \neq j$, the number of subsets $\mathcal{S}$ that include $i$ and exclude $j$ depends only on $i$ and $j$. In other words, once $i$ and $j$ are fixed, the remaining $|\mathcal{S}| - 1$ elements of $\mathcal{S}$

must be chosen from the remaining $|\mathcal{M}| - 2$ elements (excluding $i$ and $j$), giving exactly $\binom{|\mathcal{M}|-2}{|\mathcal{S}|-1}$ subsets. Therefore, each term $(3\epsilon_{\gamma\beta}^2 - 2\boldsymbol{h}_j\boldsymbol{h}_i^\top)$ appears precisely $\binom{|\mathcal{M}|-2}{|\mathcal{S}|-1}$ times in the sum. This lets us rewrite the original triple sum as a double sum over all ordered pairs $(i, j)$ with $i \neq j$, multiplied by the constant $\binom{M-2}{N-1}$, simplifying into:

$$\sum_{\mathcal{S} \subseteq \mathcal{M}} \sum_{i \in \mathcal{S}} \sum_{j \notin \mathcal{S}} (3\epsilon_{\gamma\beta}^{+2} - 2\boldsymbol{h}_j\boldsymbol{h}_i^\top) = \sum_{i=1}^{|\mathcal{M}|} \sum_{\substack{j=1 \\ j \neq i}}^{|\mathcal{M}|} \binom{|\mathcal{M}|-2}{|\mathcal{S}|-1} (3\epsilon^2 - 2\boldsymbol{h}_i\boldsymbol{h}_j^\top). \tag{41}$$

Substituting Eq. 41 into Eq. 40, we obtain:

$$\mathbb{E}_{i,\mathcal{S}}\left[\sqrt{\boldsymbol{Z}_S^i}\right] \tag{42}$$

$$\leq \sqrt{\frac{1}{|\mathcal{S}|\binom{|\mathcal{M}|}{|\mathcal{S}|}} \sum_{i=1}^{|\mathcal{M}|} \sum_{\substack{j=1 \\ j \neq i}}^{|\mathcal{M}|} \binom{|\mathcal{M}|-2}{|\mathcal{S}|-1}(3\epsilon_{\gamma\beta}^{+2} - 2\boldsymbol{h}_i\boldsymbol{h}_j^\top)} \tag{43}$$

$$= \sqrt{\frac{|\mathcal{S}|!(|\mathcal{M}|-|\mathcal{S}|)!}{|\mathcal{M}|!|\mathcal{S}|} \times \frac{(|\mathcal{M}|-2)!}{(|\mathcal{S}|-1)!(|\mathcal{M}|-|\mathcal{S}|-1)!} \sum_{i=1}^{|\mathcal{M}|} \sum_{\substack{j=1 \\ j \neq i}}^{|\mathcal{M}|}(3\epsilon_{\gamma\beta}^2 - 2\boldsymbol{h}_j\boldsymbol{h}_j^\top)} \tag{44}$$

$$= \sqrt{\frac{|\mathcal{M}|-|\mathcal{S}|}{|\mathcal{M}|(M-1)} \sum_{i=1}^{|\mathcal{M}|} \sum_{\substack{j=1 \\ j \neq i}}^{|\mathcal{M}|}(3\epsilon_{\gamma\beta}^2 - 2\boldsymbol{h}_j\boldsymbol{h}_j^\top)} \tag{45}$$

$$= \sqrt{\frac{|\mathcal{M}|-|\mathcal{S}|}{|\mathcal{M}|(|\mathcal{M}|-1)}} \times \sqrt{\sum_{i=1}^{|\mathcal{M}|} \sum_{\substack{j=1 \\ j \neq i}}^{|\mathcal{M}|}(3\epsilon_{\gamma\beta}^{+2} - 2\boldsymbol{h}_j\boldsymbol{h}_j^\top)}. \tag{46}$$

We now bound the expectation of Eq. 37 over all possible missing modality patterns ($\mathcal{S}$) as follows:

$$\mathbb{E}_{\mathcal{S}}\left[\|\boldsymbol{y}^{\mathcal{S}} - \boldsymbol{y}^{\emptyset}\|\right] = \frac{1}{\binom{|\mathcal{M}|}{|\mathcal{S}|}} \sum_{\mathcal{S} \subseteq \mathcal{M}} \left\|\boldsymbol{y}^{\mathcal{S}} - \boldsymbol{y}^{\emptyset}\right\| \tag{47}$$

$$\leq \sqrt{\frac{\mu^2|\mathcal{S}|^2}{|\mathcal{M}|-|\mathcal{S}|}} \frac{1}{|\mathcal{S}|\binom{|\mathcal{M}|}{|\mathcal{S}|}} \sum_{\mathcal{S} \subseteq \mathcal{M}} \left[\sum_{i \in \mathcal{S}}\left(\sum_{j \notin \mathcal{S}}(3\epsilon_{\gamma\beta}^{+2} - \boldsymbol{h}_i\boldsymbol{h}_j^\top)\right)^{\frac{1}{2}}\right] \tag{48}$$

$$\leq \sqrt{\frac{\mu^2|\mathcal{S}|^2}{|\mathcal{M}|-|\mathcal{S}|}} \mathbb{E}_{i,\mathcal{S}}\left[\sqrt{\boldsymbol{Z}_{\mathcal{S}}^i}\right] \tag{49}$$

$$\leq \sqrt{\frac{\mu^2|\mathcal{S}|^2}{|\mathcal{M}|-|\mathcal{S}|}} \sqrt{\frac{|\mathcal{M}|-|\mathcal{S}|}{|\mathcal{M}|(|\mathcal{M}|-1)}} \sqrt{\sum_{i=1}^{|\mathcal{M}|} \sum_{\substack{j=1 \\ j \neq i}}^{|\mathcal{M}|}(3\epsilon_{\gamma\beta}^{+2} - 2\boldsymbol{h}_i\boldsymbol{h}_j^\top)} \tag{50}$$

$$\leq \sqrt{\frac{\mu^2|\mathcal{S}|^2}{|\mathcal{M}|(|\mathcal{M}|-1)}} \sqrt{\sum_{i=1}^{|\mathcal{M}|} \sum_{\substack{j=1 \\ j \neq i}}^{|\mathcal{M}|}(3\epsilon_{\gamma\beta}^{+2} - 2\boldsymbol{h}_i\boldsymbol{h}_j^\top)}. \tag{51}$$

In summary, in this section, we derive an upper bound for the expected outcome deviation in missing- and full-modality scenarios over the missing scenarios ($\mathcal{S}$) as:

$$\mathbb{E}_{\mathcal{S}}\left[\|\boldsymbol{y}^{\mathcal{S}} - \boldsymbol{y}^{\emptyset}\|\right] \leq \sqrt{\frac{\mu^2|\mathcal{S}|^2}{|\mathcal{M}|(|\mathcal{M}|-1)}} \sqrt{\sum_{i=1}^{|\mathcal{M}|} \sum_{\substack{j=1 \\ j \neq i}}^{|\mathcal{M}|}(3\epsilon_{\gamma\beta}^{+2} - 2\boldsymbol{h}_i\boldsymbol{h}_j^\top)} \tag{52}$$

### D.3.2 Generalize over Instances

This section describes how we generalize the bound in Eq. 52 to batch- or dataset-level. Furthermore, we reveal the connection between our theoretical bound and the training loss function that we propose, indicating the effectiveness of training loss in our proposal. To address this, we start by considering the mean difference over a dataset $\mathcal{D}$ with cardinality $|\mathcal{D}|$:

$$\frac{1}{|\mathcal{D}|}\sum_{d=1}^{|\mathcal{D}|}\mathbb{E}_S\left[\|\boldsymbol{y}_{\boldsymbol{x}_d}^{\mathcal{S}} - \boldsymbol{y}_{\boldsymbol{x}_d}^{\emptyset}\|\right] \leq \sqrt{\frac{\mu^2|\mathcal{S}|^2}{|\mathcal{M}|(|\mathcal{M}|-1)}}\frac{1}{|\mathcal{D}|}\sum_{d}^{|\mathcal{D}|}\sqrt{\sum_{i=1}^{|\mathcal{M}|}\sum_{\substack{j=1\\j\neq i}}^{|\mathcal{M}|}(3\epsilon_{\gamma\beta}^{+2} - 2\boldsymbol{h}_{di}\boldsymbol{h}_{dj}^{\top})} \quad (53)$$

$$\leq \frac{\sqrt{|\mathcal{D}|}\mu|\mathcal{S}|}{|\mathcal{D}|\sqrt{|\mathcal{M}|(|\mathcal{M}|-1)}}\sqrt{\sum_{d=1}^{|\mathcal{D}|}\sum_{i=1}^{|\mathcal{M}|}\sum_{j\neq i}^{|\mathcal{M}|}(3\epsilon_{\gamma\beta}^2 - 2\boldsymbol{h}_{di}\boldsymbol{h}_{dj}^{\top})} \quad (54)$$

in which Eq. 53 is transformed to Eq. 54 by using triangle inequality. To avoid confusion, we analyze the right-hand term separately, as it plays a central role in the transformation process. Let $\tilde{h}_{di}$ denote the $\ell_2$-normalized feature, i.e., $\tilde{h}_{di} = h_{di}/\|h_{di}\|$.

$$\sum_{d=1}^{|\mathcal{D}|}\sum_{i=1}^{|\mathcal{M}|}\sum_{\substack{j=1\\j\neq i}}^{|\mathcal{M}|}(3\epsilon_{\gamma\beta}^{+2} - 2\boldsymbol{h}_{di}\boldsymbol{h}_{dj}^{\top}) \quad (55)$$

$$\leq \epsilon_{\gamma\beta}^{-2}\sum_{d}^{|\mathcal{D}|}\sum_{i=1}^{|\mathcal{M}|}\sum_{\substack{j=1\\j\neq i}}^{|\mathcal{M}|}(3\frac{\epsilon_{\gamma\beta}^{+2}}{\epsilon_{\gamma\beta}^{-2}} - 2\tilde{\boldsymbol{h}}_{di}\tilde{\boldsymbol{h}}_{dj}^{\top}) \quad (56)$$

$$\leq 3|\mathcal{D}||\mathcal{M}|(|\mathcal{M}|-1)\epsilon_{\gamma\beta}^{+2}$$
$$+ 2\epsilon_{\gamma\beta}^{-2}\sum_{d=1}^{|\mathcal{D}|}\sum_{i=1}^{|\mathcal{M}|}\sum_{\substack{j=1\\j\neq i}}^{|\mathcal{M}|}\left(-\tilde{\boldsymbol{h}}_{di}\tilde{\boldsymbol{h}}_{dj}^{\top} + \log\left(|\mathcal{D}|(|\mathcal{D}|-1)|\mathcal{M}|^2\right) - \frac{\epsilon_{\gamma\beta}^{+2}}{\epsilon_{\gamma\beta}^{-2}} + \frac{\epsilon_{\gamma\beta}^{+2}}{\epsilon_{\gamma\beta}^{-2}}\right) \quad (57)$$

$$\leq 5|\mathcal{D}||\mathcal{M}|(|\mathcal{M}|-1)\epsilon_{\gamma\beta}^{+2} + 2\epsilon_{\gamma\beta}^{-2}\sum_{d,i,j\neq i}\left(-\tilde{\boldsymbol{h}}_{di}\tilde{\boldsymbol{h}}_{dj}^{\top} + \log\left(|\mathcal{D}|(|\mathcal{D}|-1)|\mathcal{M}|^2\right) - \frac{\epsilon_{\gamma\beta}^{+2}}{\epsilon_{\gamma\beta}^{-2}}\right) \quad (58)$$

$$\leq 5|\mathcal{D}||\mathcal{M}|(|\mathcal{M}|-1)\epsilon_{\gamma\beta}^{+2}$$
$$+ 2\epsilon_{\gamma\beta}^{-2}\sum_{d,i,j\neq i}\left(-\tilde{\boldsymbol{h}}_{di}\tilde{\boldsymbol{h}}_{dj}^{\top} + \log\left(|\mathcal{D}|(|\mathcal{D}|-1)|\mathcal{M}|^2\exp(-(\frac{\epsilon_{\gamma\beta}^{+}}{\epsilon_{\gamma\beta}^{-}})^2))\right)\right) \quad (59)$$

$$\leq 5|\mathcal{D}||\mathcal{M}|(|\mathcal{M}|-1)\epsilon_{\gamma\beta}^{+2}$$
$$+ 2\epsilon_{\gamma\beta}^{-2}\sum_{d,i,j\neq i}\left(-\log\exp(\tilde{\boldsymbol{h}}_{di}\tilde{\boldsymbol{h}}_{dj}^{\top}) + \log\left(\sum_{d_1}^{|\mathcal{D}|}\sum_{d_2\neq x_1}^{|\mathcal{D}|}\sum_{k_1}^{|\mathcal{M}|}\sum_{k_2}^{|\mathcal{M}|}\exp(-(\frac{\epsilon_{\gamma\beta}^{+}}{\epsilon_{\gamma\beta}^{-}})^2))\right)\right) \quad (60)$$

$$\leq 5|\mathcal{D}||\mathcal{M}|(|\mathcal{M}|-1)\epsilon_{\gamma\beta}^{+2} + 2\epsilon_{\gamma\beta}^{-2}\sum_{d,i,j\neq i} -\log\frac{\exp(\tilde{\boldsymbol{h}}_{di}\tilde{\boldsymbol{h}}_{dj}^{\top})}{\sum_{d_1}^{|\mathcal{D}|}\sum_{d_2\neq d_1}^{|\mathcal{D}|}\sum_{k_1}^{|\mathcal{M}|}\sum_{k_2}^{|\mathcal{M}|}\exp(-(\frac{\epsilon_{\gamma\beta}^{+}}{\epsilon_{\gamma\beta}^{-}})^2))} \quad (61)$$

$$\leq 5|\mathcal{D}||\mathcal{M}|(|\mathcal{M}|-1)\epsilon_{\gamma\beta}^{+2} + 2\epsilon_{\gamma\beta}^{-2}\sum_{d,i,j\neq i} -\log\frac{\exp(\tilde{\boldsymbol{h}}_{di}\tilde{\boldsymbol{h}}_{dj}^{\top})}{\sum_{d_1}^{|\mathcal{D}|}\sum_{d_2\neq d_1}^{|\mathcal{D}|}\sum_{k_1}^{\mathcal{M}}\sum_{k_2}^{\mathcal{M}}\exp(\tilde{\boldsymbol{h}}_{d_1k_1}\tilde{\boldsymbol{h}}_{d_2k_2}^{\top})} \quad (62)$$

$$\leq 5|\mathcal{D}||\mathcal{M}|(|\mathcal{M}|-1)\epsilon_{\gamma\beta}^{+2} + 2\epsilon_{\gamma\beta}^{-2}|\mathcal{D}|\mathcal{L}_{ds}(\boldsymbol{x}_d, \emptyset) \quad (63)$$

where $\mathcal{L}_{ds}(\cdot, \cdot)$ is defined in Section 2.2.1. Substitute Eq. 63 into Eq. 54, we have:

$$\frac{1}{|\mathcal{D}|}\sum_{d=1}^{|\mathcal{D}|}\mathbb{E}_{\mathcal{S}}\left[\|\boldsymbol{y}_{\boldsymbol{x}_d}^{\mathcal{S}} - \boldsymbol{y}_{\boldsymbol{x}_d}^{\emptyset}\|\right] \leq \mu|\mathcal{S}|\sqrt{5\epsilon_{\gamma\beta}^{+2} + \frac{2\epsilon_{\gamma\beta}^{-2}}{|\mathcal{D}||\mathcal{M}|(|\mathcal{M}|-1)}\sum_{d=1}^{|\mathcal{D}|}\mathcal{L}_{ds}(\boldsymbol{x}_d, \emptyset)} \quad (64)$$

We now investigate how the presence of missing modalities impacts the bound, and consequently, the effectiveness of our approach. Assume each instance $\boldsymbol{x}_d \in \mathcal{D}$ has a missing set $\mathcal{S}_d$ with the same cardinality $|\mathcal{S}|$, i.e., $\mathcal{S} \subset \mathcal{M}$, $|\mathcal{S}_d| = |\mathcal{S}| \; \forall d$. Hence,

$$\sum_{d=1}^{|\mathcal{D}|} \mathcal{L}_{ds}(\boldsymbol{x}_d, \emptyset) = \sum_{d, i, j \neq i} - \log \frac{\exp(\tilde{\boldsymbol{h}}_{di} \tilde{\boldsymbol{h}}_{dj}^\top)}{\sum_{d_1}^{|\mathcal{D}|} \sum_{d_2 \neq d_1}^{|\mathcal{D}|} \sum_{d_1}^{|\mathcal{M}|} \sum_{d_2}^{|\mathcal{M}|} \exp(\tilde{\boldsymbol{h}}_{d_1 k_1} \tilde{\boldsymbol{h}}_{d_2 k_2}^\top)} \tag{65}$$

$$= \sum_{d, i, j \neq i} \log \exp(-\tilde{\boldsymbol{h}}_{di} \tilde{\boldsymbol{h}}_{dj}^\top) + \log \left( \sum_{d_1}^{|\mathcal{D}|} \sum_{d_2 \neq d_1}^{|\mathcal{D}|} \sum_{k_1}^{|\mathcal{M}|} \sum_{k_2}^{|\mathcal{M}|} \exp(\tilde{\boldsymbol{h}}_{d_1 k_1} \tilde{\boldsymbol{h}}_{d_2 k_2}^\top) \right) \tag{66}$$

Let $A_1 \triangleq \sum_{d, i, j \neq i} -\tilde{\boldsymbol{h}}_{di} \tilde{\boldsymbol{h}}_{dj}^\top$ and $A_2 \triangleq \sum_{d_1}^{|\mathcal{D}|} \sum_{d_2 \neq d_1}^{|\mathcal{D}|} \sum_{k_1}^{|\mathcal{M}|} \sum_{k_2}^{|\mathcal{M}|} \exp(\tilde{\boldsymbol{h}}_{d_1 k_1} \tilde{\boldsymbol{h}}_{d_2 k_2}^\top)$, we now further expand each term as follows:

Consider $A_1$:

$$A_1 = \sum_d^{|\mathcal{D}|} \sum_{i, j \neq i}^{|\mathcal{M}|} -\tilde{\boldsymbol{h}}_{di} \tilde{\boldsymbol{h}}_{dj}^\top \tag{67}$$

$$= \frac{|\mathcal{M}|(|\mathcal{M}| - 1)}{(|\mathcal{M}| - |\mathcal{S}|)|\mathcal{S}|} \frac{(|\mathcal{M}| - |\mathcal{S}|)|\mathcal{S}|}{|\mathcal{M}|(|\mathcal{M}| - 1)} \sum_d^{|\mathcal{D}|} \sum_{i, j \neq i}^{|\mathcal{M}|} -\tilde{\boldsymbol{h}}_{di} \tilde{\boldsymbol{h}}_{dj}^\top \tag{68}$$

$$= \frac{|\mathcal{M}|(|\mathcal{M}| - 1)}{(|\mathcal{M}| - |\mathcal{S}|)|\mathcal{S}|} \frac{(|\mathcal{M}| - |\mathcal{S}|)! |\mathcal{S}|!}{|\mathcal{M}|!} \frac{(|\mathcal{M}| - 2)!}{(|\mathcal{S}| - 1)!(|\mathcal{M}| - |\mathcal{S}| - 1)!} \sum_d^{|\mathcal{D}|} \sum_{i, j \neq i}^{|\mathcal{M}|} -\tilde{\boldsymbol{h}}_{di} \tilde{\boldsymbol{h}}_{dj}^\top \tag{69}$$

$$= \frac{|\mathcal{M}|(|\mathcal{M}| - 1)}{(|\mathcal{M}| - |\mathcal{S}|)|\mathcal{S}|} \frac{1}{\binom{|\mathcal{M}|}{|\mathcal{S}|}} \binom{|\mathcal{M}| - 2}{|\mathcal{S}| - 1} \sum_d^{|\mathcal{D}|} \sum_{i, j \neq i}^{|\mathcal{M}|} -\tilde{\boldsymbol{h}}_{di} \tilde{\boldsymbol{h}}_{dj}^\top \tag{70}$$

$$= \frac{|\mathcal{M}|(|\mathcal{M}| - 1)}{(|\mathcal{M}| - |\mathcal{S}|)|\mathcal{S}|} \mathbb{E}_{\mathcal{S}_d} \left[ \sum_d^{|\mathcal{D}|} \sum_{i \in \mathcal{S}_d} \sum_{j \notin \mathcal{S}_d} -\tilde{\boldsymbol{h}}_{di} \tilde{\boldsymbol{h}}_{dj}^\top \right] \tag{71}$$

Under missing modality scenarios, i.e., $\mathcal{S}_d \neq \emptyset$, $\tilde{\boldsymbol{h}}_{di}, \forall\, i \in \mathcal{S}_d$ is approximated as $\frac{1}{|\mathcal{M}| - |\mathcal{S}|} \sum_{j \notin \mathcal{S}_d} \tilde{\boldsymbol{h}}_{dj}$. In other words, we can express Eq. 71 as:

$$A_1 = \frac{|\mathcal{M}|(|\mathcal{M}| - 1)}{(|\mathcal{M}| - |\mathcal{S}|)|\mathcal{S}|} \mathbb{E}_{\mathcal{S}_d} \left[ \sum_d^{|\mathcal{D}|} \sum_{i \in \mathcal{S}_d} \sum_{j \notin \mathcal{S}_d} -\tilde{\boldsymbol{h}}_{di} \tilde{\boldsymbol{h}}_{dj}^\top \right] \tag{72}$$

$$= \frac{|\mathcal{M}|(|\mathcal{M}| - 1)}{(|\mathcal{M}| - |\mathcal{S}|)|\mathcal{S}|} \mathbb{E}_{\mathcal{S}_d} \left[ \sum_d^{|\mathcal{D}|} \sum_{i \in \mathcal{S}_d} \sum_{j \notin \mathcal{S}_d} \frac{1}{|\mathcal{M}| - |\mathcal{S}|} \sum_{k \notin \mathcal{S}_d} -\tilde{\boldsymbol{h}}_{dk} \tilde{\boldsymbol{h}}_{dj}^\top \right] \tag{73}$$

$$\leq \frac{|\mathcal{M}|(|\mathcal{M}| - 1)}{(|\mathcal{M}| - |\mathcal{S}|)^2 |\mathcal{S}|} \mathbb{E}_{\mathcal{S}_d} \left[ -\sum_d^{|\mathcal{D}|} \sum_{i \in \mathcal{S}_d} \sum_{j \notin \mathcal{S}_d} \sum_{k \notin \mathcal{S}_d} \tilde{\boldsymbol{h}}_{dk} \tilde{\boldsymbol{h}}_{dj}^\top \right] \tag{74}$$

$$\leq \frac{|\mathcal{M}|(|\mathcal{M}| - 1)}{(|\mathcal{M}| - |\mathcal{S}|)^2} \mathbb{E}_{\mathcal{S}_d} \left[ -\sum_d^{|\mathcal{D}|} \sum_{j \notin \mathcal{S}_d} \sum_{k \notin \mathcal{S}_d} \tilde{\boldsymbol{h}}_{dk} \tilde{\boldsymbol{h}}_{dj}^\top \right] \tag{75}$$

$$\tag{76}$$

Consider $A_2$:

$$A_2 = \sum_{d_1}^{|\mathcal{D}|} \sum_{d_2}^{|\mathcal{D}|} \sum_{k_1}^{|\mathcal{M}|} \sum_{k_2}^{|\mathcal{M}|} \exp(\tilde{\boldsymbol{h}}_{d_1 k_1} \tilde{\boldsymbol{h}}_{d_2 k_2}^\top) \tag{77}$$

$$= \sum_{d_1, d_2 \neq d_1}^{|\mathcal{D}|} \sum_{k_1}^{|\mathcal{M}|} \left[ \sum_{j_2 \notin \mathcal{S}_{d_2}} \exp(\tilde{\boldsymbol{h}}_{d_1 k_1} \tilde{\boldsymbol{h}}_{d_2 j_2}^\top) + \sum_{i_2 \in \mathcal{S}_{d_2}} \exp(\tilde{\boldsymbol{h}}_{d_1 k_1} \tilde{\boldsymbol{h}}_{d_2 i_2}^\top) \right] \tag{78}$$

$$
= \sum_{d_1, d_2 \neq d_1}^{|\mathcal{D}|} \Bigg\{ \sum_{j_1 \notin \mathcal{S}_{d_1}} \Big[ \sum_{j_2 \notin \mathcal{S}_{d_2}} \exp(\tilde{\bm{h}}_{d_1 j_1} \tilde{\bm{h}}_{d_2 j_2}^{\top}) + \sum_{i_2 \in \mathcal{S}_{d_2}} \exp(\tilde{\bm{h}}_{d_1 j_1} \tilde{\bm{h}}_{d_2 i_2}^{\top}) \Big]
$$

$$
+ \sum_{i_1 \in \mathcal{S}_{d_1}} \Big[ \sum_{j_2 \notin \mathcal{S}_{d_2}} \exp(\tilde{\bm{h}}_{d_1 i_1} \tilde{\bm{h}}_{d_2 j_2}^{\top}) + \sum_{i_2 \in \mathcal{S}_{d_2}} \exp(\tilde{\bm{h}}_{d_1 i_1} \tilde{\bm{h}}_{d_2 i_2}^{\top}) \Big] \Bigg\} \tag{79}
$$

$$
= \sum_{d_1, d_2 \neq d_1}^{|\mathcal{D}|} \Bigg\{ \sum_{j_1 \notin \mathcal{S}_{d_1}} \sum_{j_2 \notin \mathcal{S}_{d_2}} \exp(\tilde{\bm{h}}_{d_1 j_1} \tilde{\bm{h}}_{d_2 j_2}^{\top}) + \sum_{j_1 \notin \mathcal{S}_{d_1}} \sum_{i_2 \in \mathcal{S}_{d_2}} \exp(\tilde{\bm{h}}_{d_1 j_1} \tilde{\bm{h}}_{d_2 i_2}^{\top})
$$

$$
+ \sum_{i_1 \in \mathcal{S}_{d_1}} \sum_{j_2 \notin \mathcal{S}_{d_2}} \exp(\tilde{\bm{h}}_{d_1 i_1} \tilde{\bm{h}}_{d_2 j_2}^{\top}) + \sum_{i_1 \in \mathcal{S}_{d_1}} \sum_{i_2 \in \mathcal{S}_{d_2}} \exp(\tilde{h}_{d_1 i_1} \tilde{h}_{d_2 i_2}^{\top}) \Bigg\} \tag{80}
$$

$$
= \sum_{d_1, d_2 \neq d_1}^{|\mathcal{D}|} \Bigg\{ \sum_{\substack{j_1 \notin \mathcal{S}_{d_1} \\ j_2 \notin \mathcal{S}_{d_2}}} \exp(\tilde{\bm{h}}_{d_1 j_1} \tilde{\bm{h}}_{d_2 j_2}^{\top}) + \sum_{\substack{j_1 \notin \mathcal{S}_{d_1} \\ i_2 \in \mathcal{S}_{d_2}}} \exp \Big( \tilde{h}_{d_1 j_1} \frac{1}{|\mathcal{M}| - |\mathcal{S}|} \sum_{j_2 \notin \mathcal{S}_{d_2}} \tilde{h}_{d_2 j_2} \Big)
$$

$$
+ \sum_{\substack{i_1 \in \mathcal{S}_{d_1} \\ j_2 \notin \mathcal{S}_{d_2}}} \exp \Big( \frac{1}{|\mathcal{M}| - |\mathcal{S}|} \sum_{j_1 \notin \mathcal{S}_{d_1}} \tilde{\bm{h}}_{d_1 j_1} \tilde{\bm{h}}_{d_2 j_2}^{\top} \Big)
$$

$$
+ \sum_{\substack{i_1 \in \mathcal{S}_{d_1} \\ i_2 \in \mathcal{S}_{d_2}}} \exp \Big( \frac{1}{(|\mathcal{M}| - |\mathcal{S}|)^2} \sum_{j_1 \notin \mathcal{S}_{d_1}} \sum_{j_2 \notin \mathcal{S}_{d_2}} \tilde{\bm{h}}_{d_1 j_1} \tilde{\bm{h}}_{d_2 j_2}^{\top} \Big) \Bigg\} \tag{81}
$$

$$
= \sum_{d_1, d_2 \neq d_1}^{|\mathcal{D}|} \Bigg\{ \sum_{\substack{j_1 \notin \mathcal{S}_{d_1} \\ j_2 \notin \mathcal{S}_{d_2}}} \exp(\tilde{\bm{h}}_{d_1 j_1} \tilde{\bm{h}}_{d_2 j_2}^{\top}) + |\mathcal{S}| \sum_{j_1 \notin \mathcal{S}_{d_1}} \exp \Big( \tilde{\bm{h}}_{d_1 j_1} \frac{1}{|\mathcal{M}| - |\mathcal{S}|} \sum_{j_2 \notin \mathcal{S}_{d_2}} \tilde{\bm{h}}_{d_2 j_2} \Big)
$$

$$
+ |\mathcal{S}| \sum_{j_2 \notin \mathcal{S}_{d_2}} \exp \Big( \frac{1}{|\mathcal{M}| - |\mathcal{S}|} \sum_{j_1 \notin \mathcal{S}_{d_1}} \tilde{\bm{h}}_{d_1 j_1} \tilde{\bm{h}}_{d_2 j_2}^{\top} \Big)
$$

$$
+ |\mathcal{S}|^2 \exp \Big( \frac{1}{(|\mathcal{M}| - |\mathcal{S}|)^2} \sum_{j_1 \notin \mathcal{S}_{d_1}} \sum_{j_2 \notin \mathcal{S}_{d_2}} \tilde{\bm{h}}_{d_1 j_1} \tilde{\bm{h}}_{d_2 j_2}^{\top} \Big) \Bigg\} \tag{82}
$$

**Observation.** Exponential is a convex function, hence we apply Jensen's inequality to the followings:

1. $|\mathcal{S}|^2 \exp(\frac{1}{(|\mathcal{M}| - |\mathcal{S}|)^2} \sum_{\substack{j_1 \notin \mathcal{S}_{d_1} \\ j_2 \notin \mathcal{S}_{d_2}}} \tilde{\bm{h}}_{d_1 j_1} \tilde{\bm{h}}_{d_2 j_2}^{\top}) \leq \frac{1}{(|\mathcal{M}| - |\mathcal{S}|)^2} \sum_{\substack{j_1 \notin \mathcal{S}_{d_1} \\ j_2 \notin \mathcal{S}_{d_2}}} \exp(\tilde{\bm{h}}_{d_1 j_1} \tilde{\bm{h}}_{d_2 j_2}^{\top})$

2. $|\mathcal{S}| \sum_{j_1 \notin \mathcal{S}_{d_1}} \exp(\tilde{h}_{d_1 j_1} \frac{1}{|\mathcal{M}| - |\mathcal{S}|} \sum_{j_2 \notin \mathcal{S}_{d_2}} \tilde{\bm{h}}_{d_2 j_2}^{\top}) \leq \frac{1}{|\mathcal{M}| - |\mathcal{S}|} \sum_{\substack{j_1 \notin \mathcal{S}_{d_1} \\ i_2 \notin \mathcal{S}_{d_2}}} \exp(\tilde{\bm{h}}_{d_1 j_1} \tilde{\bm{h}}_{d_2 j_2}^{\top})$

3. $|\mathcal{S}| \sum_{j_2 \notin \mathcal{S}_{d_2}} \exp(\frac{1}{|\mathcal{M}| - |\mathcal{S}|} \sum_{j_1 \notin \mathcal{S}_{d_1}} \tilde{\bm{h}}_{d_1 j_1} \tilde{h}_{d_2 j_2}^{\top}) \leq \frac{|\mathcal{S}|}{|\mathcal{M}| - |\mathcal{S}|} \sum_{\substack{j_1 \notin \mathcal{S}_{d_1} \\ j_2 \notin \mathcal{S}_{d_2}}} \exp(\tilde{\bm{h}}_{d_1 j_1} \tilde{\bm{h}}_{d_2 j_2}^{\top})$

which derive Eq. 82 into:

$$
A_2 \leq \Big[ 1 + 2 \frac{|\mathcal{S}|}{|\mathcal{M}| - |\mathcal{S}|} + \Big( \frac{|\mathcal{S}|}{|\mathcal{M}| - |\mathcal{S}|} \Big)^2 \Big] \sum_{d_1, d_2 \neq d_1}^{|\mathcal{D}|} \sum_{\substack{j_1 \notin \mathcal{S}_{d_1} \\ j_2 \notin \mathcal{S}_{d_2}}} \exp(\tilde{\bm{h}}_{d_1 j_1} \tilde{\bm{h}}_{d_2 j_2}^{\top}) \tag{83}
$$

$$
\leq \Big( \frac{|\mathcal{S}|}{|\mathcal{M}| - |\mathcal{S}|} + 1 \Big)^2 \sum_{d_1, d_2 \neq d_1}^{|\mathcal{D}|} \sum_{\substack{j_1 \notin \mathcal{S}_{d_1} \\ j_2 \notin \mathcal{S}_{d_2}}} \exp(\tilde{\bm{h}}_{d_1 j_1} \tilde{\bm{h}}_{d_2 j_2}^{\top}) \tag{84}
$$

$$\leq \Big(\frac{|\mathcal{M}|}{|\mathcal{M}| - |\mathcal{S}|}\Big)^2 \sum_{d_1,d_2 \neq d_1} \sum_{\substack{j_1 \notin \mathcal{S}_{d_1} \\ j_2 \notin \mathcal{S}_{d_2}}} \exp(\tilde{\boldsymbol{h}}_{d_1 j_1} \tilde{\boldsymbol{h}}_{d_2 j_2}^\top) \tag{85}$$

Substitute Eq. 71 and 85 into Eq. 66, we have:

$$\sum_{d=1}^{|\mathcal{D}|} \mathcal{L}_{ds}(\boldsymbol{x}_d, \emptyset) \tag{86}$$

$$\leq \frac{|\mathcal{M}|(|\mathcal{M}| - 1)}{(|\mathcal{M}| - |\mathcal{S}|)^2} \mathbb{E}_{\mathcal{S}_d} \bigg[ - \sum_d^{|\mathcal{D}|} \sum_{j \notin \mathcal{S}_d} \sum_{k \notin \mathcal{S}_d} \tilde{\boldsymbol{h}}_{dk} \tilde{\boldsymbol{h}}_{dj} \bigg]$$
$$+ \sum_{d,i,j \neq i} \log \bigg[ \Big(\frac{|\mathcal{M}|}{|\mathcal{M}| - |\mathcal{S}|}\Big)^2 \sum_{d_1,d_2 \neq d_1}^{|\mathcal{D}|} \sum_{\substack{j_1 \notin \mathcal{S}_{d_1} \\ j_2 \notin \mathcal{S}_{d_2}}} \exp(\tilde{\boldsymbol{h}}_{d_1 j_1} \tilde{\boldsymbol{h}}_{d_2 j_2}) \bigg] \tag{87}$$

$$\leq \frac{|\mathcal{M}|(|\mathcal{M}| - 1)}{(|\mathcal{M}| - |\mathcal{S}|)^2} \mathbb{E}_{\mathcal{S}_d} \bigg[ - \sum_d^{|\mathcal{D}|} \sum_{j \notin \mathcal{S}_d} \sum_{k \notin \mathcal{S}_d} \tilde{\boldsymbol{h}}_{dk} \tilde{\boldsymbol{h}}_{dj} \bigg]$$
$$+ |\mathcal{M}|(|\mathcal{M}| - 1) \sum_d^{|\mathcal{D}|} \log \bigg[ \Big(\frac{|\mathcal{M}|}{|\mathcal{M}| - |\mathcal{S}|}\Big)^2 \sum_{d_1,d_2 \neq d_1}^{|\mathcal{D}|} \sum_{\substack{j_1 \notin \mathcal{S}_{d_1} \\ j_2 \notin \mathcal{S}_{d_2}}} \exp(\tilde{\boldsymbol{h}}_{d_1 j_1} \tilde{\boldsymbol{h}}_{d_2 j_2}) \bigg] \tag{88}$$

$$\leq \frac{|\mathcal{M}|(|\mathcal{M}| - 1)}{(|\mathcal{M}| - |\mathcal{S}|)^2} \sum_d^{|\mathcal{D}|} \bigg\{ \mathbb{E}_{\mathcal{S}_d} \sum_{j \notin \mathcal{S}_d} \sum_{k \notin \mathcal{S}_d} \bigg[ - \log \exp \tilde{\boldsymbol{h}}_{dk} \tilde{\boldsymbol{h}}_{dj}$$
$$+ \log \bigg[ \Big(\frac{|\mathcal{M}|}{|\mathcal{M}| - |\mathcal{S}|}\Big)^2 \sum_{d_1,d_2 \neq d_1}^{|\mathcal{D}|} \sum_{\substack{j_1 \notin \mathcal{S}_{d_1} \\ j_2 \notin \mathcal{S}_{d_2}}} \exp(\tilde{\boldsymbol{h}}_{d_1 j_1} \tilde{\boldsymbol{h}}_{d_2 j_2}) \bigg] \bigg] \bigg\} \tag{89}$$

$$\leq \frac{|\mathcal{M}|(|\mathcal{M}| - 1)}{(|\mathcal{M}| - |\mathcal{S}|)^2} \sum_d^{|\mathcal{D}|} \bigg\{ \mathbb{E}_{\mathcal{S}_d} \bigg[ - \sum_{j \notin \mathcal{S}_d} \sum_{k \notin \mathcal{S}_d} \log \frac{\exp \tilde{\boldsymbol{h}}_{dk} \tilde{\boldsymbol{h}}_{dj}}{\sum_{d_1,d_2 \neq d_1}^{|\mathcal{D}|} \sum_{\substack{j_1 \notin \mathcal{S}_{d_1} \\ j_2 \notin \mathcal{S}_{d_2}}} \exp(\tilde{\boldsymbol{h}}_{d_1 j_1} \tilde{\boldsymbol{h}}_{d_2 j_2})} \bigg]$$
$$+ \sum_{j \notin \mathcal{S}_d} \sum_{k \notin \mathcal{S}_d} \log \Big(\frac{|\mathcal{M}|}{|\mathcal{M}| - |\mathcal{S}|}\Big)^2 \bigg] \bigg\} \tag{90}$$

$$\leq \frac{|\mathcal{M}|(|\mathcal{M}| - 1)}{(|\mathcal{M}| - |\mathcal{S}|)^2} \sum_d^{|\mathcal{D}|} \bigg\{ \mathbb{E}_{\mathcal{S}_d} \bigg[ \mathcal{L}_{ds}(\boldsymbol{x}_d, \mathcal{S}_d) \bigg] + (|\mathcal{M}| - |\mathcal{S}|)^2 \log \Big(\frac{|\mathcal{M}|}{|\mathcal{M}| - |\mathcal{S}|}\Big)^2 \bigg\} \tag{91}$$

Substitute Eq. 91 in Eq. 97, we have:

$$\frac{1}{|\mathcal{D}|} \sum_{d=1}^{|\mathcal{D}|} \mathbb{E}_{\mathcal{S}} \bigg[ \|\boldsymbol{y}_{\boldsymbol{x}_d}^{\mathcal{S}} - \boldsymbol{y}_{\boldsymbol{x}_d}^{\emptyset}\| \bigg] \tag{92}$$

$$\leq \mu|\mathcal{S}| \sqrt{5\epsilon_{\gamma\beta}^{+2} + \frac{2\epsilon_{\gamma\beta}^{-2}}{|\mathcal{D}||\mathcal{M}|(|\mathcal{M}| - 1)} \sum_{d=1}^{|\mathcal{D}|} \mathcal{L}_{ds}(\boldsymbol{x}, \emptyset)} \tag{93}$$

$$\leq \mu|\mathcal{S}| \sqrt{5\epsilon_{\gamma\beta}^{+2} + \frac{2\epsilon_{\gamma\beta}^{-2}}{(|\mathcal{M}| - |\mathcal{S}|)^2} \bigg\{ \frac{1}{|\mathcal{D}|} \sum_d^{|\mathcal{D}|} \mathbb{E}_{\mathcal{S}_d} \bigg[ \mathcal{L}_{ds}(\boldsymbol{x}_d, \mathcal{S}_d) \bigg] \bigg\} + 2\epsilon_{\gamma\beta}^{-2} \log \frac{|\mathcal{S}|^2}{(|\mathcal{M}| - |\mathcal{S}|)^2}} \tag{94}$$

which is equivalent to:

$$\mathbb{E}_{\boldsymbol{x}, \mathcal{S}} \bigg[ \|\boldsymbol{y}_{\boldsymbol{x}}^{\mathcal{S}} - \boldsymbol{y}_{\boldsymbol{x}}^{\emptyset}\| \bigg] \tag{95}$$

$$\leq \mu|\mathcal{S}|\sqrt{5\epsilon_{\gamma\beta}^{+2} + \frac{2\epsilon_{\gamma\beta}^{-2}}{(|\mathcal{M}|-|\mathcal{S}|)^2}\mathbb{E}_{\boldsymbol{x},\mathcal{S}}\left[\mathcal{L}_{ds}(\boldsymbol{x},\mathcal{S})\right] + 2\epsilon_{\gamma\beta}^{-2}\log\frac{|\mathcal{M}|^2}{(|\mathcal{M}|-|\mathcal{S}|)^2}} \tag{96}$$

$$\leq \mathcal{O}\left(\mu|S|\sqrt{\frac{\mathbb{E}_{\boldsymbol{x},\mathcal{S}}[\mathcal{L}_{ds}(\boldsymbol{x},\mathcal{S})]}{(|\mathcal{M}|-|\mathcal{S}|)^2} + \log\frac{|\mathcal{M}|^2}{(|\mathcal{M}|-|\mathcal{S}|)^2}}\right) \tag{97}$$

# E Complexity Analysis

## E.1 Analysis

We start by introducing the time complexity of traditional FL algorithms, such as `FedAvg`, `FedProx` as a baseline to analyze the time complexity of `PEPSY`. Let:

- $d$: feature extractor size
- $\tau$: number of embedding controls in local data-missing profile.
- $m$:
- $d_p$: embedding control dimensionality.
- $d_k$: key vector dimensionality
- $\kappa$: number of embedding controls selected per query (small constant)
- $E$: local epochs
- $B$: batch size
- $n_k$: local data size
- $M$: number of optimization iterations of PFPT-based clustering

**Table 5:** Comparison of Time and Communication Complexity of PEPSY and traditional FL

| Component | Traditional FL | PEPSY |
|---|---|---|
| Local Computation | $\mathcal{O}\left(E \cdot \frac{n_k}{B} \cdot d\right)$ | $\mathcal{O}\left(\frac{n_k}{B} \cdot E\left[(d + m \cdot d_p) + \tau \cdot d_k\right]\right)$ |
| Client Communication | $\mathcal{O}(d)$ | $\mathcal{O}(d + pd_p)$ |
| Server Aggregation | $\mathcal{O}(Kd)$ | $\mathcal{O}(Kd + MK^2p^2d_p^2)$ |

**Traditional FL.** Each client updates its local parameters over $E$ iterations, with batch size of $B$ using a model of size $d$. This cost: $\mathcal{T}_{local} = \mathcal{O}(E \cdot \frac{n_k}{B} \cdot d)$ Subsequently, the modal parameters of all clients are sent to the server costing: $\mathcal{T}_{com} = \mathcal{O}(d)$ On the server side, all parameters of $K$ clients are combined, typically using variants of weighted average leading to aggregation time cost: $\mathcal{T}_{server} = \mathcal{O}(K \cdot d)$

**PEPSY Modification.** In `PEPSY`, the added cost comes from each client's data-missing profile and the PFPT-based clustering [58]. The time complexities are as follows:

*Embedding Controls Selection.* Each client computes a key vector $q \in \mathbb{R}^{d_k}$ per batch, compares it with $\tau$ controls, and selects top-$\kappa$ controls. If done once per batch, this adds $\mathcal{O}\left(\frac{n_k}{B} \cdot p \cdot d_k\right)$ per round. The selected controls are injected into the model and used during both forward and backward passes. This adds gradient updates with cost $\mathcal{O}(d + \kappa \cdot d_p)$. Over $E \cdot \frac{n_k}{B}$ steps, the total local computation cost is: $\mathcal{T}_{local} = \mathcal{O}\left(\frac{n_k}{B} \cdot E \cdot [(d + m \cdot d_p) + \tau \cdot d_k]\right)$

*Communication Cost.* Clients also send their $p$ with $p \leq \tau$ selected control embeddings (a subset of data-missing profile), adding to the model upload cost: $\mathcal{T}_{com} = \mathcal{O}(d + p \cdot d_p)$

*Server Aggregation and Clustering.* Model aggregation stays at $\mathcal{O}(Kd)$, but PFPT adds overhead from bi-level optimization (over $M$ iterations) and Hungarian matching. Clustering over $Kp$ points add more time complexity to the cost: $\mathcal{T}_{server} = \mathcal{O}(Kd + MK^3p^3d_p^3)$

**Discussion.** While reducing the cost associated with the data-missing profile is nontrivial, the computational cost of the PFPT-based clustering algorithm can be optimized. If we fix the model

**Table 6:** Empirical computational overhead of baselines and proposal comparison.

| Method | Computation Metric | 0.2/0.2 | 0.2/0.4 | 0.2/0.6 | 0.2/0.8 | 0.2/1.0 |
|---|---|---|---|---|---|---|
| FedProx | Training time per round (s) | 50.21 | 50.43 | 50.41 | 49.8 | 49.77 |
| | Inference time (s) | 3.56 | 3.57 | 3.6 | 3.74 | 3.59 |
| | GPU for training (GB) | 2.72 | 2.72 | 2.72 | 2.72 | 2.72 |
| MIFL | Training time per round (s) | 94.11 | 94.04 | 93.92 | 92.98 | 93.34 |
| | Inference time (s) | 4.11 | 4.12 | 4.13 | 4.1 | 4.16 |
| | GPU for training (GB) | 3.26 | 3.26 | 3.26 | 3.26 | 3.26 |
| FedInMM | Training time per round (s) | 100.23 | 97.71 | 97.95 | 99.28 | 96.44 |
| | Inference time (s) | 4.89 | 4.86 | 4.83 | 4.95 | 4.89 |
| | GPU for training (GB) | 2.55 | 2.55 | 2.55 | 2.55 | 2.55 |
| FedMSplit | Training time per round (s) | 86.34 | 86.63 | 86.56 | 86.67 | 86.18 |
| | Inference time (s) | 3.59 | 3.58 | 3.6 | 3.6 | 3.6 |
| | GPU for training (GB) | 3.21 | 3.21 | 3.21 | 3.21 | 3.21 |
| FedMAC | Training time per round (s) | 51.77 | 51.11 | 51.07 | 51.19 | 51.21 |
| | Inference time (s) | 4.56 | 4.98 | 4.69 | 4.69 | 4.88 |
| | GPU for training (GB) | 1.99 | 1.99 | 1.99 | 1.99 | 1.99 |
| PEPSY | Training time per round (s) | 141.12 | 153.95 | 137.66 | 140.12 | 146.48 |
| | Inference time (s) | 4.69 | 4.99 | 4.9 | 4.73 | 4.87 |
| | GPU for training (GB) | 2.61 | 2.63 | 2.15 | 2.86 | 2.8 |

architecture and instead adopt a standard federated learning (FL) approach on the server side, the clustering step is removed, and the total server cost becomes $\mathcal{O}(Kd + K\tau d_p)$, where each client sends its full data-missing profile of size $\tau$.

## E.2 Empirical Overhead

As shown in Table 6, we compared the computational overhead of PEPSY with existing baselines in different $p_m/p_s$ scenarios and found that the additional cost in PEPSY is primarily incurred during training, regardless of the missingness scenario. This aligns with the time complexity analysis, as the PFPT-based clustering algorithm requires more time for clustering. In contrast, PEPSY's inference time and GPU usage remain comparable to other methods, while still delivering superior performance. This is because the data-missing profile is relatively small compared to the model size, adding minimal overhead to each forward pass.

## E.3 Recommended Solution

To improve PEPSY's computational efficiency to match that of traditional FL, we can tune the PFPT clustering cost to stay within this bound. Specifically, by setting $\mathcal{O}(MK^3p^3d_p^3) = \mathcal{O}(K\tau d_p)$. We can solve for $p$ to determine the number of selected controls each client needs to send. Assuming $M$, $K$, and $d_p$ are fixed system parameters, this yields: $p = \mathcal{O}\left(\sqrt[3]{\frac{\tau}{MKd_p}}\right)$. In practice, this can be implemented by having each client transmit only the top-$p$ most frequently selected controls from its profile.

**Table 7:** Impact of top-$p$ most frequently selected controls from each client's profile on overall performance. Experiments are conducted on EDF datasets.

| Method | Overall Accuracy (%) |
|---|---|
| FedProx | 43.24 |
| MIFL | 43.18 |
| FedInMM | 40.56 |
| FedMSplit | 45.18 |
| FedMAC | 49.8 |
| PEPSY ($p = 5$) | 50.77 |
| PEPSY ($p = 10$) | 50.45 |
| PEPSY ($p = 20$) | 51.51 |
| PEPSY (no limit) | 56.36 |

As can be seen from Table 7, our method still outperforms the baselines substantially even when $p$ is reduced to match the cost of `FedAvg`. This is run on the EDF dataset, with $0.2/0.2$ missingness, and for each client, we only take $p$ most selected embedding controls to sent to the server.

# F  Additional Experimental Results

## F.1  Additional Comparison with Baselines

**Extensive Missing Scenarios Analysis.** In addition to the results in the main text, we conducted further experiments comparing the performance of PEPSY (our method) with baselines under more varied missing modality scenarios. Specifically, we expanded the values of $p_m$ and $p_s$ to include 0.4, 0.6, and 1.0, covering a range from 0.2 to 1.0. The results are shown in Tab. 8 and Tab. 10.

As can be seen in these tables, PEPSY consistently outperforms all baselines across all testing scenarios. For the PTBXL dataset (see Tab. 8), the performance gap is small (3% - 4%) when the missing degree is low, e.g., $p_m = 0.2$. However, as the missing degree increases (e.g., $p_m = 0.8$ and $p_m = 1.0$), PEPSY maintains a clear advantage over other methods in both IID and NonIID settings, with a significant gap of approximate $11\%$ in accuracy. Similarly, for the EDF dataset, PEPSY outperforms baselines by a significant margin - up to nearly 10% - across additional missing modality scenarios. This demonstrates the effectiveness and robustness of our approach to missing modalities in federated learning systems, regardless of data heterogeneity.

**Table 8:** Performance of baselines on the PTBXL dataset under various missing patterns in train and test sets, for both IID and Non-IID scenarios. The best and second-best results are highlighted in **bold red** and blue, respectively. We use a hyphen (–) to denote $p_m/p_s = 1.0/1.0$, indicating that all modalities are missing and these cases are excluded from evaluation.

| pm\ps | Method | IID | | | | | NonIID | | | | |
|---|---|---|---|---|---|---|---|---|---|---|---|
| | | 0.2 | 0.4 | 0.6 | 0.8 | 1.0 | 0.2 | 0.4 | 0.6 | 0.8 | 1.0 |
| 0.4 | FedProx | 71.63% | 63.81% | 65.57% | 64.69% | 45.76% | 47.79% | 45.27% | 39.97% | 33.67% | 37.58% |
| | MIFL | 71.37% | 65.95% | 66.46% | 45.02% | 53.85% | 52.59% | 39.22% | 37.33% | 38.08% | 37.20% |
| | FedInMM | 69.61% | 68.35% | 64.69% | 63.43% | 64.19% | 63.43% | 66.33% | 62.29% | 61.66% | 59.52% |
| | FedMSplit | 70.62% | 62.93% | 60.28% | 60.66% | 38.97% | 53.97% | 48.17% | 43.17% | 46.27% | 34.30% |
| | FedMAC | 75.79% | 74.02% | 73.52% | 73.64% | 67.84% | 69.48% | 52.21% | 45.65% | 43.76% | 47.41% |
| | PEPSY | **78.44%** | **77.55%** | **76.04%** | **76.29%** | **71.37%** | **71.12%** | **71.12%** | **68.10%** | **70.87%** | **70.62%** |
| 0.6 | FedProx | 72.38% | 69.74% | 65.07% | 63.18% | 47.41% | 44.01% | 38.08% | 37.45% | 28.75% | 29.00% |
| | MIFL | 70.99% | 67.59% | 55.61% | 49.81% | 25.47% | 56.75% | 43.76% | 43.00% | 35.69% | 25.60% |
| | FedInMM | 67.21% | 61.79% | 59.14% | 58.26% | 25.60% | 62.42% | 59.14% | 49.56% | 56.36% | 49.43% |
| | FedMSplit | 69.10% | 63.81% | 51.45% | 40.48% | 37.07% | 40.73% | 47.29% | 38.71% | 35.43% | 26.48% |
| | FedMAC | 75.28% | 74.02% | 73.52% | 73.64% | 56.75% | 51.45% | 50.44% | 50.06% | 27.87% | 46.15% |
| | PEPSY | **76.55%** | **74.53%** | **74.15%** | **74.15%** | **57.63%** | **70.87%** | **69.23%** | **68.47%** | **68.98%** | **58.76%** |
| 1.0 | FedProx | 75.03% | 72.63% | 68.73% | 58.51% | - | 61.03% | 51.57% | 42.62% | 33.29% | - |
| | MIFL | 73.52% | 71.37% | 66.09% | 47.54% | - | 59.64% | 50.44% | 39.60% | 33.67% | - |
| | FedInMM | 62.80% | 62.42% | 53.97% | 49.68% | - | 59.02% | 54.85% | 50.06% | 41.86% | - |
| | FedMSplit | 72.13% | 68.10% | 66.46% | 54.48% | - | 57.25% | 52.08% | 45.02% | 33.92% | - |
| | FedMAC | 75.16% | 74.40% | 72.38% | 69.74% | - | 59.52% | 44.51% | 51.32% | 41.74% | - |
| | PEPSY | **76.04%** | **77.05%** | **75.03%** | **72.76%** | - | **71.25%** | **67.21%** | **68.60%** | **59.14%** | - |

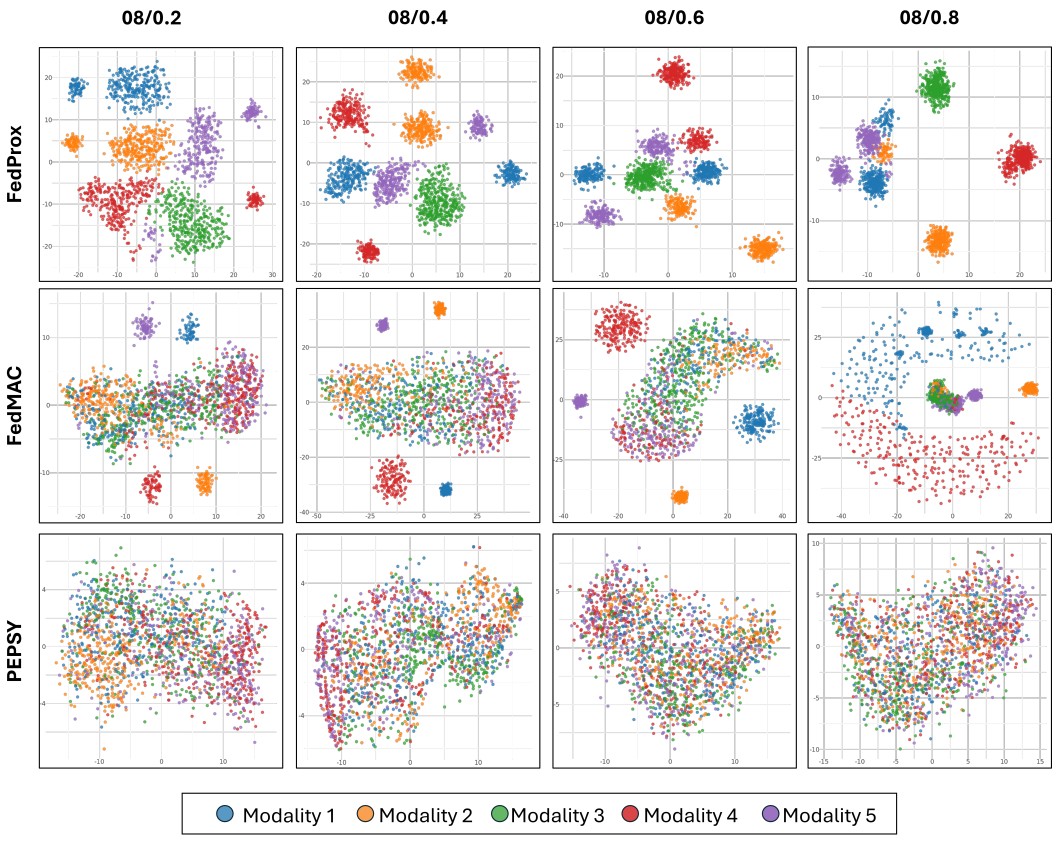

**Figure 7:** Modality representations of different methods under multiple missing scenarios. We train and provide t-SNE 2D visualizations of modality representations constructed by three methods, including our proposal, in different $p_m/p_s$ settings. All experiments are conducted on EDF dataset, nonIID setting.

**Table 9:** Ablation studies on crucial components of PEPSY under different missing statistics ($p_m/p_s$). We report top-1 accuracy across multiple experiments on the EDF dataset, in NonIID setting.

| Method | 0.8/0.2 | 0.8/0.4 | 0.8/0.6 | 0.8/0.8 | 0.8/1.0 |
|---|---|---|---|---|---|
| PEPSY-NP | 46.49% | 47.92% | 52.42% | 52.08% | 43.98% |
| PEPSY-NR | 43.30% | 43.47% | 43.47% | 43.58% | 19.97% |
| PEPSY | **51.80%** | **51.06%** | **55.05%** | **52.25%** | **46.09%** |

**Modality Alignment Analysis.** Fig. 7 compares modality alignment of our proposed PEPSY and two other baselines, namely FedProx and FedMAC, which correspond to traditional FL method and second-best approach in most evaluation experiments. Intuitively, to achieve high performance regardless of available modalities, an optimal solution should align modalities well in a representation space, which hence discards reliance on present modalities. As can be seen from Fig. 7, FedProx and FedMAC fail to align different modalities, indicating their strong dependence on different available modality sets. This is because FedProx does not have a mechanism for modality alignment, while FedMAC discards modality-specific information. In contrast, our proposed PEPSY integrates both modality- and data-specific information, which are futher reconfigured by a shareable data-missing profile leading to less reliance on modalities. The figures show how all modalities are aligned after PEPSY's training, highlighting effectiveness of the proposal under missing modality scenarios.

## F.2  Additional Ablation Studies

In this section, we conduct additional ablation studies on two crucial components in our design: data-missing profile, along with the relevance loss term, and modality fusion, along with the reconfiguration regularization. Correspondingly, we introduce two variants of PEPSY, namely PEPSY-NP (No Profile)

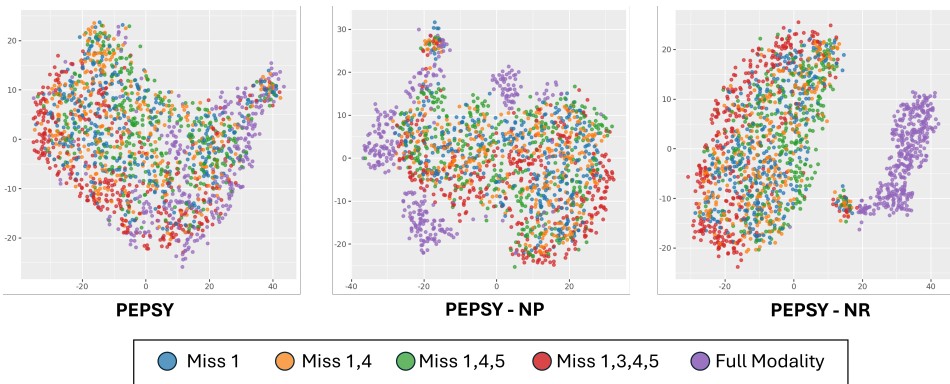

**Figure 8:** Stability of modality representations under different missing modality scenarios. Ideally, a modality's representation should remain stable regardless of which other modalities are missing. This stability is not achieved when either the data-missing profile is removed (-NP version) or the reconfiguration signal is omitted (-NR version) from our proposed PEPSY.

**Table 10:** Performance of baselines on the EDF dataset under various missing patterns in train and test sets, for both IID and Non-IID scenarios. The best and second-best results are highlighted in **bold red** and blue, respectively. We use a hyphen (–) to denote $p_m/p_s = 1.0/1.0$, indicating that all modalities are missing and these cases are excluded from evaluation.

| pm\ps | Method | IID | | | | | NonIID | | | | |
|---|---|---|---|---|---|---|---|---|---|---|---|
| | | 0.2 | 0.4 | 0.6 | 0.8 | 1.0 | 0.2 | 0.4 | 0.6 | 0.8 | 1.0 |
| 0.4 | FedProx | 44.38% | 44.25% | 43.70% | 44.95% | 43.07% | 45.00% | 44.55% | 44.61% | 44.55% | 44.72% |
| | MIFL | 43.35% | 44.72% | 43.72% | 44.89% | 44.66% | 44.61% | 44.67% | 44.72% | 44.49% | 40.27% |
| | FedInMM | 40.50% | 40.50% | 40.67% | 40.56% | 40.90% | 40.62% | 42.38% | 40.50% | 40.45% | 41.19% |
| | FedMSplit | 44.95% | 45.10% | 44.61% | 44.61% | 44.67% | 44.43% | 44.38% | 44.61% | 44.10% | 44.21% |
| | FedMAC | 50.49% | 48.26% | 48.09% | 50.03% | 41.93% | 49.80% | 46.49% | 46.66% | 44.72% | 46.83% |
| | PEPSY | **55.68%** | **55.33%** | **54.54%** | **55.45%** | **49.91%** | **58.02%** | **52.54%** | **49.80%** | **48.32%** | **51.97%** |
| 0.6 | FedProx | 34.91% | 34.23% | 33.14% | 29.89% | 42.61% | 41.24% | 42.50% | 42.56% | 43.18% | 40.45% |
| | MIFL | 44.32% | 42.84% | 43.98% | 44.78% | 44.61% | 45.18% | 44.38% | 44.38% | 44.10% | 44.27% |
| | FedInMM | 40.67% | 40.44% | 40.56% | 40.62% | 40.45% | 41.47% | 41.7% | 40.73% | 40.62% | 40.67% |
| | FedMSplit | 44.38% | 44.55% | 44.61% | 44.44% | 43.47% | 44.15% | 44.55% | 44.15% | 42.27% | 43.53% |
| | FedMAC | 50.99% | 49.40% | 48.66% | 48.20% | 16.71% | 47.80% | 47.46% | 45.58% | 43.64% | 38.62% |
| | PEPSY | **51.28%** | **50.54%** | **50.26%** | **50.60%** | **44.66%** | **48.66%** | **51.12%** | **49.67%** | **51.85%** | **45.07%** |
| 1.0 | FedProx | 36.22% | 35.14% | 33.89% | 31.72% | - | 44.38% | 44.67% | 44.44% | 43.75% | - |
| | MIFL | 42.56% | 42.90% | 41.19% | 41.47% | - | 44.15% | 43.75% | 44.27% | 44.21% | - |
| | FedInMM | 40.45% | 40.56% | 40.50% | 40.22% | - | 40.56% | 40.39% | 40.38% | 40.27% | - |
| | FedMSplit | 43.47% | 43.47% | 42.56% | 41.42% | - | 42.44% | 43.98% | 43.70% | 44.89% | - |
| | FedMAC | 40.22% | 40.45% | 40.96% | 38.11% | - | 47.22% | 46.83% | 46.44% | 46.15% | - |
| | PEPSY | **54.93%** | **52.48%** | **48.49%** | **45.41%** | - | **50.09%** | **48.26%** | **49.67%** | **49.96%** | - |

and PEPSY-NR (No Reconfiguration). To evaluate their contributions in our proposal, we analyse both quantiative and qualitative results.

**Quantitative Results.** Tab. 9 shows impacts of different components on the final components. First, when we remove data-missing profile (see PEPSY-NP variant), the performance drops from 0.2% to 4%, indicating the importance data-missing profile to stablize output performance. In this variant, the reconfiguration supervision signal, a contrastive alignment - based loss, is preserved, hence ensuring modalities are aligned, which are eventually similar to modality fusion in previous works [54, 39]. On the other hand, omitting reconfiguration signal and modality fusion, which results in PEPSY-NR variant, worsen final performance by a larger margin, up to more than 26%. This is because without the reconfiguration signal, the data-missing profile lacks guidance to reconfigure the biased information generated from raw data into complete ones, hence failing to handle missing modalities efficiently. In summary, both components are crucial in our design to ensure robust and stable performance in multimodal federated learning.

**Qualitative Results.** We further visualize representations that each PEPSY variant constructs for an individual modality under different missing scenarios, given the same trained backbone. In particular, each variant is trained on a specific missing statistic $p_m/p_s = 0.8/0.8$ in NonIID setting and tested on

**Table 11:** Performance of baselines under image-sensor modality settings, conduced on EDF dataset across two representative missing scenarios. The best and second-best results are highlighted in **bold red** and blue, respectively.

| pm/ps | FedProx | MIFL | FedInMM | FedMSplit | FedMAC | PEPSY |
|-------|---------|------|---------|-----------|--------|-------|
| 0.2/0.2 | 44.32 | 44.89 | 40.22 | 43.93 | 39.70 | **44.95** |
| 0.8/0.8 | 41.76 | 43.07 | 40.21 | 42.56 | 38.16 | **44.78** |

handcrafted missing tests, including: Miss 1 (modality 1 is missed); Miss 1, 4; Missing 1, 4, 5; Miss 1, 3, 4, 5; Full modality. Intuitively, a representation constructed for modality 1 should remain closely aligned across all tests. As can be seen in Fig. 8, while two ablated variants `PEPSY-NP` and `PEPSY-NR` fail to ensure this stability, our proposed `PEPSY` can construct closely aligned representations in all settings, highlighting its stable feature construction. This is because our data-missing profile effectively distills data-missing information from raw data, which are used later for reconfiguration. These visualizations further emphasize completeness of our design.

### F.3 Ablation on different forms of modality

To evaluate our proposed `PEPSY` framework more comprehensively, we conduct an additional experiment in an image-sensor multi-modal setting to show the broad generality of the proposal, instead of sensor-based modality settings as the original benchmark datasets. In specific, we converted one signal-based modality into an image showing fluctuation of the signal, leading to an image-based modality. Our algorithm has access to this image-based modality but not the original signal-based modality. It will learn to combine this image-based modality with other signal-based modality to make accurate predictions. We further replaced the corresponding feature extractor as a simple convolutional neural network to handle image-based modality, and run several experiments to show the efficiency of our algorithm on different modality domains. Each image modality is of size $128 \times 64$, and normalized to scale from 0 to 1, as presented in Table 11.

Table 11 shows that even under missing settings with different modality forms, `PEPSY` still outperforms all other baselines. This additional experiment futher emphasizes the superiority of `PEPSY` in the ability to handle severe missingness.

## G  Limitations

Although `PEPSY` outperforms prior methods in handling heterogeneous data-missing patterns in multimodal federated learning, it may face challenges when downstream task domains vary significantly. Large domain shifts can create distinct, domain-specific missing data profiles that require more trainable embeddings for effective adaptation. A key open question is whether we can quantify these shifts and bound the number of embeddings needed for reconfiguration—an issue beyond this work's scope but important for future research, especially in federated settings with clients operating in diverse domains and missing data patterns. Moreover, this study relies on training models from scratch and does not leverage pretrained foundation models. Future efforts could explore incorporating pretrained encoders to build shareable missing data profiles, improving representation learning efficiency and effectiveness.

