# OpenReview forum: "Learning Reconfigurable Representations for Multimodal Federated Learning with Missing Data"
_NeurIPS.cc/2025/Conference — NeurIPS 2025 poster_

### Official Review · Reviewer_iEDv · 2025-07-02

**Clarity:** 1
**Significance:** 3
**Originality:** 3
**Rating:** 4
**Confidence:** 4

**Summary:**

This paper addresses the challenge of multimodal federated learning (MMFL) in the presence of incomplete and heterogeneous data, where clients may have access to different subsets of modalities and suffer from partial feature missingness within each modality. The authors propose PEPSY, a novel federated learning framework that introduces learnable client-specific embedding controls to capture and communicate each client’s unique data-missing patterns. These embeddings serve as reconfiguration signals that align the global model with each client's local context, enabling more effective and robust representation learning.

The framework operates by:
1.	Extracting modality-specific and data-specific features on the client side.
2.	Using a query-key mechanism to select relevant embeddings from a shared pool that reflect the missing patterns.
3.	Aggregating embedding controls across clients via a probabilistic clustering-based alignment on the server.

Theoretical analysis provides performance bounds showing that PEPSY’s design minimizes prediction error due to missing modalities. Empirical evaluations on PTBXL and Sleep-EDF datasets under IID and Non-IID settings demonstrate significant improvements over baselines, achieving up to 36.45% accuracy gains in high missingness scenarios.

**Questions:**

Before addressing specific questions, I strongly encourage the authors to review the **Clarity** section under **Weaknesses**, particularly issues raised in Sections 2.1–2.3. Several definitions, notational choices, and design rationales lack sufficient explanation. Addressing these concerns would significantly improve the comprehension of the paper and may positively influence the overall assessment.

Below are the key questions that, if clarified, could strengthen the paper further:
1. **Clarify the Construction and Scope of the Data-Missing Profile ($\Psi$)**
   How is the data-missing profile $\Psi$ initialized and updated during training? Is there a single global profile, or does each client maintain a local version? Why is there no client index in the notation if the profiles encode client-specific missingness patterns?

2. **Explain and Justify the Optimization Formulation (Eq. 3)**
   Why is the original federated loss in Eq. (1) replaced with a difference involving the utility term $u_k(\theta, \Psi)$? What does this subtraction represent, and how should it be interpreted?

3. **Clarify the Use of Contrastive Learning for Reconfiguration (Section 2.2.3)**
   Why is contrastive loss suitable for aligning partial and full-modality representations? What exactly are the positive and negative sample pairs used during training?

4. **Describe PFPT and Its Role in Server Aggregation (Section 2.3)**
   What are the main characteristics of PFPT, and why is it an appropriate choice for aggregating the data-missing profile across clients?

5. **Discuss the Lipschitz Continuity Assumption in Theorem 3.1**
   Given that neural networks rarely satisfy strict Lipschitz conditions, how practical is this assumption? Can the bound be extended or relaxed under more realistic constraints?

6. **Clarify the Definition of "Modality" Used in the Dataset Description**
   The paper states that PTBXL has 12 modalities and Sleep-EDF has 5, but these datasets are generally not considered to contain that many distinct modalities under standard definitions (e.g., text, audio, image). Could the authors clarify what they count as a modality in this context?

7. **Provide an Analysis of Computational Complexity**
   The proposed method introduces multiple components—embedding control pools, query-key relevance selection, contrastive regularization, and PFPT-based aggregation. Can the authors provide an estimate or breakdown of computational and communication overhead introduced by each component?

**Ethical Concerns:**

["NO or VERY MINOR ethics concerns only"]

**Final Justification:**

After a few back-and-forth rounds, I was satisfied with the author's response during the rebuttal.

**Limitations:**

Yes

**Quality:**

2

**Strengths And Weaknesses:**

## **Strengths**

### **Quality**
- The method addresses an important and realistic challenge in multimodal federated learning (MMFL): handling both *missing modalities across clients* and *missing features within modalities*. The proposed PEPSY framework is designed to tackle this dual challenge and is evaluated through solid empirical experiments.

### **Clarity**
- The paper clearly motivates the real-world relevance of MMFL under partial observability and articulates the limitations of prior work. The distinction between different types of "missing-ness" (modality-level and feature-level) is helpful for framing the contribution.

### **Significance**
- The addressed problem is important for real-world applications involving distributed sensing and privacy-preserving learning, particularly in domains like healthcare and IoT. The strong empirical gains and general design framework make this work a promising direction for advancing robust and flexible MMFL.

### **Originality**
- While the paper proposes a novel solution, there is some conceptual proximity to existing work:
  - **Personalized Federated Learning (PFL)**: Many PFL approaches use local embeddings or client-specific signals to personalize global models. PEPSY similarly leverages client-side latent information via embedding control pools.
  - **Domain Adaptation**: The idea of learning latent codes to adapt shared models to local contexts mirrors approaches in domain adaptation, especially under distribution shifts.
---

## **Weaknesses**

### **Clarity**

While the paper presents a compelling high-level idea, the technical exposition—especially in **Section 2**—is difficult to follow. Several core concepts and mathematical formulations lack clarity or motivation, making it hard to understand the method in depth. Specific issues include:

- **Data-Missing Profile Ambiguity (Section 2.1–2.2.2)**
  It is unclear how the data-missing profile $\Psi$ is computed or updated, whether it is global or client-specific, and how it's initialized. The absence of a client index in the notation adds confusion, especially given the claim that the profile reflects local missingness patterns.

- **Equation Transitions (Section 2.1)**
  The transition from Eq. (1) to Eq. (3) introduces a new utility term $u_k$ without sufficient justification. Its role in the optimization problem, and why the objective is now a difference of two terms, is not explained intuitively or mathematically.

- **Inconsistent Notation (Section 2.1)**
  In the expression for $u_k(\theta, \Psi)$, the right-hand side omits $\theta$, though it appears to depend on model parameters via the relevance function. This creates notational confusion in a key part of the formulation.

- **Unclear Objectives (Section 2.1)**
  The claim that minimizing $\ell_k(\theta, \Psi)$ helps extract data-missing features, and that maximizing $u_k(\theta, \Psi)$ helps $\Psi$ adapt to local context, is not justified or supported by theory or experiments.

- **Vague Terminology (Section 2.1–2.2.2)**
  Phrases like “reconfigured by $\Psi$” or “diluting learned data-missing profile” are conceptually vague and lack clear operational definitions, making the method harder to interpret.

- **Embedding Controls Design (Section 2.2.2)**
  The identity function is used as the key function $k(\cdot)$ in a query-key matching setup, with no justification for its effectiveness. Also, the decision to limit relevance to only $\kappa$ embeddings is explained in abstract terms without empirical motivation or ablation.

- **Contrastive Learning Justification (Section 2.2.3)**
  The use of contrastive loss for reconfiguration regularization is introduced abruptly. It is unclear why this choice is suitable for ensuring “data completeness,” and the sampling of positives and negatives is not specified.

- **Definition of $h_{di}$ (Section 2.2.1)**
  The modality-specific feature $h_{di}$ is mentioned as a key component in representation construction, but its computation is unspecified. There’s no mention of what neural layers produce it or how it fits into the overall architecture.

- **Server Aggregation with PFPT (Section 2.3)**
  The server uses PFPT (probabilistic federated prompt tuning) for aggregating data-missing profiles, but this method is not described. Readers unfamiliar with PFPT are left without context on its mechanism, benefits, or why it was chosen over alternatives.

### **Quality**

- **Computational Complexity**
  The method comprises several sophisticated components—embedding control pools, query-key matching, contrastive and alignment losses, and probabilistic clustering (PFPT). These likely incur nontrivial computational and communication overhead, but the paper does not discuss the system cost or efficiency.

- **Limited Dataset Variety**
  Although performance on PTBXL and Sleep-EDF is promising, both datasets are from biosignal/health domains. Broader validation on more diverse multimodal settings (e.g., video+audio, language+vision) would better support the generality of the approach.

### **Significance**

- **Strong Theoretical Assumptions (Section 3)**
  Theorem 3.1 assumes that the model is $\mu$-Lipschitz continuous—a strong condition not typically satisfied by deep networks. The paper does not discuss whether this assumption is realistic or how sensitive the bound is to its violation, limiting the practical applicability of the theoretical result.

### **Originality**

- **Conceptual Overlap with Prior Work (Section 2.1–2.2.2)**
  While PEPSY presents a technically novel mechanism, its conceptual structure shares similarities with:
  - **Personalized Federated Learning (PFL):** where client-specific embeddings or local statistics are used to personalize global models.
  - **Domain Adaptation:** where shared models are adapted using domain-specific latent codes.

  To more clearly distinguish the contribution, the authors could:
  - Emphasize the novelty of *cross-client collaboration via clustering* of embedding profiles.
  - Provide explicit empirical and theoretical comparisons to relevant PFL and domain adaptation baselines.

---

> ### Author Rebuttal · Authors · 2025-07-31
>
> We thank the reviewer for the detailed questions. We summarize the overall conceptual workflow of PEPSY and address the reviewer’s questions below.
>
> **Client-side**: The server sends the global model parameters and data-missing profiles encoded within a set of embedding controls to each client. Given a multimodal input, each client computes features of its data-missing patterns (Sec. 2.2.1) and uses a key-query mechanism to select relevant controls that help generate data-complete representations (Sec. 2.2.2). This selection is guided by the reconfiguration signal (Sec. 2.2.3), and the resulting features are fused via an attention module to produce the final prediction.
>
> **Server-side**: Each client maintains its own data-missing profile as a set of embedding controls encoding various missing patterns. To align embedding controls encoding similar patterns, we use PFPT - a non-parametric clustering method - to cluster and aggregate them across clients. The non-parametric nature of PFPT helps the global data-missing profile (i.e., global embedding clusters) grow with the diversity and misalignment across clients, making it suitable for the varying missingness settings.
>
> # Q1. Data-missing profile clarity
> In PEPSY, the server maintains a global data-missing profile, initialized as a small set of random embedding controls, with each client holding a local copy. This profile is refined through iterative client-side training and server-side clustering. After training, each client updates its local controls and sends them to the server. To resolve cross-client misalignment, the server applies a non-parametric clustering algorithm to group controls encoding similar patterns and compute global controls as the corresponding cluster centroids, which are then broadcast to clients for the next round.
>
> In Eq. (2), the data-missing profile does not include a client index because it denotes the global data-missing profile that we aim to learn. Eq. (2) denotes the reconfigured version of the original model given the learned data-missing profile (via FL). This data-missing profile will be learned via solving Eq. (3) which characterizes the distributed computation structure of FL.
>
> # Q2. Explain new FL formulation in Eq (3)
> Eq. (1) characterizes the standard FL formulation without accounting for missing-data patterns. Eq. (3) augments it to **explicitly model and learn the missing-data profile** encoded with the set of embedding controls $\Psi$. The subtraction of $u_k(\theta, \Psi) $ from the prediction loss $ \ell_k( \theta,\Psi) $ represents a regularization of local learning  $ \ell_k( \theta, \Psi) $ with a missing-aware signal $u_k(\theta, \Psi)$. This signal guides the client in reconfiguring its representations based on missing patterns. Minimizing the regularization loss $-u_k(\theta, \Psi) $ updates the embedding controls to reflect the client’s data context, which are then shared and aggregated by the server for the next round.
>
> # Q3. Contrastive loss for aligning partial and full modality features
> We do not use contrastive loss to align partial and full modality features, but to ensure that **reconstructed modalities of the same input map to similar coordinates in a semantic space**. Specifically, we apply the same missing-aware reconfiguration mechanism to generate a reconstructed representation $w\_{di}$ for each modality $i$ of input $x\_{di}$, regardless of its missing pattern. For the same input, we encourage all reconstructed modalities to align in the semantic space, while modalities from different inputs remain distant, ensuring semantic consistency within each instance. Let $ \hat{w}\_{di}$ the semantic coordinates of $w_{di}$ extracted via a neural projector. The loss uses positive pairs $(\hat{w}\_{di}, \hat{w}\_{dj}) $ from the same input and negative pairs $ (\hat{w}\_{d'i}, \hat{w}\_{dj}) $ from different inputs.
>
> # Q4. Main characteristics of PFPT
> A key feature of PFPT is its non-parametric clustering nature, which allows the number of clusters to grow with the data complexity. Since PEPSY represents each local data-missing profile as a set of embedding controls, scaling with missingness, PFPT is well-suited for constructing these profiles under varying missing patterns.
>
> PFPT is a prompt aggregation method that uses non-parametric clustering. It treats each local model as a set of prompts, which are shared with the server. The server merges these into global prompt sets by clustering similar prompts. The number of clusters adjusts automatically based on the complexity of the data, allowing the model to scale with data diversity.
>
> Our data-missing profile, a set of embedding controls, shares similarities with prompt sets. In PEPSY, each client updates the global profile using its private data, producing a locally augmented set of controllers. The size and complexity of these sets grow with the client's missingness level. The server's goal is to (1) match similar augmented controllers and (2) let the global profile grow with the complexity of data-missing patterns, making PFPT a natural fit.
>
> However, we emphasize that in the original work, PFPT performs clustering on the prompt set in foundation FL to handle different data distribution and label skewness patterns, while PEPSY aligns different missing profiles across clients.  **For the effectiveness of PFPT-based clustering, we refer the reviewer to Fig. 4b in our paper, showing its trajectory and convergence.**
>
> # Q5. Assumption in theoretical analysis
> Due to limited space, we refer the reviewer to our response to Reviewer nfrN (Q1).
>
> # Q6. Modality clarity in experiments
> In our setting, a **modality refers to a sensor signal**, as used in both the EDF and PTB-XL datasets. Each input comprises signals from multiple sensors, where each sensor captures a distinct aspect of the patient's condition. Since these signals vary in distribution and scale, they are treated as separate modalities. This setup is consistent with prior works [1].
>
> [1] Nguyen et al., FedMAC: Tackling Partial-Modality Missing in Federated Learning with Cross-Modal Aggregation and Contrastive Regularization, NCA 2024
>
> # Q7, Computation cost
> Our method does incur client-side communication overhead and server-side processing overhead. However, such **overhead is not substantial**. We also note that the parameter configuration of PFPT-based clustering can be tuned to match the conventional cost of FedAvg while preserving performance advantage. Due to limited space, please refer to our response to Reviewer n9eK for detailed explanation of the above.
>
> # Q8. Experiments on other multimodal domains
> Following the reviewer’s suggestion, we added an experiment in an image-sensor multimodal setting to demonstrate the method’s generality. Here, a signal-based modality was converted into an image representing its fluctuation, and the original signal was withheld. Results are shown below.
> | pm/ps   | FedProx | MIFL  | FedInMM | FedMSplit | FedMAC | PEPSY |
> |---------|---------|-------|---------|-----------|--------|-------|
> | 0.2/0.2 | 44.32   | 44.89 | 40.22   | 43.93     | 39.7   | 44.95 |
> | 0.8/0.8 | 41.76   | 43.07 | 40.21   | 42.56     | 38.16  | 44.78 |
>
> **The above reported results confirm that PEPSY still outperforms the existing baselines in both slightly and extremely missing scenarios (0.2/0.2 and 0.8/0.8)**. Similarly, we observe that PEPSY is more consistent compared to other baselines with minimal performance gap under increasing missingness.
>
> # Q9. Discussing related works on personalized FL and domain adaptation.
> While personalized FL and domain adaptation methods tailor models to individual clients or domains, PEPSY takes a different approach, focusing on the global FL setting by building a shared profile of data-missing patterns across all clients.
>
> Personalized FL tailors models to local data distributions, producing one model per client and evaluating performance across them [1,2]. Domain adaptation transfers knowledge from a source to a target domain using alignment techniques [3,4], but typically assumes centralized settings and complete data. Both approaches address distribution shifts, not missing data, and it remains unclear how they could be adapted to effectively handle modality-level missingness.
>
> In contrast, PEPSY is designed to recognize and align diverse missing data patterns across clients into a holistic data-missing profile. This is achieved via aggregating embedding controls from clients - these controls are designed to capture missing-data patterns, not local model behavior. Unlike prior work, these embeddings are not stored on-device nor used for local fine-tuning. Instead, they are regularized (as in Sec. 2) to help reconfigure missing features into complete representations, addressing the system-level bias issue outlined in Sec. 1.
>
> Additionally, PEPSY introduces a non-parametric PFPT-based clustering algorithm designed for the highly heterogeneous nature of our FL setup. Clients have diverse and misaligned missing patterns, often with varying embedding orders, making a fixed number of clusters too restrictive. The non-parametric design allows flexible grouping and better captures the structural diversity of missing-data profiles. In contrast, existing PFL methods [1,2] do not address this misalignment, which can lead to feature collapse. **To our knowledge, PEPSY is the first to explicitly model missing patterns across clients and aggregate them into a system-level data-missing profile.**
>
> [1] Marfoq, O. et al. Personalized Federated Learning through Local Memorization. ICML 2022
>
> [2] Tan et al. Federated learning from pre-trained models: a contrastive learning approach. Neurips 2022
>
> [3] Sun et al. PartialFed: Cross-Domain Personalized Federated Learning via Partial Initialization. Neurips 2021
>
> [4] Hu et al. Learn to Preserve and Diversify: Parameter-Efficient Group with Orthogonal Regularization for Domain Generalization. ECCV 2024

---

> > ### Comment · Reviewer_iEDv · 2025-08-06
> >
> > I thank the authors for their responses. I have one follow-up suggestion and one follow-up question:
> >
> > I suggest that the authors rewrite section 2.3 server aggregation using their answer to our question “Main characteristics of PFPT”. Also, mention “the server applies a non-parametric clustering algorithm to group controls encoding similar patterns and compute global controls as the corresponding cluster centroids, which are then broadcast to clients for the next round.” in this section to help reader understand how the server sends the global profile to all clients.
> >
> > As for the follow-up question: in the paper, you mention that the number of embedding controls is \tau. At the same time, you also mention in the response that the number of clusters adjusts automatically based on the complexity of the data, allowing the model to scale with data diversity. So is the number of embedding controls changing from global round to global round or fixed?

---

> > > ### Author Response · Authors · 2025-08-07
> > > **Clarify the number of embedding controls across rounds**
> > >
> > > Dear Reviewer iEDv,
> > >
> > > Thank you for your suggestion. We will revise Section 2.3 as you suggested.
> > >
> > > To answer your question:
> > >
> > > **Yes, the number of embedding controls varies from round to round**. Specifically, $\tau$ denotes the number of global embedding controls sent to each client per round. After receiving the global control embeddings from the server, each client selects most relevant control embeddings to update locally and share its control set with the server which clusters them and re-computes global control embeddings as cluster centroids. This updated control set will be sent to each participating client in the next round. The number of global control embeddings $\tau$ (i.e., no. of embedding clusters) can change across rounds as new clients participate and reveal new data-missing patterns. This mechanism is important because (1) the algorithm does not know in advance how many distinct data-missing patterns exist, and (2) not all clients participate in each round (per standard FL settings), so it may take several rounds for the server to observe all patterns.
> > >
> > > We hope this helps address the question. If so, we hope the reviewer will consider re-evaluate the overall rating of our work. Otherwise, please let us know if you still have follow-up questions as the author-reviewer period has been extended until Aug 8. We look forward to hearing your final thoughts.
> > >
> > > Thank you very much.
> > >
> > > Best regards,
> > >
> > > Authors

---

> > > > ### Comment · Reviewer_iEDv · 2025-08-07
> > > >
> > > > Thanks. I have adjusted my score.

---

> > > > > ### Author Response · Authors · 2025-08-08
> > > > > **Thank you for engaging in the discussion**
> > > > >
> > > > > Dear Reviewer iEDv,
> > > > >
> > > > > Thank you for your constructive feedback and engaging in the discussion.
> > > > >
> > > > > We are glad our response has addressed your concerns.
> > > > >
> > > > > Best regards,
> > > > >
> > > > > Authors

---

### Official Review · Reviewer_n9eK · 2025-07-02

**Clarity:** 4
**Significance:** 4
**Originality:** 3
**Rating:** 5
**Confidence:** 4

**Summary:**

The paper studies multimodal federated learning under both missing modalities and missing features. The authors proposed a reconfiguration based framework with a data-missing embedding profile. The missing data will be carefully reconstructed based on the profile, which will be then fused with the available data. There is also a theoretical analysis on the convergence behavior of the model under missing data. Comprehensive experiments agains SOTA baselines show accuracy improvements under severe data incompleteness.

**Questions:**

* What is the overhead of the method compared to existing baselines?
* How will k, which controls the number of relevant controls selected for each instance, affect the performance?

**Ethical Concerns:**

["NO or VERY MINOR ethics concerns only"]

**Final Justification:**

The authors have provided the results to address my concerns. I would recommend accepting the paper, as it is a solid paper that addresses a real-world problem and has conducted comprehensive experiments.

**Limitations:**

Would be good to discuss the limitations in the main paper instead in the appendix

**Quality:**

3

**Strengths And Weaknesses:**

Strength:
* The paper is organized clearly and easy to follow
* The proposed idea of using learnable client-side embedding controls makes sense and is interesting
* Comprehensive experiments are conducted under different missing statistics.

Weakness:
* The overhead of the method compared to existing baselines are not analyzed
* The sensitivity of parameter k, which controls the size of the neighborhood in embedding control, is not analyzed

---

> ### Author Rebuttal · Authors · 2025-07-31
>
> We thank the Reviewer for recognizing our contribution and for the constructive feedback. We address all questions from the reviewer below.
> > **Q1. The overhead of the method compared to existing baselines are not analyzed**
>
> Our method does incur client-side communication overhead and server-side processing overhead. However, such **overhead is not substantial**. We analyze these overheads via a detailed time complexity analysis below.
>
> We start by introducing the time complexity of traditional FL algorithms (FedAvg, FedProx) as a baseline to analyze the time complexity of PEPSY.
>
> ## Notations and Definitions
>
> Let:
> * $d$: feature extractor size
> * $\tau$: number of embedding controls in local data-missing profile.
> * $d_p$: embedding control dimensionality.
> * $d_k$: key vector dimensionality
> * $\kappa$: number of embedding controls selected per query (small constant)
> * $E$: local epochs
> * $B$: batch size
> * $n_k$: local data size
> * $p$: number of embedding controls each client sends per round
> * $M$: number of optimizing steps in PFPT
>
> | Component            | Traditional FL                                          | PEPSY                                                                                               |
> |----------------------|---------------------------------------------------------|-----------------------------------------------------------------------------------------------------|
> | Local Computation    | $\mathcal{O}\left(E \cdot \frac{n\_k}{B} \cdot d\right)$ | $\mathcal{O}\left( \frac{n\_k}{B} \cdot E \left[ (d + m \cdot d\_p) + \tau \cdot d\_k \right] \right)$ |
> | Client Communication | $\mathcal{O}(d)$                                        | $\mathcal{O}(d + p d_p)$                                                                            |
> | Server Aggregation   | $\mathcal{O}(K d)$                                      | $\mathcal{O}( Kd + MK^2p^2d_p^2 )$                                                                  |
>
> ## Detailed Analysis
>
> **Traditional FL**. Each client updates its local parameters over $E$ iterations, with batch size of $B$ using a model of size $d$. This cost:
>
> $ \mathcal{T}_{local} = \mathcal{O}(E \cdot \frac{n_k}{B} \cdot d)$
>
> Subsequently, the modal parameters of all clients are sent to the server costing:
>
> $\mathcal{T}_{com} = \mathcal{O}(d)$
>
> On the server side, all parameters of $K$ clients are combined, typically using variants of weighted average leading to aggregation time cost:
>
> $\mathcal{T}_{server} = \mathcal{O}(K \cdot d)$
>
> **PEPSY Modification**.
> In PEPSY, the added cost comes from each client's data-missing profile and the PFPT-based clustering. The time complexities are as follows:
>
> *Embedding Controls Selection*.
>
> Each client computes a key vector $q \in \mathbb{R}^{d_k}$ per batch, compares it with $\tau$ controls, and selects top-$\kappa$ controls. If done once per batch, this adds $\mathcal{O}\left( \frac{n_k}{B} \cdot p \cdot d_k \right)$ per round. The selected controls are injected into the model and used during both forward and backward passes. This adds gradient updates with cost $\mathcal{O}( d + \kappa \cdot d_p )$. Over $E \cdot \frac{n_k}{B}$ steps, the total local computation cost is:
> $\mathcal{T}_{\text{local}} = \mathcal{O}\left( \frac{n_k}{B} \cdot E \cdot \left[ (d + m \cdot d_p) + \tau \cdot d_k \right] \right)$
>
> *Communication Cost*.
>
> Clients also send their $p$ with $\ p \leq \tau$ selected control embeddings (a subset of data-missing profile), adding to the model upload cost:
> $\mathcal{T}_{\text{com}} = \mathcal{O}(d + p \cdot d_p)$
>
> *Server Aggregation and Clustering*.
>
> Model aggregation stays at $\mathcal{O}(K d)$, but PFPT adds overhead from bi-level optimization (over $M$ iterations) and Hungarian matching. Clustering over $Kp$ points add more time complexity to the cost:
> $\mathcal{T}_{\text{server}} = \mathcal{O}( K d + M K^2 p^2 d_p^2 )$
>
> Assuming a fixed model architecture, standard FL methods (e.g., FedAvg, FedProx) incur a cost of $\mathcal{O}(K d + K \tau d_p)$.
> To match this, PFPT can achieve similar efficiency by setting $p = \mathcal{O}\left( \sqrt{ \frac{\tau}{M K d_p} } \right)$, given fixed $M$, $K$, and $d_p$. We have verified this via the following ablation studies on $p$.
>
> **As reported below, our method still outperforms the baseline substantially even when $p$ is reduced to match the computation cost of FedAvg.**
> This is run on the EDF dataset, with 0.2/0.2 missingness. For each client, we only take $p$ most selected embedding controls to send to the server per round.
>
>
> |      Method      | Accuracy (%) |
> |:----------------:|:------------:|
> |      FedProx     |     43.24    |
> |       MIFL       |     43.18    |
> |      FedInMM     |     40.56    |
> |     FedMSplit    |     45.18    |
> |      FedMAC      |     49.8     |
> |   PEPSY (p = 5)  |     50.77    |
> |  PEPSY (p = 10)  |     50.45    |
> |  PEPSY (p = 20)  |     51.51    |
> | PEPSY (no limit) |     56.36    |
>
> ## Empirical estimation
>
> We compared the computational overhead of PEPSY with existing baselines in different $p_m/p_s$ scenarios and found that the additional cost in PEPSY is primarily incurred during training, regardless of the missingness scenario. This aligns with the time complexity analysis, as the PFPT-based clustering algorithm requires more time for clustering.
> In contrast, **PEPSY's inference time and GPU usage remain comparable to other methods, while still delivering superior performance**. This is because the data-missing profile is relatively small compared to the model size, adding minimal overhead to each forward pass.
>
> | Method    | Metric                      | 0.2/0.2 | 0.2/0.4 | 0.2/0.6 | 0.2/0.8 | 0.2/1.0 |
> |-----------|-----------------------------|:-------:|:-------:|:-------:|:-------:|:-------:|
> | FedProx   | Training time per round (s) |  50.21  |  50.43  |  50.41  |   49.8  |  49.77  |
> |           | Inference time (s)          |   3.56  |   3.57  |   3.6   |   3.74  |   3.59  |
> |           | GPU for training (GB)       |   2.72  |   2.72  |   2.72  |   2.72  |   2.72  |
> | MIFL      | Training time per round (s) |  94.11  |  94.04  |  93.92  |  92.98  |  93.34  |
> |           | Inference time (s)          |   4.11  |   4.12  |   4.13  |   4.1   |   4.16  |
> |           | GPU for training (GB)       |   3.26  |   3.26  |   3.26  |   3.26  |   3.26  |
> | FedInMM   | Training time per round (s) |  100.23 |  97.71  |  97.95  |  99.28  |  96.44  |
> |           | Inference time (s)          |   4.89  |   4.86  |   4.83  |   4.95  |   4.89  |
> |           | GPU for training (GB)       |   2.55  |   2.55  |   2.55  |   2.55  |   2.55  |
> | FedMSplit | Training time per round (s) |  86.34  |  86.63  |  86.56  |  86.67  |  86.18  |
> |           | Inference time (s)          |   3.59  |   3.58  |   3.6   |   3.6   |   3.6   |
> |           | GPU for training (GB)       |   3.21  |   3.21  |   3.21  |   3.21  |   3.21  |
> | FedMAC    | Training time per round (s) |  51.77  |  51.11  |  51.07  |  51.19  |  51.21  |
> |           | Inference time (s)          |   4.56  |   4.98  |   4.69  |   4.69  |   4.88  |
> |           | GPU for training (GB)       |   1.99  |   1.99  |   1.99  |   1.99  |   1.99  |
> | PEPSY     | Training time per round (s) |  141.12 |  153.95 |  137.66 |  140.12 |  146.48 |
> |           | Inference time (s)          |   4.69  |   4.99  |   4.9   |   4.73  |   4.87  |
> |           | GPU for training (GB)       |   2.61  |   2.63  |   2.15  |   2.86  |   2.8   |
>
>
> &nbsp;
>
> > **Q2. How will $\kappa$ , which controls the number of relevant controls selected for each instance, affect the performance?**
>
>
> We provide below additional ablation studies on evaluating the performance of our method with different values of $\kappa$ on the EDF dataset.
>
> |   $\kappa$   |   1   |   3  |   5   |   10  |   15  |   20  |
> |:------------:|:-----:|:----:|:-----:|:-----:|:-----:|:-----:|
> | Accuracy (%) | 47.06 | 48.7 | 51.17 | 52.71 | 55.11 | 52.03 |
>
> The reported results generically show that increasing $\kappa$ tends to improve overall performance as it allows more capacity and flexibility for the data-missing profile to learn local missing patterns. However, beyond some points (e.g., $\kappa = 15$), the information distilled from missing patterns to the missing profile becomes too fragmented (due to selecting too many controllers), which decreases the performance.

---

### Official Review · Reviewer_iy3T · 2025-07-02

**Clarity:** 3
**Significance:** 2
**Originality:** 2
**Rating:** 4
**Confidence:** 4

**Summary:**

This paper proposes to dynamically reconstruct the representation of each client through embedding controls, enabling the global model to maintain stable performance when faced with multiple complex data missing patterns (missing modality、missing feature).

**Questions:**

Please refer to the weakness.

**Ethical Concerns:**

["NO or VERY MINOR ethics concerns only"]

**Final Justification:**

I appreciate the authors' efforts in addressing my concerns. I have carefully read the rebuttal. I will keep my score.

**Limitations:**

Yes

**Quality:**

3

**Strengths And Weaknesses:**

Strength：

（1）This paper addresses the issues of modality missing and feature missing under multimodal federated learning.

（2）Model missing data as structured vectors and enable their sharing within a federated framework.

（3）This paper theoretically analyzed the impact of data missing on prediction output and proposed a clear upper bound of error, enhancing the credibility of the method.

Weakness：

（1）	Although the paper uses PFPT to aggregate the embedding controls of each client, the order and semantics of the controllers learned by each client are not consistent. Does this paper perform semantic interpretability or consistency checks on the embeddings after clustering? If different clients represent different missing semantics, would blind aggregation compromise representation quality?

（2）	The controller selection mechanism is based on cosine similarity matching. Does this similarity matching seek to find a specific controller that matches the current client's missing pattern?

（3）	Without providing the code, it is incredible that performance can be maintained at 16% even with 80% missing data. Which module do you think can maintain model performance in a severely missing data scenario and provide the corresponding basis?

（4）	In the study of multimodal federated learning scenarios, there has been some work on cross-modal and missing modality, but this paper does not provide a thorough review of related work. To my knowledge, there is also some FL+ missing modality work that considers the use of parameters from other clients.

---

> ### Author Rebuttal · Authors · 2025-07-31
>
> > **Q1. The order and semantics of the controllers learned by each client are not consistent?**
>
> We would like to assure that using PFPT as a procedure in our method PEPSY, the order and semantic of the learned and aggregated controllers are consistent, as explained below.
>
> Regarding **order consistency**, by design, our client-side representation reconfiguration mechanism is invariant against controllers' order permutation. Our server-side clustering scheme (via PFPT) is also permutation invariant by design. PFPT identifies and aggregates controllers encoding similar data-missing profiles, rather than aggregating them blindly on a default order.
> Thus, the aggregation outcome won’t be affected by the different orders in which local clients encode data-missing information into their controllers.
>
> Regarding **semantic consistency**, PEPSY, by design, ensures that clients with similar missing profiles will create similar queries defining a particular “missing direction” and hence, the selected controllers of these clients would be aligned with those “missing directions” via the cosine metric.
> As PFPT uses a clustering-based aggregation mechanism which only aggregates controllers in the same cluster, the semantic of those controllers are aligned and can be aggregated without collapsing important information. As such, the aggregated controller in each cluster preserves semantic consistency.
>
> To verify this, **one empirical indication of semantic consistency is the diversity across clusters of controllers**.
> In particular, if controllers with similar semantics are merged appropriately into a centroid, there would be different separated centroids encoding different semantic representatives, leading to increasing diversity in the profile (over time).
> In contrast, violated semantic consistency results in new collapsed embeddings, and fails to build well-separated clusters. This violation, however, is not the case according to Fig. 4b in our paper, which shows the t-SNE plots of the (learned) data-missing profile over 500 communication rounds on PTBXL.
> The plots show that each embedding control follows a different trajectory with increasing diversity (across clusters), suggesting that the data-missing profile aligns semantically similar controllers over multiple rounds.
>
> &nbsp;
>
> > **Q2. The controller selection mechanism is based on cosine similarity matching. Does this similarity matching seek to find a specific controller that matches the current client's missing pattern?**
>
> Semantically, PEPSY ensures that clients with similar missing profiles will create similar queries defining a particular missing direction, leading to different directions for different missing patterns. Hence, the selected controllers of these clients will be aligned with those missing directions based on the directional similarity. This suggests that the cosine metric which measures directional similarities in feature space [1,2] are most suitable in this case.
>
> To ablate the impact of using metric measuring directional similarity, we compare performances of our original design (cosine similarity) and other common distances such as (negative) MSE, MAE as presented below.
>
> |     Relevance Metric     | 0.2/0.2 | 0.8/0.8 |
> |:------------------------:|:-------:|:-------:|
> | Cosine similarity (ours) |  48.76  |  52.78  |
> |          (-) MAE         |  45.18  |  48.83  |
> |          (-) MSE         |  45.01  |  45.75  |
>
> The results show that when we use metrics that are non-directional, the performance decreases. This is because in these cases, clients with similar missing profiles fail to construct a missing direction, hence making the aggregation ineffective.
>
> [1] Zifeng Wang et al., “Learning to Prompt in Continual Learning”. (CVPR 2022)
>
> [2] Xuefei Zhe et al., “Directional statistics-based deep metric learning for image classification and retrieval”, Pattern Recognition, Volume 93, 2019
>
> &nbsp;
>
>
> > **Q3. Without providing the code, it is incredible that performance can be maintained at 16% even with 80% missing data. Which module do you think can maintain model performance in a severely missing data scenario and provide the corresponding basis?**
>
> In our PTBXL experiments with 80/80 missingness, the model achieves reasonable 75%, and 69% accuracy in IID and Non-IID settings, despite substantial data loss.
> To elaborate, we note that [1] has shown that reasonable prediction can be made using data from 1/12 modalities.
> This suggests that for this particular prediction task, essential predictive information can be found within a small fraction (says, 20%) of the dataset.
> Furthermore, the missing patterns are also independent across clients which means even at 80% data loss, the chance of an unrecoverable piece of information being lost across all clients is low.
> Thus, with accurate and comprehensive data-missing profiling providing effective representation reconfiguration, our method is able to integrate the available data across clients effectively, recovering most of the key information and thus preserving reasonable performance despite substantial data loss.
>
> The most important component that helps PEPSY to maintain high performance is the **data-missing profile and the reconfiguration signal**. The core insight is that our approach treats **missingness as structured, learnable information**, rather than as random noise. In other words, each incomplete instance is decomposed as missingness and raw data, both are scattered across clients, and need aggregation to improve overall performance of a given FL system.
>
> The embedding controls form a data-missing profile that encodes each missing pattern as a distinct direction in latent space. These controllers act as functional priors, guiding the model in how to interpret incomplete inputs. Instead of relying solely on observed data, the model leverages this profile to reconfigure biased features to approximate full semantic representations. By explicitly modelling missingness, rather than imputing or ignoring it, the framework remains stable and effective, even under severe data loss.
>
> PEPSY consistently generates data-complete representations across varying missing patterns (Fig. 7). Without the missing-data profile (PEPSY-NP) or reconfiguration signal (PEPSY-NR), the model fails to recover missing information, leading to notably worse performance - especially under high missingness (Tab. 5).
>
> [1] Dongdong Zhang et al., “Interpretable deep learning for automatic diagnosis of 12-lead electrocardiogram”, iScience, 2021.
>
> &nbsp;
>
> > **Q4. This paper does not provide a thorough review of related work. To my knowledge, there is also some FL+ missing modality work that considers the use of parameters from other clients.**
>
> As suggested, we extend our literature review which an be grouped into three main sub-categories.
>
> **Prototype Sharing for Missing Modalities** [1,4]. These methods share class- or modality-level prototypes across clients, which are used for regularization and, in some cases, to impute missing modalities. However, the signals they construct are not tailored to each input and fail to capture missing patterns at the instance level. In contrast, PEPSY enables input-specific reconfiguration through a controller selection mechanism and guided supervision. Moreover, these methods construct prototypes heuristically, resulting in less informative modality representations. Instead, we learn optimized missingness representations that produce more accurate, input-aware signals for reconstructing features tailored to each input. Our method is also the first that establishes a theoretical guarantee for heterogenous missing patterns across clients. Furthermore, these methods [1,4] focus only on modality-missing across clients and do not account for data-missingness.
>
> **Prompt-based Personalization Under Missing Modalities** [3]. This method uses collaborative prompts to personalize pretrained foundation models for clients with different available modalities. While prompts offer some flexibility, the method does not model missingness or adapt to absent data, lacking input-level awareness and useful prompt selection for reconstruction. In contrast, our approach learns a detailed missingness profile from clients using a set of embedding controls. These controls are selected to augment the input representation during inference, allowing fine-grained adaptation at both the instance and modality levels. Additionally, we focus on standard FL rather than prompt-based methods, as no existing foundation model in our health monitoring setting covers all modalities.
>
> **Prototype Sharing for General Heterogeneity** [2,5]. There also exist FL methods that share a similar goal of improving cross-client generalization through global prototype alignment. These approaches focus on unimodal settings and do not account for missing modalities. As a result, they are not well-suited for multimodal scenarios with missing inputs.. In contrast, our method explicitly models each client’s missing modalities using a learned controller set, which is integrated into the training loss. This enables learning and aggregation to be directly informed by the structure of missing data.
>
> [1] Huy Q. Le et al. “Cross-Modal Prototype based Multimodal Federated Learning under Severely Missing Modality”, CoRR 2024
>
> [2] Yue Tan et al. “FedProto: Federated Prototype Learning across Heterogeneous Clients”. AAAI 2022.
>
> [3] Wenli Li et al., “MFLCP: Personalized Multimodal Federated Learning via Collaborative Prompting with Missing Modalities”, ICMR 2025.
>
> [4] Guangyin Bao et al. “Multimodal Federated Learning with Missing Modality via Prototype Mask and Contrast”, arxiv 2024
>
> [5] Jianqing Zhang et al. “FedTGP: Trainable Global Prototypes with Adaptive-Margin-Enhanced Contrastive Learning for Data and Model Heterogeneity in Federated Learning”, AAAI 2024
>
> We will include the above literature review in our revision.

---

### Official Review · Reviewer_nfrN · 2025-07-02

**Clarity:** 3
**Significance:** 3
**Originality:** 4
**Rating:** 4
**Confidence:** 4

**Summary:**

This paper addresses the issue of incomplete and heterogeneous data in multimodal federated learning (MMFL), where clients may possess different modality subsets and suffer from missing features within modalities. The authors propose a novel framework called PEPSY, which leverages learnable client-side embedding controls to encode data-missing patterns and generate locally adaptive representations. These embeddings serve as reconfiguration signals to align globally aggregated representations with each client’s local context, and embeddings from clients with similar missing patterns are algorithmically aggregated to enhance the robustness of reconfiguration signals for global representation adaptation. The key contributions include: developing the PEPSY framework to handle both modality and feature missing scenarios by integrating missing-pattern information into representation reconfiguration without raw data sharing; establishing a theoretical performance bound via theorem proof to demonstrate PEPSY’s stability under random missing patterns by minimizing prediction discrepancies between complete and missing modality cases; and empirically validating that PEPSY achieves up to 36.45% performance improvement over baselines on federated multimodal benchmarks (PTBXL and SleepEDF) under severe data incompleteness, showcasing robust performance in both IID and Non-IID settings. This work provides an effective solution for missing data in MMFL, addressing the gap in existing methods that fail to handle combined modality and feature missing challenges.

**Questions:**

1. The theoretical analysis in the paper is based on the assumption of μ-Lipschitz continuity of the model, which may not fully cover real-world scenarios with complex and irregular data distributions. Additionally, the performance boundaries in extreme missing scenarios (such as when only a single modality is available) are not clearly demonstrated. It is recommended that the authors relax the theoretical assumptions by introducing more universal continuity conditions or validate the model's performance in extreme missing scenarios through experiments simulating complex data distributions, so as to enhance the practical applicability of the theory.

2. The dynamic selection of client-side embedding controls and the server-side PFPT probabilistic clustering algorithm are likely to cause a significant increase in communication overhead in large-scale federated learning scenarios with numerous clients, but the paper does not provide a detailed analysis of the computational efficiency in this regard. It is suggested that the authors supplement the communication complexity assessment in large-scale scenarios or propose targeted lightweight optimization strategies, such as sparse embedding selection or more efficient clustering algorithms, to improve the feasibility of model deployment in practical large-scale environments.

3. The current experiment uses a binary missing matrix generated with fixed probabilities, which fails to fully simulate real-world continuous missing patterns (such as prolonged sensor failures) or inter-modality dependencies (where the absence of one modality may affect the validity of another). It is recommended that the authors introduce more realistic complex missing pattern generation mechanisms in future experiments, such as simulating continuous missing based on temporal dependencies or considering potential correlations between modalities, to verify the model's adaptability in more authentic scenarios.

4. The need for manual tuning of contrastive loss weights (λ, η) may severely limit the model's automated deployment and application across different domains. It is suggested that the authors explore integrating automated tuning mechanisms, such as Bayesian optimization or model-based meta-learning, and optimize the tuning strategy through testing on cross-domain datasets to enhance the model's convenience and performance stability in various application domains.

**Ethical Concerns:**

["NO or VERY MINOR ethics concerns only"]

**Limitations:**

Yes

**Quality:**

3

**Strengths And Weaknesses:**

**Strengths**
1. The PEPSY framework is proposed to address the co-occurrence of modality and feature missing in multimodal federated learning. It encodes data-missing patterns through client-side embedding controls, enabling global aggregated representations to adapt to the local context of each client, which fills the gap of traditional methods in handling mixed missing scenarios.
2. Experiments validate the effectiveness of PEPSY on PTBXL and SleepEDF datasets, achieving up to 36.45% improvement over baselines under severe data incompleteness. It demonstrates robustness to heterogeneous missing patterns in both IID and Non-IID settings, with theoretical analysis consistent with empirical results.

**Weaknesses**
1. The theoretical analysis relies on the assumption of μ-Lipschitz continuity of the model, failing to cover non-smooth data distributions. It also does not clarify the performance bounds in extreme missing scenarios (e.g., only single modality available), and the correlation between the dimension of embedding controls and the complexity of missing patterns lacks theoretical interpretation.
2. The dynamic selection of client-side embedding controls and server-side PFPT probabilistic clustering may lead to a surge in communication overhead in large-scale federated scenarios (e.g., thousands of clients). The weight coefficients of contrastive losses (λ, η) require manual tuning, and the cross-domain application lacks an automated adaptation mechanism.
3. The experiments generate binary missing matrices with fixed probabilities, failing to simulate continuous missing in time-series data or inter-modality dependencies (such as synchronous audio-visual missing). Additionally, it does not verify the adaptability to newly added modalities after training, limiting the expandability in dynamic IoT scenarios.

---

> ### Author Rebuttal · Authors · 2025-07-31
>
> We thank the reviewer for recognizing our contribution and for the constructive feedback. We address all questions from the reviewer below.
>
> &nbsp;
>
> > **Q1. The theoretical analysis in the paper is based on the assumption of μ-Lipschitz continuity of the model, which may not fully cover real-world scenarios with complex and irregular data distributions. Additionally, the performance boundaries in extreme missing scenarios (such as when only a single modality is available) are not clearly demonstrated.**
>
> &nbsp;
>
> ## Lipschitz assumption
>
> Although neural networks are highly nonlinear, the Lipschitz smoothness assumption - i.e., that the gradient of the loss is Lipschitz-continuous - is widely regarded as reasonable for theoretical analysis, especially in federated learning. This is because optimization typically occurs within a bounded parameter region, where the loss landscape is empirically smooth. As such, while global Lipschitz continuity may not hold, local smoothness is often sufficient and observed in practice. Consequently, this assumption is standard in many foundational FL analyses [1, 2, 3].
>
> Having said that, we agree with the Reviewer that it’s beneficial to explore alternative, more relaxed continuity conditions in future work. One potential direction that we will pursue is to explore the semi-smoothness assumption [4], which is a result based on overparameterized neural network theory [5]. This assumption is proved to be held for deep learning in the centralized setting [5] and extended for FL, which could help our bound become more practical.
>
> [1] Tian Li, Anit Kumar Sahu et al., "Federated Optimization in Heterogeneous Networks," MLSys 2020.
>
> [2] Hung T. Nguyen, Vikash Sehwag et al. “Fast-Convergent Federated Learning”, IEEE Transactions on Cognitive Communications and Networking, 2021
>
> [3] Farzin Haddadpour, Mohammad Mahdi Kamani et al. “Federated Learning with Compression: Unified Analysis and Sharp Guarantees”, AISTATS, 2021.
>
> [4] X. Li, Z. Song, R. Tao and G. Zhang, "A Convergence Theory for Federated Average: Beyond Smoothness," 2022 IEEE International Conference on Big Data (Big Data), 2022
>
> [5] Zeyuan Allen-Zhu et al., “A Convergence Theory for Deep Learning via Over-Parameterization”, ICML 2019
>
> &nbsp;
>
> ## Performance bound under extreme case
>
> We would like to demonstrate our performance bound under extremely missing cases. Suppose there is only one modality given an input, i.e., $|S| = |M| - 1$, the RHS of the provided theorem becomes  $ \mathcal{O} \left( \mu (|M| - 1) \sqrt{ \mathbb{E}\_{x, \mathcal{S}} \left[ \mathcal{L}\_{ds}(x, \mathcal{S}) \right] + 2 \log |M| } \right) $, which depends on Lipschitz constant $\mu$ and empirical loss $\mathcal{L}\_{ds}$, aligned with our observations and empirical results in the paper. If we can optimize empirical $\mathbb{E}\_{x, \mathcal{S}} [ \mathcal{L}\_{ds} (x, \mathcal{S}) ] = 0$, i.e., the performance bound now depends only on the smoothness constant $\mu$ of the model (stated in line 214).
>
> &nbsp;
>
> ## Empirical validation on extreme missing scenarios
>
> Furthermore, we presented empirical experiments of multiple extreme modality-missing scenarios, including the case when there is only one modality available for each client (80% of 5 modality are missed), in **Table 7 and Table 8, Appendix E**.
>
> &nbsp;
>
> > **Q2. The dynamic selection of client-side embedding controls and the server-side PFPT probabilistic clustering algorithm are likely to cause a significant increase in communication overhead in large-scale federated learning scenarios with numerous clients.**
>
> The communication overhead over conventional FL methods (e.g., FedAvg)  is negligible in our framework. It also has a minimal effect on the computational cost overall. To see this, let the model size be $O(d)$ in FedAvg, **the conventional communication cost per iteration, per client is then $O(d)$**. Now, in our context, we need to additionally communicate the set of $p$ controllers encoding the data-missing profiles with size $O(d_p)$ each. As $p \times d_p$ is relatively much smaller in comparison to $O(d)$ (e.g., $p \times d_p / d$ ~ 0.1-0.2%), **the overall communication cost remains $O(d)$**. For a thorough comparison, we present numbers of bits in communication and show that the overhead is not significant as below:
>
> | Method   |  0.2/0.2  |  0.4/0.4  |  0.6/0.6  |  0.8/0.8  |
> |----------|:---------:|:---------:|:---------:|:---------:|
> | FedProx  | 125360192 | 125360192 | 125360192 | 125360192 |
> | PEPSY    | 125558930 | 125593828 | 125557128 | 125583219 |
> | Overhead |     0.16% |     0.19% |     0.16% |     0.18% |
>
> &nbsp;
>
> **Time complexity analysis**.
>
> We also appreciate the Reviewer's suggestion on providing a more thorough complexity analysis. It is provided below in comparison to seminal FL work (FedAvg, FedProx) as their communication and compute costs are relatively least expensive. Due to limited space, we would like to refer the Reviewer to our response to Reviewer n9eK for details.
>
> As shown in the analysis, **the added overhead cost grows linearly in the number of sent embedding controls from clients to the server, and is not substantial**. Furthermore, we note that PFPT's hyperparameters can also be tuned to keep server-side computation comparable to traditional FL without harming the performance significantly. As a result, PEPSY still outperform the baseline substantially even when the number of sent embeddings is reduced to match computation cost of FedAvg. Please refer to our response to the Reviewer n9eK for detailed analysis.
>
> &nbsp;
>
> > **Q3. The current experiment uses a binary missing matrix generated with fixed probabilities, which fails to fully simulate real-world continuous missing patterns (such as prolonged sensor failures) or inter-modality dependencies (where the absence of one modality may affect the validity of another).**
>
>
> We agree with the Reviewer that validating the proposed method on more complex scenarios with continuous missing patterns (**inter-modality dependencies**) will broaden its applicability. We also want to emphasize that scenarios **with independent missing patterns (though simpler) also occur in numerous real-world scenarios** such as safety consideration [1], patient dropout [2] or simply high cost [3].
>
> Given this, we believe that the practical impact of our proposed work remains substantial. In our future work, we will evaluate our methods more extensively in the reviewer’s recommended settings.
> For now, we have conducted additional experiments to evaluate our method on simulated scenarios with continuous missing patterns with inter-modality dependencies, following the reviewer’s suggestions. The results are positive as reported below.
>
> For now, we have conducted additional experiments to evaluate our method on simulated scenarios with continuous missing patterns with **inter-modality dependencies**, following the reviewer’s suggestions. The results are positive as reported below.
>
> These are data-missing scenarios where the absence of one modality can influence the validity of another. In particular, we first generate the binary missing matrix which initializes independent missing events. We then simulate inter-modality missing dependencies using the following rule: if modality $x_i$​ is missing and $i+2\leq N$, then modality $x_{i+2}$​ is also set to missing. This enforces that certain modality combinations - such as (1, 3), (2, 4), ..., (N−2, N) - cannot co-occur in the same input. Using this rule and our original matrix generator, we generate a variety of test scenarios. The results are conducted on the EDF dataset (5 modalities) and the performances are presented below.
>
> |  Methods  | 0.4/0.2 | 0.4/0.6 | 0.4/1.0 |
> |:---------:|:-------:|:-------:|:-------:|
> |  FedProx  |  45.56  |  44.61  |  40.84  |
> |    MIFL   |  44.84  |   44.1  |  44.32  |
> |  FedInMM  |  40.96  |   40.5  |  40.56  |
> | FedMSplit |  44.78  |  43.93  |  43.81  |
> |   FedMAC  |  41.76  |   42.9  |  47.18  |
> |   PEPSY   |  53.45  |  54.91  |  53.23  |
>
> **This demonstrates that under inter-modality missing scenarios, PEPSY also consistently outperforms other baselines.**
> Intuitively, this means the more complex and sophisticated patterns of inter-modality missingness can also be distilled effectively into structured data-missing profiles of PEPSY.
> This highlights the robustness and effectiveness of our developed method in practical settings.
>
> [1] Pedro Ramos-Cabrer, JPM Van Duynhoven, A Van der Toorn, and K Nicolay. 2004. MRI of hip prostheses using single-point methods: in vitro studies towards the artifact-free imaging of individuals with metal implants. Magnetic resonance imaging 22, 8 (2004), 1097–1103.
>
> [2] Yongsheng Pan, Mingxia Liu, Yong Xia, and Dinggang Shen. 2021. Disease-image specific Learning for Diagnosis-oriented Neuroimage Synthesis with Incomplete. Multi-Modality Data. IEEE Transactions on Pattern Analysis and Machine Intelligence (2021).
>
> [3] FM Ford and J Ford. 2000. Non-attendance for Social Security medical examination: patients who cannot afford to get better? Occupational medicine 50, 7 (2000), 504–507.
>
> &nbsp;
>
> > **Q4. The need for manual tuning of contrastive loss weights (λ, η) may severely limit the model's automated deployment and application across different domains.**
>
> We thank the Reviewer for the suggestions and constructive feedback. We will explore that in future work where such elaborated tuning can help in scenarios with more sophisticated, inter-modality dependent missing patterns. In the current focus of this paper on independent missing patterns, these parameters are less sensitive according to our earlier investigation. For example, setting $\lambda=0.1, \eta=0.1$ works well for all experiments.
>
> Given the above, we hope the Reviewer can take another look and consider upgrading the rating if our responses have addressed all concerns sufficiently. Otherwise, please let us know if you still have questions for us

---

### Note · Authors · 2025-08-12

We thank the AC and all reviewers for their time and effort in evaluating our paper. In particular, we appreciate Reviewers nfrN, iy3T, and n9eK for retaining their acceptance scores after the rebuttal, and Reviewer iEDv for acknowledging that our rebuttal resolved the clarity issue and adjusting the score accordingly.

We would like to summarize the concerns which we have addressed during the rebuttal.
* Additional Positioning with Relevant Literature (Reviewer iy3T, iEDv): We have provided further positioning of our method with the additional references suggested by the reviewers. We also highlight that our work is the first to directly model missing patterns across clients and build a data-missing profile at a system level, advancing multimodal FL with various missingness levels. This is concretely supported by our comprehensive theoretical analysis and experiments
* Computational Overhead (Reviewer nfrN, n9eK): We have conducted a detailed time complexity analysis and compared our approach with traditional FL algorithms, demonstrating that its computational overhead is minimal and can be further optimized to match that of conventional methods.
* Theoretical Assumption (Reviewer nfrN, iEDv): We have demonstrated that Lipschitz assumptions are widely adopted as a standard approach for analyzing the performance of FL algorithms[1,2]. Following the reviewers’ feedback, we have discussed a promising alternative that could further enhance our theoretical framework in future work.
* Additional Experiments on Different Domains/Settings and Ablation Studies (Reviewer nfrN, iy3T, n9eK, iEDv): In response to the reviewers’ questions, we conducted additional experiments across diverse domains and settings, which consistently demonstrate the impact of important components in our workflow and also show that our approach significantly outperforms baseline methods.
* Clarity (Reviewer iEDv): We have addressed all concerns regarding clarity in both the writing and the experimental settings. As suggested by the reviewer, we will further refine the manuscript to improve clarity in the final revision.

[1] Tian Li et al., "Federated Optimization in Heterogeneous Networks," MLSys 2020.

[2] Farzin Haddadpour et al. “Federated Learning with Compression: Unified Analysis and Sharp Guarantees”, AISTATS, 2021.

---

### Decision · Program_Chairs · 2025-09-17

**Decision:**

Accept (poster)

**Comment:**

After the discussion phase all reviewers agree that the submission is of high enough quality for publication at NeurIPS. As Area Chair, I agree and recommend acceptance.